# Plasmonic metasurfaces of cellulose nanocrystal matrices with quadrants of aligned gold nanorods for photothermal anti-icing

Jeongsu Pyeon [1,5], Soon Mo Park[2,3,5], Juri Kim[4], Jeong-Hwan Kim [1], Yong-Jin Yoon[1], Dong Ki Yoon [3,4] ✉ & Hyoungsoo Kim [1] ✉

Cellulose nanocrystals (CNCs) are intriguing as a matrix for plasmonic meta-surfaces made of gold nanorods (GNRs) because of their distinctive properties, including renewability, biodegradability, non-toxicity, and low cost. Nevertheless, it is very difficult to precisely regulate the positioning and orientation of CNCs on the substrate in a consistent pattern. In this study, CNCs and GNRs, which exhibit tunable optical and anti-icing capabilities, are employed to manufacture a uniform plasmonic metasurface using a drop-casting technique. Two physical phenomena—(i) spontaneous and rapid self-dewetting and (ii) evaporation-induced self-assembly—are used to accomplish this. Additionally, we improve the CNC-GNR ink composition and determine the crucial coating parameters necessary to balance the two physical mechanisms in order to produce thin films without coffee rings. The final homogeneous CNC-GNR film has consistent annular ring patterns with plasmonic quadrant hues that are properly aligned, which enhances plasmonic photothermal effects. The CNC-GNR multi-array platform offers above-zero temperatures on a substrate that is subcooled below the freezing point. The current study presents a physicochemical approach for functional nanomaterial-based CNC control.

Cellulose nanocrystals are the most abundant functional building block in nature (Fig. 1a), which have been highlighted in materials science and engineering due to their molecular functionalities and structural anisotropy[1]. The crystallized cellulose derived from wood and bacteria has various molecular functional groups that can interact with other additives[2,3]. They are densely packed on the surface, with the chance to be altered by postmodification (e.g., acid hydrolysis and organic synthesis)[4,5]. And the rod-like CNCs develop liquid crystal (LC) phases above a certain concentration[4,5], fitting the criterion in Onsager's theory[6] that pairs of anisotropic colloids have

the tendency to minimize their excluded volume. The self-assembling behavior of CNCs forms a hierarchy of length scales, from nanometer-scale rod-like crystals to millimeter-scale nematic ordering, and provides structural anisotropy to the functional materials for optical[7–9], thermal[10], and mechanical applications[11,12]. However, to make the most of these advantages, it is necessary to have the techniques to create a well-organized and directional CNC matrix for specific applications. For this reason, various studies are currently underway in the engineering field to control CNC ordering and alignment.

[1]Department of Mechanical Engineering, Korea Advanced Institute of Science and Technology, Daejeon 34141, Republic of Korea. [2]Department of Chemical and Biomolecular Engineering, Cornell University, Ithaca, NY 14853, USA. [3]Graduate School of Nanoscience and Technology, Korea Advanced Institute of Science and Technology, Daejeon 34141, Republic of Korea. [4]Department of Chemistry, Korea Advanced Institute of Science and Technology, Daejeon 34141, Republic of Korea. [5]These authors contributed equally: Jeongsu Pyeon, Soon Mo Park. ✉e-mail: nandk@kaist.ac.kr; hshk@kaist.ac.kr

**Fig. 1 | A homogeneous quadrant circular CNC matrix. a** Schematic illustration of CNC template fabrication processes. A tree image by macrovector and a vial image by vectorpocket, both on Freepik were used. **b–d** Sequential changes of CNC optical textures in real-time using POM with a retardation plate. The POM images were compared for different MeOH concentrations [(**b**) 0, (**c**) 30, and (**d**) 70 vol.% in DI water] with CNCs (2.85 wt%) added to the liquid solution for all cases. The fifth column of (**b–d**) showed the final CNC crystalline structures after all liquid solvents completely evaporated ($t_e$: total evaporation time). **e** Comparison of drying uniformity of the CNC matrices depending on the initial CNC concentration ($C_{CNC}$). All the scale bars are 500 μm.

Here, we propose a facile strategy for manufacturing CNC matrix films in which all CNCs are well-aligned and exhibit excellent uniformity upon drying. To achieve this, we adopt the evaporation-induced self-assembly (EISA) technique[7–9,11,13,14] due to its simplicity in fabrication. During evaporation, the collective behavior of anisotropic colloidal particles occurs, leaving a dried film pattern. This advantage makes EISA widely utilized for aligning bio-based anisotropic building blocks,

including CNCs[9,15–17]. Especially, a drop-casting method is broadly used due to its simplicity, ease, and rapid streamlined procedure. When a droplet is drop-cast onto a solid substrate and then undergoes evaporation, the liquid-gas interface of the droplet typically takes on a curved shape if the surface tension effect dominates over the gravitational effect, specifically if the Bond number is less than unity and there is a hydrophilic contact angle. The low contact angle (< 45°) and thin

droplet interface shape lead to a nonuniform evaporative flux along the droplet interface, creating an evaporatively-driven capillary flow inside the evaporating droplet. In this situation, the CNCs align in a uniaxial direction rather than a helical structure[18–20]. However, this evaporatively-driven capillary flow (referred to as a coffee-ring flow[21]) resulted in an inhomogeneous dried morphology[22] akin to a coffee-ring stain. This issue is a critical hurdle in coating and patterning applications.

Recently, several potential solutions have been suggested to achieve uniformity, i.e., (i) solutal-Marangoni flows generated by a volatile vapor-saturated environment[23,24], (ii) thermal-Marangoni effects on a heated solid substrate[25,26], and (iii) viscosity enhancement by gelation[27,28]. But, unfortunately, each method has limitations. In general, the solutal-Marangoni effect generates uncontrollable flow structures[29] due to its inherent instabilities at a liquid-gas interface. For this reason, the self-assembling mechanism of CNC particles is systematically unstable for repetitive mass production of CNC matrices. For the case of the heating method, internal flows driven by thermal-Marangoni stresses of approximately $1-10~\mu m~s^{-1}$ [25] are not sufficiently strong to restrain the coffee-ring flows of approximately $1~\mu m~s^{-1}$ [30] compared to the solutal-Marangoni cases of approximately $10-100~\mu m~s^{-1}$ [31,32]. Furthermore, it is very delicate and fastidious to control the temperature uniformity using external facilities. Lastly, it is unclear what effect temperature has on the self-assembled CNC structure. Besides, adding gelling agents (salts[27] and glucose[28]) to a CNC coating solution poses a risk of interfering with the control of the self-assembly process of the CNCs. The interactions between the CNCs and the agents can increase the complexity of the CNCs' self-assembly. Also, the introduction of the agents can potentially lead to the incorporation of impurities into the final CNC matrix template. Therefore, to produce a systematically uniform CNC matrix with the physicochemical stability of CNCs, it is important to minimize the external force applied to the CNC materials and reduce the process complexity. Thus, there is still a need to develop a CNC deposition method that utilizes simple evaporation under room temperature and atmospheric pressure.

In addition, many efforts have been made to modify the properties (e.g., wettability[33,34], anticorrosion[35], mechanical[35,36], thermal[36], and electrical[37] properties) of target surfaces using various coating techniques. In particular, there is a growing interest in patterning functional nanomaterials driven by their advantages such as transparency (i.e., small particle size), improved interaction between the coating materials and the substrate, and enhanced durability of the coating layer. Under this circumstance, we realized that although GNRs were considered one of the most useful and promising nanoparticles due to their biocompatible, chemically stable, relatively easy synthesis, and tunable surface plasmon resonance (SPR), achieving a uniform GNR film remains a significant challenge[38,39]. Previous studies have demonstrated success in vertically aligning GNRs through long evaporation times; however, this approach severely hampers productivity[40,41]. Therefore, our objective is to employ a CNC template film to create a homogeneous plasmonic metasurface of GNRs and harness the plasmonic heating effects. Here, the plasmonic effect is highly competitive as it does not require any special surface modification compared to conventional anti-icing systems, such as (i) chemically coated smooth surfaces using self-assembled monolayers (SAMs)[42] or physically coated surfaces using spin and spray techniques[43], (ii) textured surfaces[44–47], and (iii) liquid-infused slippery wet surfaces[45,47,48]. These complex surfaces help prevent frost problems due to their low water wettability (i.e., high water repellency), but they have many challenges under specific circumstances. For instance, when a self-assembled monolayer is deposited on a target substrate, the compatibility (i.e., chemisorption) between the layer materials and substrates should be taken into consideration. Additionally, larger surfaces and confined areas of textured substrates can increase the rate of condensation below the dew point, which leads to the trapping of condensed liquid drops in a Wenzel state[49]. Moreover, the liquid-infused surfaces require lubricant replenishment, posing

challenges and prerequisites[47]. Based on this, we speculate that the plasmonic effect of the CNC-GNR film will be significantly superior to previous anti-icing designs.

In this study, we develop a facile and mass-producible technique for achieving uniform quadrant CNC-GNR metasurfaces using the drop-casting technique without any complex manufacturing processes or preconditions. We accomplish this by exploiting the spontaneous and fast self-dewetting of CNC-containing methanol (MeOH) and deionized (DI) water mixture droplets. During the evaporation process, the dewetting motion not only aligns the suspended CNCs but also uniformly deposits them. Meanwhile, the EISA phenomenon is continuously generated near the dewetting contact line. We observe that the spontaneous dewetting contact line feature occurs at the relatively high concentration of highly volatile liquids (in this case, MeOH)[50], while the CNC particles are uniformly self-assembled and crystallized along the moving contact line. Here, due to the high vapor pressure of MeOH and its selective evaporation, more MeOH molecules tend to move to the contact line[51,52]. This evaporative-triggered segregation induces the spontaneous self-dewetting motion in the early stages of evaporation. Subsequently, the contact line dewets smoothly without any stick-slip motion due to several factors, including the hydrophilic surface of the CNCs[53], the relatively low concentration of CNCs near the moving contact line[54], which is not sufficient to cause self-pinning behavior, and the fast evaporation of MeOH. To sum up, the key factor of the self-aligned and uniform CNC matrix is the controlled deposition rate of nanoparticles at the self-receding contact line due to the empirically optimized ink composition.

Besides, using polarized optical microscopy (POM), we measure the dewetting speed ($U_d$) and compare it with the speed of coffee-ring flows ($U_c$) close to a contact line. Here, the magnitude of the coffee-ring flow speed is estimated from theoretical models[30] and experimental results[55] of previous studies. To validate this, we also conduct micro-particle image velocimetry ($\mu$-PIV) experiments that provide a good agreement between the flow field measurement results and the theoretical values. Through speed analysis, we find that the uniform CNC matrices are made if $U_d \geq U_c$. In this situation, the coffee-ring flow causes the accumulation of CNC particles near the contact line, resulting in the LC transition and crystallization. At the same time, a relatively fast dewetting motion leads to the formation of homogeneous CNC matrices. Based on this critical coating condition, we finally acquire a co-assembled GNR film in the CNC matrix with excellent uniformity, where all GNRs are well-oriented parallel to CNCs. This plasmonic metasurface shows homogeneous quadrant-shaped plasmonic color changes depending on polarization conditions. Furthermore, using a thermal imaging camera, we also validate that the homogeneous CNC-GNR metasurface exhibits better plasmonic photothermal performance[56] compared to a conventional ring-shaped CNC-GNR film when exposed to visible light across a broad range of wavelengths. Here, we show that if the CNC-GNR metasurfaces are deposited in multiple arrangements, the surface temperature is above zero ($\approx 5-8$ °C), although the bottom substrate has $-8$ °C. The multi-array CNC-GNR metasurface is experimentally demonstrated to show potential for anti- and de-icing capabilities. From this result, we believe that this drop-casting CNC-GNR metasurface array can be potentially used as a plasmonic photothermal film for anti-icing applications.

## Results

### Spontaneous and fast dewetting by selective MeOH evaporation
In this work, to achieve a uniform and well-aligned CNC film, we used a water-based binary mixture solution containing a highly volatile liquid solvent [here, MeOH], in which the CNC particles were well-dispersed. The sample preparation process for the CNC film is summarized (see also Fig. 1a): (i) First, we dispersed rod-like CNCs in deionized (DI) water, (ii) Next, we added MeOH to the DI water solution dispersed

with CNCs. (iii) After that, bath sonication was performed for 30 min to disperse the CNCs well in the solution of MeOH and DI water mixture. (iv) Using a drop-casting method, we dropped a CNC-containing droplet with $2.00 \pm 0.04$ μL on a glass substrate and evaporated it at room temperature ($\approx 22$ °C). Here, the size of the droplets was controlled to maintain a diameter of 3 mm by selective plasma treatment (i.e., hydrophilic surface treatment) on the substrate (see detailed processes in the Methods). During the evaporation process, the droplet and the measurement area were always covered by an acrylic box to prevent any disturbances that could have affected the droplet evaporation and internal flow.

It is well-known that selective evaporation of the highly volatile solvent component occurs in the binary mixture, which can trigger phase segregation[51,52] near the contact line, resulting in the production of Marangoni stress. Eventually, the selective evaporation of MeOH near the contact line induces the spontaneous self-dewetting of the contact line. The preceding study reported that the receding contact line driven by self-dewetting of evaporating droplets can make tangential alignments of the CNC particles (i.e., parallel to the contact line) rather than radial alignments[20]. Tangential alignment with an annular pattern occurs near the triple-phase contact line, which is the boundary between the droplet, the surrounding air, and the substrate. As the droplet evaporated, the contact angle became smaller and the meniscus adopted a wedge-shaped configuration, leading to splay deformation in the CNCs through the radially outward capillary flows. In this process, there was a major competition between the dilative stress along the $r$-direction near the retreating contact line and the elasticity of the CNCs. It is widely recognized that anisotropic particles, such as DNA[15], virus[16], and carbon nanotubes (CNTs)[57], dispersed in droplets possess higher elastic free energy associated with splay deformation in comparison to bend and twist deformation during evaporation. Thus, it can be inferred that the CNCs exhibit similar elastic behavior to the anisotropic metarials[15,16,57]. The CNC's structure was adjusted to achieve an energetically favorable and stable state by minimizing the changes in its elastic free energy in response to the distortions. Consequently, the CNCs withstand the stress induced by splay deformation and instead exhibit an annular alignment (ring-like patterns) as the process of evaporation continues. Based on this alignment mechanism, the spontaneous and fast self-dewetting driven by selective evaporation MeOH near the contact line helps to create a uniformly aligned CNC pattern.

To investigate the effect of the spontaneous and fast self-dewetting of the evaporating binary mixture drops on the morphology of a dried CNC film, we recorded and compared sequential POM results for different cases depending on the MeOH concentration (0, 30, and 70 vol.%) of the CNC solution as shown in Fig. 1b–d (see the setup information of POM measurements in the Methods). The vapor pressure of MeOH was much higher than that of DI water (i.e., $P_{v,MeOH} \approx 13.02$ kPa[58] $\gg P_{v,water} \approx 2.33$ kPa[59] at surrounding temperature 22 °C), so the self-dewetting and the decrease of the droplet volume were mainly due to the evaporation of MeOH during the initial stage of the evaporation. As the initial concentration of MeOH was high, the evaporation rate of the drops increased, the droplet quickly evaporated, and then the self-dewetting from the contact line was rapidly triggered by the fast evaporation. For the case of the DI water droplet evaporation, all CNCs were intensively stacked near an initial contact line due to its slow dewetting process (dewetting occurs at $t \approx 550$–600 s, see Supplementary Movie 1). In this case, highly concentrated CNCs anchored near the contact line, which hindered the dewetting behavior[60]. Subsequently, we conducted tests with high concentrations of 30% and 70% MeOH by volume ratio. We observed that the self-dewetting began earlier than the DI water droplet case (dewetting starts at $t \approx 240$–260 s for 30 vol.% MeOH, and at $t \approx 0$–30 s for 70 vol.% MeOH, see Supplementary Movies 2–3) as shown in Fig. 1c–d. As the concentration of MeOH increased further, POM results showed that CNC birefringence colors

(blue and yellow domains) more covered the middle of the droplet area (see Fig. 1b–d and Supplementary Movies 1–3). Here, the self-dewetting occurred due to the high volatility of MeOH, where the self-dewetting force driven by the fast evaporation became predominant compared to the self-pinning force caused by anchoring CNCs near the contact line (i.e., self-dewetting force $\gg$ self-pinning force)[54]. Once the contact line was dewetted, the CNCs were continuously self-assembled along the moving contact line without stick-slip motions[60]. This continuous dewetting phenomenon can be caused by hydrophilic CNC particles, not enough concentration of CNC particles at the contact line, and fast evaporation of solvents. It is reported that the stick-slip phenomenon can be promoted if the particle surface has less hydrophilicity[53] or the initial particle concentration is very high[54]. Thus, the self-pinning effect was minor in the current condition, resulting in a freely receding contact line toward the center of the drop without discontinuous motions. As a consequence, we finally suppressed the ring-like CNC structures and created much pronounced CNC textures in the vicinity of the droplet center. In fact, to obtain uniform and clear CNC birefringence quadrant color patterns, the initial concentration of CNC particles is crucial. As illustrated in Fig. 1d, when the concentration of CNCs was insufficient, there were not enough CNCs deposited in the center of the droplet, which rendered it unsuitable as a host template for co-assembly with functional materials. To overcome this issue, we increased the CNC concentration as depicted in Fig. 1e. Our experiments revealed that a CNC concentration of approximately 3.40 wt% led to more homogeneous CNC patterns, as evidenced by Supplementary Movie 4 and side-view SEM images in Supplementary Fig. 3. To sum up, the optimal condition for the coating ink compositions was set as MeOH : DI water : CNCs = 62.75 : 33.85 : 3.40 wt% or 68.71 : 29.45 : 1.84 vol.% to obtain uniform quadrant CNC matrices through the drop-casting process.

## Critical condition for homogeneous CNC matrices

To comprehend the mechanism of uniform patterning, it is important to understand the drying process. We have summarized the evaporation and deposition processes of CNC particles in Fig. 2. During the early stages of evaporation, solutal-Marangoni flows occurred due to the selective evaporation of MeOH, as shown in Fig. 2a. Because the solutal-Marangoni effect became predominant compared to the diffusion effect, with Pe (Péclet number) = $UR/D_{12} \gg 1$ and Ma (Marangoni number) = $\Delta\gamma R/\mu D_{12} \gg 1$, where the surface tension gradient $\Delta\gamma \approx \mathcal{O}(10^{-1}$ mN m$^{-1})$[32], the droplet radius $R \approx \mathcal{O}(10^{-3}$ m), the fluid viscosity $\mu \approx \mathcal{O}(10^{-1}$ mPa s), the flow speed $U \approx \mathcal{O}(10^{-4}$ m s$^{-1}$), and the mutual diffusion coefficient $D_{12}$ of binary mixture of the MeOH (species 1) and DI water (species 2) ($\approx 1.30 \times 10^{-9}$ m$^2$ s$^{-1}$)[61]. Thus, initially, the suspended CNC particles were well-mixed by solutal-Marangoni flows driven by surface tension gradients between the MeOH and DI water components along the liquid-gas interface due to Marangoni instabilities[29,62]. Simultaneously, according to the non-uniform evaporative flux[21], in general, the surface tension might have been at the maximum value ($\gamma_{high}$) at the contact line, while the surface tension might have been at the minimum value ($\gamma_{low}$) at the apex of the droplet due to the selective evaporation of MeOH. However, the complicated mixing flows were observed due to Marangoni instabilities along the droplet interface, rather than circulating flows from the top of the droplet towards the contact line. This mixing pattern resulted from the interplay between the non-uniform evaporative flux and the Marangoni instabilities at the liquid-gas interface. In the subsequent stage (Fig. 2b), as the droplet evaporated, MeOH rapidly evaporated and the droplet volume decreased while keeping a constant wetting area. After a few seconds, self-dewetting occurred due to the relatively fast evaporation of MeOH in the evaporating droplet. After this stage, we observed that the contact line uniformly receded in the direction of the droplet center (see greenish arrows in Fig. 2c). At the same time, evaporatively-driven coffee-ring flow occurred toward the moving contact line (see the arrows in Fig. 2c). The solutal-Marangoni stresses became milder as the

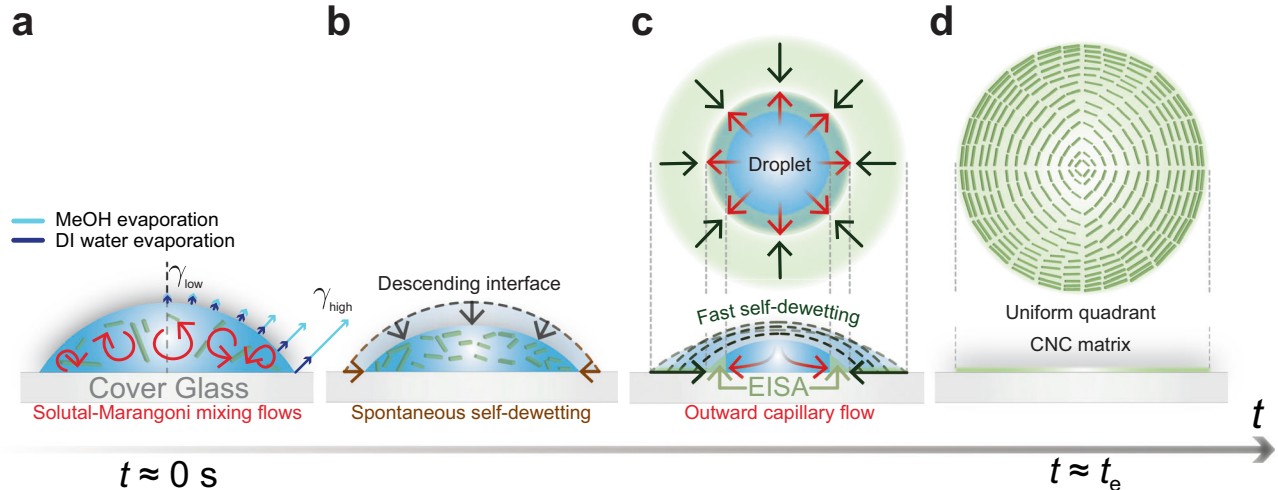

**Fig. 2 | Deposition principle for the formation of a homogeneous quadrant CNC matrix.** Hydrodynamic mechanisms for drying homogeneity of the CNC matrix. It undergoes the following steps: **a** solutal-Marangoni flows generated by selective evaporation mixed the suspended CNCs well, **b** the droplet contact angle was decreased due to fast evaporation of volatile liquid components of evaporating drops while keeping the wetting area. A few seconds later, the spontaneous self-dewetting of the initial contact line was induced. **c** Evaporation-induced self-assembly (EISA) occurred in the vicinity of the moving contact line as a result of competition between the fast self-dewetting (inward direction) and evaporatively-driven capillary flow (outward direction), and **d** the uniformly distributed quadrant-shaped CNC matrix was achieved.

concentration of the highly volatile liquid component, which had a lower surface tension, decreased. The Marangoni flows disappeared completely when the contact line continuously dewetted[29,63]. Experimental results for MeOH at 70 vol.% in water showed that the solutal-Marangoni flows vanished completely at $t/t_e < 0.25$, where $t_e$ is the total evaporation time (see Supplementary Movie 3). In the third regime (Fig. 2c), two opposite speeds of self-dewetting (inward) $U_d$ and coffee-ring flows (outward) $U_c$ competed with each other, which determined whether the final dried CNC films were uniform or not. If the speed of the coffee-ring flows $U_c$ was much faster than the dewetting speed $U_d$ (i.e., $U_d \ll U_c$), CNC particles accumulated intensively near the initial droplet contact line and a ring-shaped CNC crystalline pattern was obtained, whereas when the dewetting speed $U_d$ was comparable to the coffee-ring flow speed $U_c$ (i.e., $U_d \geq U_c$), two physical phenomena, i.e., the deposition of CNC materials on the substrate and the motion of dewetting contact line, occurred at the same time. As a result, we obtained a uniformly crystallized CNC matrix in a quadrant orientation as illustrated in Fig. 2d, without any changes to their LC structures (see the blue and yellow textures of Fig. 1e and the AFM images of Supplementary Fig. 4e).

In the previous paragraph, we mentioned that the speed ratio ($=U_d/U_c$) is an important parameter for achieving a homogeneous CNC matrix. Therefore, we calculated these two speeds from the POM images in Fig. 1b–d using the following process. First, we estimated the coffee-ring flow speed $U_c$ close to the contact line by applying the theoretical model from a previous study[30] (see the detailed procedure in Methods). The results showed that $U_c$ was on the order of 1 µm s⁻¹ for the MeOH and DI water mixture and 0.1 µm s⁻¹ for pure water during the early stages of evaporation. Here, we can presume that the coffee-ring flow speed $U_c$ would not vary considerably until the droplets were entirely evaporated because the droplet freely dewetted in both cases, as observed in a preceding study[55]. In the regime of continuous dewetting, the variation of the contact angle was relatively small. For instance, droplets containing CNCs in a mixture of MeOH and DI water (= 70 : 30 vol.%) on a glass substrate experienced a contact angle variation of only around 5° while they were dewetted, as shown in the regime (2) of Supplementary Fig. 1 and Supplementary Movie 5. So, the shape of the droplets was not significantly changed just before the complete evaporation. To further support our estimation, we

conducted PIV experiments after the intense Marangoni mixing flows (see the results in Fig. 3a). In PIV results, a nearly constant coffee-ring flow speed $U_c$ was observed until the droplet completely evaporated, as shown in the (4) regime of Fig. 3a and Supplementary Movie 6.

However, in the case of evaporation of binary mixture drops (square and hexagonal symbols of Fig. 3b–c), the calculation of $U_c$ was different for the early ($t/t_e < 0.25$) and mid-late ($0.25 \leqslant t/t_e \leqslant 1.00$) evaporation stages, respectively. This was because the MeOH component evaporated quickly during the initial stage, and almost only the DI water component remained thereafter (see the sharp decrease in solutal-Marangoni flows in Supplementary Movies 2–3). Consequently, we redefined $U_c$ for MeOH and DI water mixture drops as follows: $U_c \approx \mathcal{O}(1 \, \mu m \, s^{-1})$ at $t/t_e < 0.25$ and $U_c \approx \mathcal{O}(0.1 \, \mu m \, s^{-1})$ at $0.25 \leqslant t/t_e \leqslant 1.00$. The PIV results presented in Fig. 3a also confirmed that, after the solutal-Marangoni flows almost disappeared, the coffee-ring flows were observed, which showed an average speed $U_c$ of approximately 1–2 µm s⁻¹ [(1) and (2) regimes in Fig. 3a]. Afterward, MeOH components totally evaporated, and then the remaining pure DI water evaporation caused the coffee-ring flows, which exhibited an average speed $U_c$ of approximately 0.4–1 µm s⁻¹ [(3) and (4) regimes in Fig. 3a], consistent with our previous prediction. Next, we measured the dewetting speed $U_d$ by tracking the growth of CNC crystal structures (blue and yellow textures) from sequential POM images of Fig. 1b–d, respectively (see Fig. 3b). Finally, we obtained a time-dependent speed ratio $U_d/U_c$ with three different MeOH concentrations (0, 30, and 70 vol.%) in the water, as shown in the graph in Fig. 3c. Based on the graph, we found that achieving uniform CNC matrices was possible only when the self-dewetting was continuously generated and balanced with EISA in the vicinity of the contact line driven by the coffee-ring flows, which satisfied the critical condition, $U_d/U_c \geq 1$ (see the green dashed arrow in Fig. 3c). On the other hand, when this condition was not met, $U_d/U_c \ll 1$, we observed ring-shaped crystalline CNC patterns (see the red dashed arrow in Fig. 3c). This critical condition was further corroborated by the film thickness profiles depicted in Fig. 3d.

## Plasmonic metasurfaces of CNC-GNR films
We used an evaporation-induced self-assembly fabrication method, driven by the equilibrium between two evaporation phenomena:

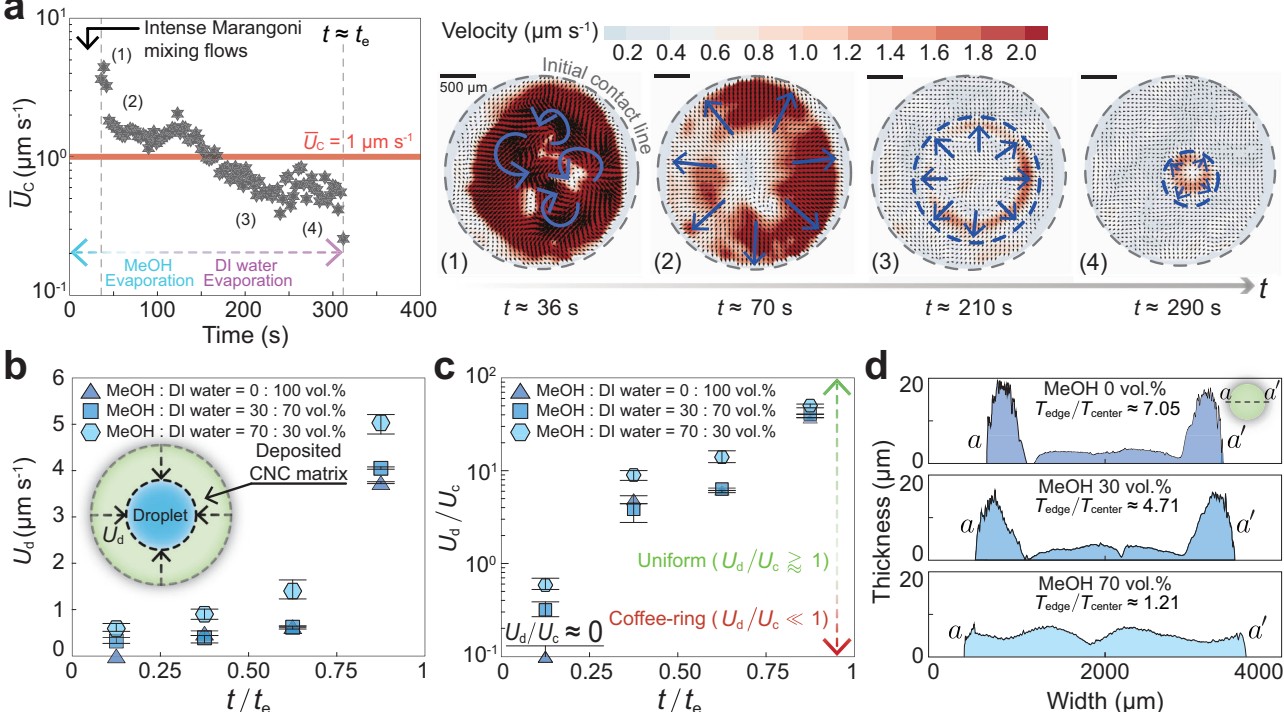

**Fig. 3 | Critical condition for drying morphology (uniform pattern Vs. coffee-ring pattern) of a CNC matrix depending on the dewetting speed ($U_d$) and the coffee-ring flow speed ($U_c$). a** Results of micro-particle image velocimetry ($\mu$-PIV) experiments for droplet evaporation of MeOH and DI water mixture (70 : 30 vol.%) containing 3.40 wt% CNCs observed after the solutal-Marangoni flows almost disappeared at $t \approx 36$ s. We observed the temporal evolution of average $U_c$. The four sequential flow field snapshots (1)-(4) were summarized, depicting the weakening of the solutal-Marangoni mixing flow in (1), followed by the formation of a coffee-ring flow in (2). The coffee-ring flow persisted in (3) and (4) due to the evaporation of the remaining DI water components. Typical flow structures were indicated by blue arrows and moving contact lines were marked with blue dashed circles. **b** Dewetting speed $U_d$ depending on MeOH concentration (0, 30, and 70 vol.%) in DI water dispersed with 2.85 wt% CNC. **c** Relative speed ratio (= $U_d/U_c$) depending on MeOH concentrations (0, 30, and 70 vol.%) with 2.85 wt% CNC. The results showed that the coffee-ring effect ($U_d/U_c \ll 1$) became predominant in the direction of the red dashed arrow. When the critical condition was satisfied ($U_d/U_c \gtrsim 1$), all CNCs were uniformly deposited and oriented during evaporation, represented by the green dashed line. $t_e$ indicated the time when the CNC-containing drops were completely evaporated. All error bars were obtained from four samples through independent experiments. **d** Film thickness profiles of (**c**) along the $a$-$a'$ line (in this case, $C_{CNC} \approx 3.40$ wt%). $T_{edge}/T_{center}$ represents the ratio of film thickness between the edge and the center regions. Source data are provided as a Source Data file.

spontaneous and rapid self-dewetting with freely receding motion and coffee-ring flow, to obtain a uniform quadrant CNC matrix on the substrate. To produce self-aligned CNC films without coffee-ring formation, we established the critical coating condition $U_d/U_c \geq 1$ (see details in Fig. 3). Under this coating condition, we successfully fabricated both optical (Fig. 4) and thermal (Fig. 5) plasmonic metasurfaces by adding GNRs in a MeOH and DI water mixture-based CNC coating solution. In the absence of the CNCs, most of the GNR particles were accumulated at the contact line, i.e., coffee-ring stain, as shown in Supplementary Fig. 6. Then, uniform deposition and alignment of GNR particles cannot be achieved, as confirmed in Supplementary Fig. 6. Figure 4a outlines the procedures in detail, which are also elaborated on in the Methods section. We used a CNC-GNR solution drop with a volume of $2.00 \pm 0.04$ μL on a smooth glass substrate (MeOH : DI water : CNCs : GNRs = 62.57 : 33.76 : 3.39 : 0.28 wt% or 68.70 : 29.44 : 1.84 : 0.01 vol.%). From the POM measurements in Fig. 4b, we confirmed that the speed ratio $U_d/U_c$ satisfied the critical condition during the evaporation process (see Supplementary Fig. 5c-e and Supplementary Movie 7-8), resulting in a uniformly dried CNC matrix with quadrant alignments, as shown in Fig. 4b (see the film thickness profile result in Supplementary Fig. 5f).

Next, to evaluate the potential of the co-assembled GNRs in the CNC matrix as a plasmonic optical metasurface, we measured and compared the non-polarized (Fig. 4c) and linearly polarized (Fig. 4d) optical microscope (OM) images of the CNC-GNR films, respectively. Here, the anisotropic GNRs exhibited two localized surface plasmon

resonance (LSPR) peaks (in this case, longitudinal plasmon band: 665 nm and transverse plasmon band: 515 nm) due to its anisotropic shape (see the SEM image in Supplementary Fig. 7a and the TEM image in Supplementary Fig. 13b). Thus, it emitted two different plasmonic optical colors depending on the polarization angles as well as wavelengths (or frequency) of the incident light beam. Based on the above-mentioned LSPR mechanism, in the absence of a polarizer, a homogeneous dark violet color was observed because the longitudinal and transverse LSPRs simultaneously occurred due to non-polarized incident light, as shown in Fig. 4c. However, upon irradiation with linearly polarized light to the CNC-GNR composite film, the two plasmonic colors alternately appeared in the shape of a quadrant (the quadrants 2 and 4, transverse LSPR and light violet color, and the quadrants 1 and 3, longitudinal LSPR and blue color). The results of UV-Vis absorption spectra confirmed that CNCs did not cause a significant shift in the peak of the original absorption spectrum of the GNRs (comparing Fig. 4a and Supplementary Fig. 8). This phenomenon occurred because anisotropic GNRs followed the alignment of the CNCs' structure. Here, the GNRs tended to align parallel to the CNCs in a way that maximized their freedom of movement and were not able to occupy the same space as the CNCs at the same time (so-called entropic and excluded volume effect)[64] [see the illustration (1) and (2) of Fig. 4d]. Additionally, SEM images of the CNC-GNR metasurface were provided in Supplementary Fig. 7b, which offers a more detailed visualization. We also demonstrated continuous plasmonic color changes of the CNC-GNR metasurface by

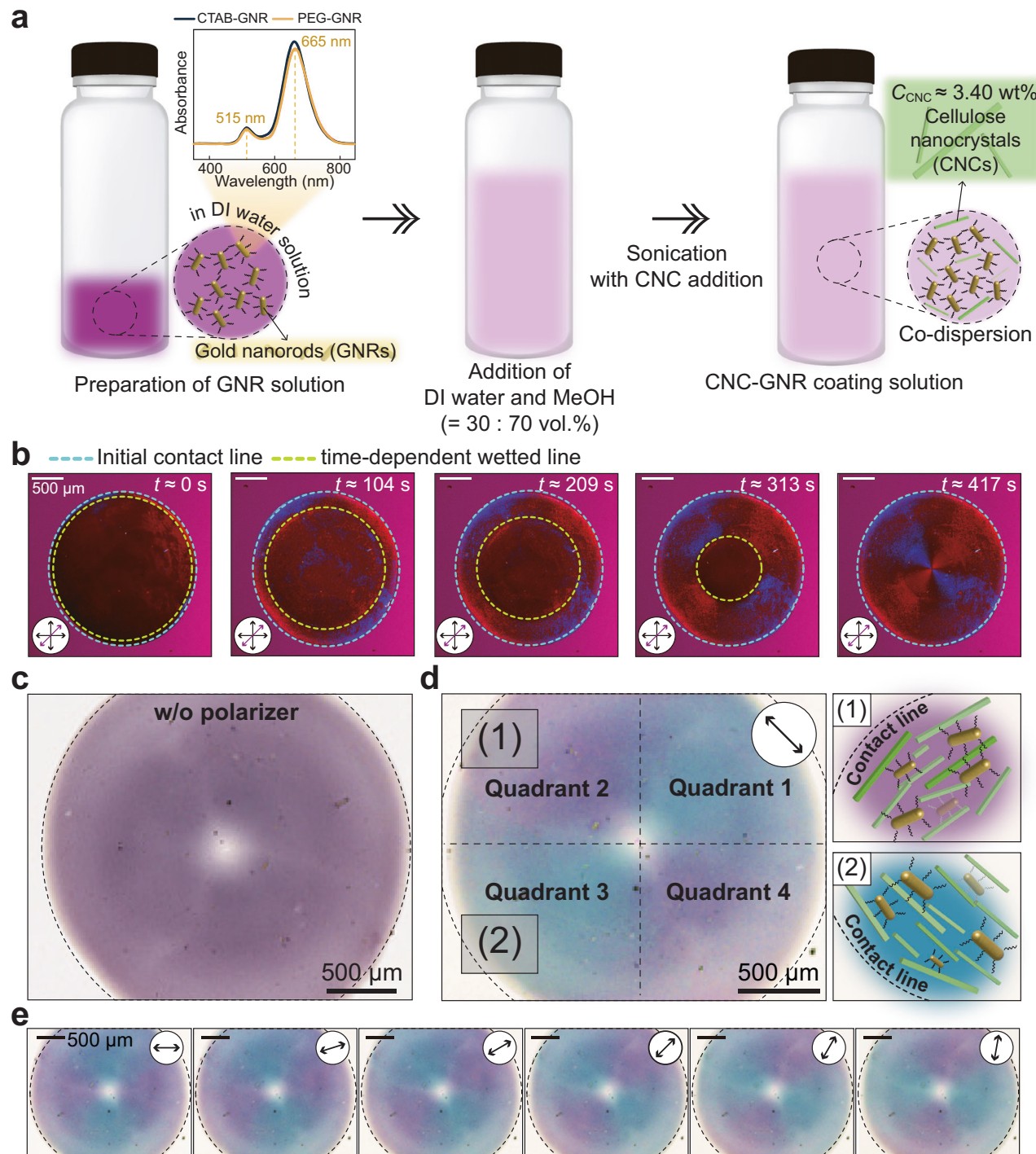

**Fig. 4 | Concentrically aligned plasmonic metasurface of CNC-GNR matrices with tunable optical properties. a** Preparation processes for a CNC-GNR solution with a mass fraction of MeOH : DI water : CNCs : GNRs = 62.57 : 33.76 : 3.39 : 0.28 wt%. The absorption spectrum of the anisotropic GNRs showed the maximum two peaks (wavelengths: 665 nm and 515 nm). Vial images by vectorpocket on Freepik were used. **b** Real-time observation of CNC-GNR droplet evaporation under POM. Here, the bluish and yellowish dashed lines represented the initial contact line ($t \approx 0$ s) and the time-dependent moving contact line, respectively. **c** An optical microscope (OM) image of (**b**) without a polarizer, which exhibited a homogeneous dark violet color. **d** The OM image of (**b**) with a single linear polarizer. Here, a black double arrow indicated the direction of a single polarizer, and two colors were emitted from the film (the quadrant 2 and 4: light violet and the quadrant 1 and 3: blue). In this case, all the GNRs were aligned parallel to CNCs as shown in (1) and (2) illustrations. **e** Sequential changes of plasmonic colors with a polarizer continuously rotating from 0° to 90°. The scale bars in all images are 500 μm.

rotating a single polarizer (see orientation-dependent images, Fig. 4e and Supplementary Fig. 9). This color shift was attributed to the selective absorption of oriented gold nanorods in response to the rotation of polarized light.

We tested the plasmonic photothermal effect of the CNC-GNR metasurface and exposed it to white light irradiation in the range of 350–800 nm, with an irradiance intensity of $500 \pm 50$ mW cm$^{-2}$. Compared to the conditions used in previous studies (around 810 nm

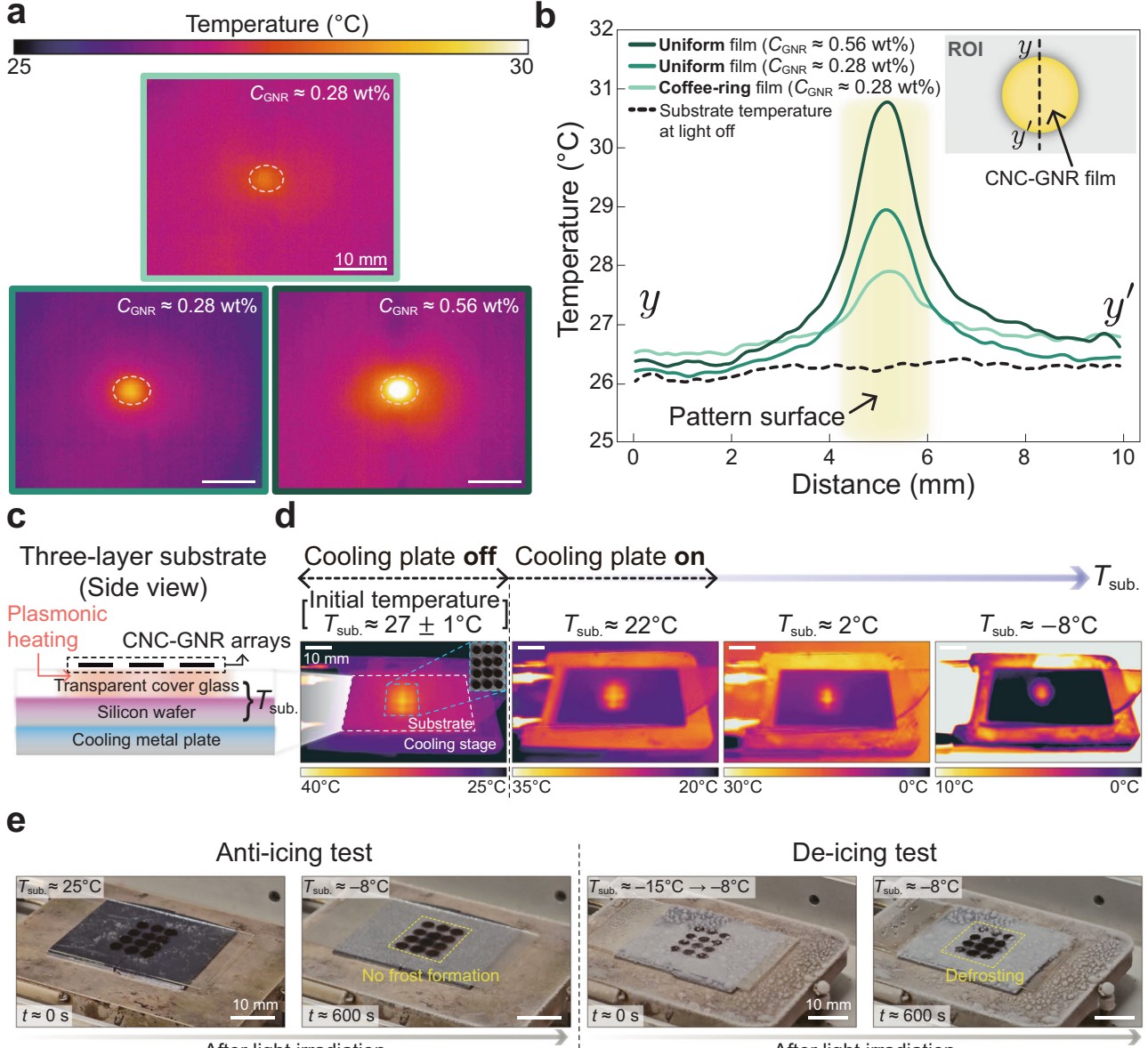

**Fig. 5 | Anti-icing application: enhancement of thermal performance of the plasmonic heater of CNC-GNR metasurfaces. a** Comparison of the plasmonic photothermal effects depending on two parameters; (i) uniformity of the CNC-GNR films [a light greenish box (coffee-ring film) vs. a greenish box (uniform film)] and (ii) GNR concentrations [a greenish box ($C_{GNR} \approx 0.28$ wt%) vs. a dark greenish box ($C_{GNR} \approx 0.56$ wt%)]. All thermal imaging snapshots were captured approximately 270 s after the light source was turned on (see Supplementary Movies 9–11). **b** Measurement results of the temperature line profile along the $y$-$y'$ line. The substrate temperature was about 26 °C without the ambient light (see the black solid line). **c** Illustration of a three-layer stacked solid substrate for controlling the substrate temperature $T_{sub.}$ of the cover glass (top) and silicon wafer (middle) layers. **d** Multi-array (4 × 3) CNC-GNR films ($C_{GNR} \approx 0.56$ wt%) were deposited on the cover glass, as shown in (**c**) and inset of (**d**), and the plasmonic photothermal performance was evaluated using an infrared camera. Here, we controlled the substrate temperature $T_{sub.}$ from 22 °C to –8 °C by adjusting the set temperature of the cooling metal plate (a bottom layer). During the measurement, the CNC-GNR arrays were exposed to a plasma light source. Detailed explanations are given in the Method. **e** Results of anti- and de-icing experiments. While the light was being irradiated, the substrate temperature $T_{sub.}$ remained constant at approximately –8 °C. All white scale bars are 10 mm.

and 2 W cm$^{-2}$)[65,66], the irradiance values were relatively small and the wavelength ranges were broad. The coffee-ring stain issue[67], which results in an inhomogeneous drying pattern due to non-uniform evaporation rates along the droplet interface, has hindered the plasmonic photothermal performance of co-assembled GNRs in various LC matrices[68]. Therefore, to examine the performance of the metasurface in potential applications, we employed a cooling metal plate to regulate the temperature of the glass substrate and created freezing conditions (see Fig. 5c). We attached a 4 × 3 array of CNC-GNR plasmonic patterns to the cooling plate and tested their performance. Even though the temperature of the bottom plate was set

at –8 °C, the multiple CNC-GNR films provided above zero temperature, which was about 5–8 °C. From this test, we concluded that the plasmonic multi-array pattern of the CNC-GNR film could be applied for an anti-icing surface. This conclusion was further supported by actual anti- and de-icing experiments, which were shown in Supplementary Movies 12–13. In these experiments, a waterproof layer was deposited on top of the completely dried CNC-GNR patterns without structural changes to the CNC-GNR structures, as observed in Supplementary Fig. 11. Figure 5e showed that the CNC-GNR films exhibited frost prevention, despite the substrate temperature dropping from 25 °C to –8 °C as shown in Supplementary

Movie 12. This prevention of frost formation can be attributed to the continuous plasmonic photothermal effect of the CNC-GNR films. Furthermore, even when a frost layer had already formed on the CNC-GNR films, it could melt away within 10 min of light irradiation under a sub-zero substrate condition. A detailed demonstration of this process is provided in Supplementary Movie 13.

Our test results also showed that uniform CNC-GNR metasurfaces had better thermal performance as plasmonic heaters than ring-shaped CNC-GNR metasurfaces (compare infrared thermal images of Fig. 5a and Supplementary Movies 9–11). For the coffee-ring film case of Supplementary Fig. 12a, the temperature increased only by less than 2.0 °C with a standard deviation of 0.1 °C (see the light greenish profile of Fig. 5b). This suggests that the coffee-ring structures caused agglomeration of GNR particles near the contact line, which in turn reduced the reactive surface area for LSPR reactions, ultimately degrading their photothermal effect[69] due to two main factors, i.e., increase in light extinction through Mie scattering and a decrease in the surface-to-volume ratio where surface plasmon resonance[70]. Therefore, the uniform single dot films with $C_{GNR} \approx 0.28$ wt% and 0.56 wt% increased up to 3 °C and 5 °C with an experimental error of ± 0.2 °C, respectively (see the greenish/dark greenish profiles of Fig. 5b and real-time temperature data of Supplementary Fig. 10). The results confirmed that plasmonic metasurfaces of homogeneous CNC matrices, derived through the evaporation-induced self-assembly method, can be potentially utilized as plasmonic thermal heaters.

## Discussion

In this work, we presented a straightforward and scalable method for producing a plasmonic metasurface made of CNCs and GNRs. To accomplish this, we utilized two physical phenomena, namely, (i) spontaneous and fast self-dewetting of evaporating drops, and (ii) evaporation-induced self-assembly (EISA), which resulted in excellent drying homogeneity and a high degree of alignment. Although CNCs are natural, renewable, biodegradable, non-toxic, and low-cost nanomaterials, it has had many limitations in serving as a template matrix for GNRs so far. Actually, recent studies have suggested that surface modification[71] of CNCs and precise adjustment of GNR concentration[72] were necessary, and slow evaporation processes[9,71,72] should be implemented to allow sufficient EISA of the CNCs to occur. In contrast, the CNC-GNR metasurface proposed in this study can be easily fabricated using a simple drop evaporation process under room temperature and atmospheric pressure conditions without any chemical additives or external energy sources. To achieve better uniformity, we set the composition (MeOH : DI water : CNCs : GNRs = 62.40 : 33.66 : 3.38 : 0.56 wt% or 68.69 : 29.44 : 1.84 : 0.03 vol.%) of the CNC coating solution and defined the critical coating condition ($U_d/U_c \geq 1$). The metasurface, under these conditions, not only exhibited a uniform film thickness but also displayed tunable optical colors depending on the polarization angles of the incident light beam. We also showed that better film uniformity enhanced the plasmonic photothermal performance of CNC-GNR films compared to ring-shaped CNC-GNR films under visible light. Moreover, when produced in a multi-array format, the CNC-GNR metasurface dramatically increased the temperature rises ($\geq 10$ °C) and remained above zero degrees even when the bottom substrate was below the freezing temperature, demonstrating excellent anti-icing and de-icing performance. It is expected that this anti- and de-icing effect can be further improved by fine-tuning the aspect ratio of anisotropic GNRs[73] and increasing the concentration of GNRs.

Compared to conventional anti-icing systems, the CNC-GNR metasurface has several advantages, such as requiring no substrate modification[42-48] or no need for anti-icing liquid spraying[74], and heat wires[75]. Moreover, in comparison to anti-icing nanomaterials, such as CNTs[76] and PEDOT:PSS[77], GNRs are dispersed well in most solvents for easy utilization and have a relatively high absorbance in visible light range, especially under natural lighting conditions, resulting in better anti-icing efficiency. We believe that the proposed CNC-GNR plasmonic metasurfaces can be printed on a large scale based on the drop-casting method and deposited well on various substrates as long as hydroxyl groups capable of forming hydrogen bonds with the hydroxyl groups on the CNCs' surface are present on the substrate surface. This technique could be applied with a multi-nozzle ink-jet coating system to a photonic platform[78], an energy harvesting technology (e.g., self-actuated devices[79], biomedical applications[80], and energy saving systems[81]), and anti-icing materials[82]. Additionally, it could also be utilized as next-generation metasurfaces for various functional nanomaterials with anisotropic shapes such as CNTs[83], iron oxide ($Fe_3O_4$) nanoparticles[84], and two-dimensional carbides and nitrides (MXene)[85].

## Methods
### Materials
Pristine CNC materials were isolated from microcrystalline cellulose (MC, purchased from Sigma-Aldrich). MC was hydrolyzed at 45 °C in 64 wt% sulfuric acids for 60 min, after which the reaction mixture was quenched using cold DI water. The reaction mixture was then centrifuged and washed with DI water five times to remove residuals and CNC powder was prepared using the conventional freeze-drying method[86]. The length, width, and height of CNCs are 104.05 ± 41.90 nm, 23.50 ± 8.15 nm, and 5.27 ± 1.45 nm, respectively, so the aspect ratio is 2 to 38 (see Supplementary Fig. 4a–d and Supplementary Fig. 13a). MeOH [≥ 99.9 % purity purchased from Sigma-Aldrich (USA)] and water distilled by an ultraviolet (UV) water purification device (Direct-Q3 UV, Sungwoo Genetech, Korea) were used. In a recent study[87], CNCs were found to disperse well in MeOH solution. AFM (Multimode-8, Bruker, USA) measurements confirmed that the swelling effect of MeOH on CNC particles was minor (compare Supplementary Fig. 4a–b and Supplementary Fig. 4c–d). And MeOH has much higher vapor pressure ($P_v$) than that of DI water (vapor pressure: $P_{v,MeOH} \approx 13.02$ kPa[58] $\gg P_{v,water} \approx 2.33$ kPa[59]) compared to other polar organic solvents[88], such as dimethyl sulfoxide (DMSO) with $P_{v,DMSO} \approx 0.08$ kPa, dimethylformamide (DMF) with $P_{v,DMF} \approx 0.52$ kPa, and ethanol (EtOH) with $P_{v,EtOH} \approx 5.95$ kPa. These solvents were also found to be effective in dispersing CNCs.

GNRs were synthesized using a seed-mediated growth method[89] in which hexadecyltrimethylammonium bromide (CTAB) was used as a surfactant. The subsequent ligand exchange procedures were performed as follows. (i) The CTAB-GNR dispersion was centrifuged (13500 x $g$ for 20 min) and redispersed in 1 mL of DI water after eliminating its supernatant liquid. This step was done twice to decrease the CTAB concentration. (ii) The aqueous solution of 6 kg mol$^{-1}$ thiol-functionalized polyethylene glycol[66] [mPEG-SH, Sigma-Aldrich (USA)] (2 mM, 250 μL) was poured into the concentrated GNR dispersion and gently mixed for 30 min at room temperature. Here, mPEG-SH was used as a colloidal stabilizer to form steric repulsion[90]. (iii) After 24 h, the PEG-GNR dispersion was purified by centrifugation (13500 x $g$ for 20 min) to remove excess reagent. The concentration of the resulting PEG-GNR dispersion was approximately 1.3 wt% in a DI water solution. As shown in SEM (SU-8230, Hitachi, Japan) images of Supplementary Fig. 7a and TEM (Tecnai F20, FEI company, USA) images of Supplementary Fig. 13b, the average aspect ratio of GNRs was approximately 3.3 to 5.3 (length: 50–80 nm and diameter: 15 nm).

To evaluate the colloidal stability of a dispersion containing CNCs or GNRs in a mixture of MeOH and water (= 70 : 30 vol.%), we measured the ζ-potential and pH using a particle size analyzer (Zetasizer Nano-ZS, Malvern Panalytical, UK) and a portable pH meter (Orion Star™ A221 pH Portable Meter, Thermo Fisher Scientific Inc., USA). From the measurements, the ζ-potential and pH were −33.9 ± 10.4 mV and 6.84 ± 0.05 for the CNC dispersion and were −5.26 ± 9.00 mV and 6.99 ± 0.03 for the PEG-GNR dispersion (see also Supplementary Fig. 14). Here, the ζ-

potential of PEG-GNR was found to be close to neutral. This indicated that there were no significant electrostatic interactions between the PEG-GNR and CNC. Instead, the PEG-GNR employed a steric hindrance effect to prevent particle aggregation, ensuring the colloidal stability in the CNC-GNR ink, as depicted in Fig. 4a. This was further validated through conductometric titration analysis [HI 98311 (DiST®5), HANNA instruments, Italy]. The analysis showed that the CNC particles are well dispersed in the solvent due to electrostatic repulsion with an approximate value of 275 mmol kg$^{-1}$ [91,92], as shown in Supplementary Fig. 15. This was further supported by Dynamic Light Scattering (DLS) analysis, which confirmed the stable dispersion of CNC or PEG-GNR particles in the mixture solution of MeOH and DI water without any aggregation as shown in Supplementary Fig. 16. Additionally, the $\zeta$-potential data presented in Supplementary Fig. 14b suggest that the ligand exchange from CTAB to PEG proceeded smoothly (see also the UV-Vis spectra in the inset of Fig. 4a). For the $\zeta$-potential and DLS data reliability, each measurement was taken at least three times. Moreover, we measured UV-Vis absorption spectra twice for each of the three cases involving GNRs: (1) CTAB-GNR in DI water, (2) PEG-GNR in DI water, and (3) PEG-GNR with a volume ratio of 70% MeOH and 30% DI water mixture, as shown in Supplementary Fig. 17. With a 1 h gap between measurements, we observed minimal changes in the UV-Vis absorption spectra over time. This suggests that GNRs are stably dispersed throughout the experiments in all employed solvent environments, reaffirming their dispersion stability.

## Sample preparation
For the experiments, the bare glasses were cleaned with acetone, ethanol, and DI water. The first objective of the experiments was to establish the optimal coating conditions for fabricating uniform CNC films. To achieve this, in the optimization experiments shown in Figs. 1 and 3, we controlled the droplet size by using plasma treatment (BD-10ASV, Electro-Technnic Products, INC., USA) on a cover glass substrate, which allowed us to minimize the effect of droplet size by controlling the diameter of the drop to $2R = 3$ mm. The effect of the initial concentration of MeOH and CNCs on the morphology of CNC films was examined under these conditions. To control the droplet size, we used a PDMS shadow mask (silicone elastomer base : curing agent = 10 : 1 wt%) with a circular hole of 3 mm on the glass substrate and performed plasma treatment on the mask. As a result, the uncovered surface had higher hydrophilicity than the rest of the masked surface (i.e., large surface energy contrast). As a result, drops containing CNCs with a diameter of 3 mm were selectively deposited on the more hydrophilic surface (i.e., the uncovered surface). The preparation of the CNC-GNR solution involved the following steps, as depicted in Fig. 4a: (i) synthesis of GNRs in a DI water solution (see the material information in the Method), (ii) addition of MeOH and DI water to the GNR solution at a constant volume ratio of 70 : 30 (= MeOH : DI water), and (iii) addition of approximately 3.40 wt% CNCs to the solution in step (ii) and subsequent dispersion using bath sonication (Branson 1510, Branson Ultrasonics, USA) for 30 min.

## Characterization
We used polarized optical microscopy (POM, Nikon Eclipse Ti2-E microscope, and DS-Ri2 detector, Japan) to investigate the aligned textures of CNCs in both the liquid crystal (LC) and crystalline phases. The colors observed in the POM results shown in Fig. 1 represented the alignment of CNCs. A first-order retardation plate with a wavelength of 530 nm was inserted at a 45° angle between the crossed polarizers. In Fig. 1, the magenta color indicated the isotropic phase of the CNC solution or no deposition on the glass substrate. As shown in the insets of the fourth column in Fig. 1b, the blue and yellow colors represented that the CNCs were oriented parallel or perpendicular to the slow axis (indicated by purple double arrows in the insets), respectively.

To observe internal flow structures inside evaporating droplets, we used micro-particle image velocimetry ($\mu$-PIV) as shown in Fig. 3a. The detailed setup is described in Supplementary Fig. 2. We added fluorescent particles (PS-FluoRed-2.0, microParticles GmbH, Germany) with a diameter of $1.9 \pm 0.1$ $\mu$m and a concentration of 2.5% w/v to a solution containing 3.40 wt% CNCs in a mixture of MeOH and DI water. The particle concentration was 1.0% v/v in the solutions. The negatively charged surface of the fluorescent particles, with a charge of $-15 \pm 5$ mV, repelled the surface of CNC particles with the same negative charge (see Materials in the Methods section), resulting in good dispersion of the particles within the solution for PIV measurements. The fluorescence signal was excited at a wavelength of 532 nm by the Nd:YAG laser (Microvec, China) and emitted at 607 nm. An optical filter with a wavelength cutoff of 540 nm was used to detect the fluorescence signal, which was captured using a camera (Fastcam Mini-AX200, Photron, Japan) installed on a microscope. The droplets were deposited on a glass substrate and allowed to evaporate at a room temperature of 22 °C. Image sequences of the illuminated fluorescent particles were recorded using a camera that captured 50 frames per second. Since the droplet height ($h$) was much smaller than the droplet radius ($R$), i.e., $h/R < 1$, the focal plane was set near the bottom of the droplet, where the depth of field was about 150–200 $\mu$m from the cover glass. The PIVlab tool[93] in MATLAB was used for flow field calculations. To obtain vector fields, we applied iterative 2D cross-correlation of the particle images with multiple interrogation windows of $64 \times 64$ pixels (first) and $32 \times 32$ pixels (second) with 50 % overlaps, where the signal-to-noise ratio (SNR > 3) was satisfied for reliable $\mu$-PIV[94]. Additionally, all fluorescent particles were able to closely follow the flow structures because their Stokes number was much smaller than unity (St = $\rho_p d_p^2 U/18\mu R \ll 1$), calculated based on the particle density $\rho_p \approx \mathcal{O}(10^3 \text{ kg m}^{-3})$, the particle diameter $d_p \approx \mathcal{O}(10^{-6} \text{ m})$, the dynamic viscosity of the solvent $\mu \approx \mathcal{O}(10^{-1} \text{ mPa s})$, the flow speed $U \approx \mathcal{O}(10^{-4} \text{ m s}^{-1})$, and the droplet radius $R \approx \mathcal{O}(10^{-3} \text{ m})$. During PIV measurements, an acrylic box was placed over the droplet to prevent external forces from affecting the internal flows inside the droplet.

## Analysis of spontaneous self-dewetting and coffee-ring flow speed
To estimate speed $U_c$ of evaporatively-driven capillary flows (i.e., coffee-ring flows) adjacent to a contact line, we used the following simplified formula[30] with three main assumptions; (i) a truncated spherical cap-shaped droplet, (ii) low Reynolds number (Re = $\rho U_c R/\mu \approx 10^{-4} \ll 1$), and (iii) thin film approximation ($h/R \approx 10^{-1} < 1$), $U_c(r,z,t) = 3\bar{U}(r,t)$ $(h(r,t)z - z^2/2)/h^2(r,t)$, where the droplet height $h(r,t) = (R^2 - r^2)\theta(t)/2R$, the height-averaged radial speed $\bar{U}(r,t) \approx D^*/[\theta(t)\sqrt{R(R - r)}]$ [here, $D^* = 2\sqrt{2}D_{va}(C_{sat} - C_\infty)/(\pi\rho)$], $R$ is the droplet radius, $\mu$ is the dynamic viscosity, and $\theta(t)$ is the time-dependent contact angle. We set the polar coordinates ($r$, $z$) at the center of the droplet's bottom. Our experiment used a solution consisting of a MeOH and DI water mixture, with MeOH having a vapor pressure almost six times larger than that of DI water at room temperature (because $P_{v,MeOH} \approx 13.02$ kPa[58] and $P_{v,water} \approx 2.33$ kPa[59]). Thus, we could neglect the DI water evaporation until the MeOH component in the droplet has completely evaporated, so the physical properties[95] of the pure MeOH were used in all equations. During evaporation, the changes in the vapor concentration around the droplet might be negligible, even though the vapors were continuously generated from the droplet. This was because the vapor diffusion rate in the ambient air is much larger than the convective mass transfer of the evaporating droplet. Mathematically, this can be expressed using Sherwood number (Sh) $= hR/D_{va} \approx V^{1/3}R/t_e D_{va} \approx \mathcal{O}(10^{-3}–10^{-4}) \ll 1$, where $h$ ($\approx V^{1/3}/t_e$) is the convective mass transfer rate from the liquid droplet to the surrounding gas (m s$^{-1}$), $V$ is the initial droplet volume $\approx \mathcal{O}(10^{-9} \text{ m}^3)$, $t_e$ is the evaporation time $\approx \mathcal{O}(10^2–10^3 \text{ s})$, $R$ is the droplet radius $\approx \mathcal{O}(10^{-3} \text{ m})$, and $D_{va}$ is the vapor diffusion coefficient $\approx \mathcal{O}(10^{-5} \text{ m}^2 \text{ s}^{-1})$. Additionally, the

experimental environment had a relatively low relative humidity (RH) level of approximately $30 \pm 2\%$, which can be considered a dry condition. This level of humidity was not expected to interfere with the diffusion of the evaporated vapors from the droplet in the air. Consequently, the evaporated vapor concentration near the droplet remains unchanged from the initial state, which can be regarded as being in a quasi-steady state[96]. Unless the droplet size is too small to evaporate quickly, the ambient vapor conditions have a minor effect on our case. Hence, the concentration gradient of MeOH vapor was assumed to be $\Delta C = C_{sat} - C_\infty \approx C_{sat} \approx 0.196$ kg m$^{-3}$ [97] (because $C_\infty$ was initially about zero in ambient air) at surrounding temperature $T \approx 22$ °C. The vapor diffusion coefficient of MeOH in the air $D_{va} \approx 1.5 \times 10^{-5}$ m$^2$ s$^{-1}$ [98], and the density of MeOH $\rho \approx 792$ kg m$^{-3}$. With the physical properties above, we calculated the coffee-ring flow speed $U_c$ near the contact line (at $z = 2$ μm and $R - r = 150$ μm) at $t = 0$ s, with the initial contact angle $\theta(0) \approx 32 \pm 2°$ and finally obtained $U_c \approx \mathcal{O}(1$ μm s$^{-1})$ for droplet evaporation of the MeOH and DI water mixture, while the MeOH component was present in the droplet. In the same way, we also calculated the coffee-ring flow speed $U_c$ for a pure DI water condition that occurred in the following two scenarios: (i) when the MeOH component in the MeOH and DI water mixture droplet was totally evaporated, leaving only DI water, and (ii) when a pure DI water droplet was evaporated. In these cases, the following values were used: $D_{va} \approx 2.5 \times 10^{-5}$ m$^2$ s$^{-1}$ [99], $\Delta C = C_{sat} - C_\infty = C_{sat}(1 - RH) \approx 0.016$ kg m$^{-3}$, RH $\approx$ 0.3, and $\rho \approx 997$ kg m$^{-3}$. The calculation result was $U_c \approx \mathcal{O}(0.1$ μm s$^{-1})$.

### Assessment of uniformity in dried patterns
To evaluate the uniformity of the CNC-GNR films, we measured the thickness profiles in the vertical and horizontal directions using a confocal laser scanning microscope (VK-X1050, Keyence, Japan) as illustrated in Supplementary Fig. 5f. The measurements were performed by setting a scanning area of $600 \times 600$ mm$^2$ using the image stitching technique, and the thickness profiles were obtained at a 10× magnification.

### Anti- and de-icing experiment setup
We utilized a temperature control plate (LTS420, Linkam, United Kingdom) to lower the substrate temperature $T_{sub.}$ from 22 °C to −8 °C. The plate was cooled with liquid nitrogen, which circulated through tubes beneath the metal plate. This temperature control plate could operate within a wide temperature range (−195–420 °C) with thermal stability below 0.1 °C. To visualize the spatial temperature distribution of the CNC-GNR films, we employed a thermal imaging camera (FLIR A35, Teledyne FLIR, USA) with a reading accuracy of ± 5%, as presented in Supplementary Fig. 18. The emissivity value of 0.95 was assigned based on the glass material used. The CNC-GNR film was irradiated and stimulated by a plasma light source (HPLS345, Thorlabs, USA). In anti- and de-icing experiments, the waterproof layer was deposited onto the completely dried CNC-GNR array pattern by brushing nail polish oil (Lucid Nail Polish, MISSHA, Korea) and letting it dry at room temperature.

## Data availability
The data that support the findings of this study are available from the corresponding author upon request. Source data are provided with this paper.

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

## Acknowledgements

This work was supported by the Basic Science Research Program through the National Research Foundation (NRF) of Korea funded by the Korean Government (MSIT: 2021R1A2C2007835) and by a grant from the National Research Foundation (NRF) funded by the Korean Government (MSIT: 2018R1A5A1025208).

## Author contributions

J.P. and S.M.P. contributed equally to this work. J.P., S.M.P., D.K.Y., and H.K. conceived the project. J.P. performed POM and deposited CNC matrix films. J.P. and S.M.P. analyzed all POM data with guidance from D.K.Y. and H.K. J.P. conducted particle micro-particle image velocimetry (*μ*-PIV) and the speed analysis. S.M.P. and J.K. synthesized and prepared GNRs dispersed in an aqueous solution, and J.P. fabricated CNC-GNR films. J.K. evaluated the dispersion stability of GNRs and measured the surface charge density of CNCs. S.M.P. conducted Atomic Force Microscopy (AFM) and Scanning Electron Microscope (SEM) measurements on CNCs, GNRs, and CNC-GNR film samples. J.P. and S.M.P. conducted plasmonic optical and photothermal tests. Y.-J.Y. and J.-H.K. measured the film thickness profile. J.P. wrote the manuscript under H.K.'s guidance.

## Competing interests

The authors declare no competing interests.
