## [Peer Review File · Nature Communications]

Plasmonic Metasurfaces of Cellulose Nanocrystal
Matrices with Quadrants of Aligned Gold Nanorods
for Photothermal Anti-IcingREVIEWER COMMENTS

Reviewer #1 (Remarks to the Author):

This manuscript has report assembly of cellulose nanocrystal (CNCs) from a mixture solution of MeOH and DI water to form concentric alignment along the contact line. AuNRs were also co-assembled to show the anti-icing effect. The authors explained that the assembly was facilitated by the fast dewetting of MeOH. However, there was not solid experimental data to supporting this. And the detailed characterization of the CNCs alignment. Furthermore, the observed phenomenon is not new, which has been reported in previously literatures, such as, J. Phys. Chem. B 2006, 110, 7090; Small Methods 2021, 5, 2100690; ACS Appl. Mater. Interfaces 2019, 11, 1538–1545 ; And the theoretical analysis and explanations were not strongly supported by the data, and not so convincible. In this stage, this work is not suitable to be published. Therefore, it is not suggested to publish in Nature Communications.

- 1.Solid characterization of CNCs assembly is recommend, as well as that of the AuNRs.
- 2.The mechanism of CNCs assembly along the contact line, as well as the drive force in the system to order the CNCs, are needed to be clearly explained.
- 3.The flow movement of during the EISA of CNCs assembly are suggested to characterized.
- 4.When the MeOH was evaporated, how would the surface tension change at the contact line, as well as the top surface of the sessile droplet, in comparison to the center of the droplet.
- 5.The logical connection is not smooth as turning the work from pure CNCs assembly to CNCs+AuNR assembly, as well as the anti-icing part. What is the purpose of the CNC+AuNRs assembly?
- 6.The authors declared that the find a facile approach to assembly CNCs. However, EISA is well-developed approach for assembly.
- 7.The introduction is needed to be improved significantly. There was lack of review about the assembly approach of CNCs, the assembly manner of CNCs and corresponding optical properties.
- 8.The AuNRs did not have a uniform rod-like morphology, and its UV-vis was lack after changing the ligand to PEG-SH. The stability of the AuNRs within the MeOH+DI mixture is also need to concerned.

Reviewer #2 (Remarks to the Author):

In this manuscript, the authors have reported a uniform broadband plasmonic metasurface by drop-casting method. The CNC-GNR metasurface was achieved by self-dewetting of binary solution consisting of methanol and deionized water. The self-dewetting force driven by the fast evaporation of methanol became predominant compared to the self-pinning force caused by anchoring of CNCs, caused continuous crystallization of CNCs and avoid stick-slip phenomenon. And the mechanism of uniform patterning and the critical condition for forming homogeneous CNC matrix were comprehended, the velocity ratio of self-dewetting and coffee-ring flow determined whether the final dried films uniform or not. The prepared thermal plasmonic metasurfaces by introducing GNRs in CNC matrix was preliminary applied in anti-icing.

In general, this work provides a new route to produce thin films without coffee ring by binary solution method, but the experimental data and the discussion for application of CNC-GNR in anti-icing are insufficient. Major revision needed as noted.

Comment 1: The CNC-GNR plasmonic metasurface was prepared by the physical adoption of CNCs and GNRs without cross-linking, what about the water resistance of the film, because the contact with water is inevitable in anti-icing.

Comment 2: The authors claimed that “we presented a straightforward and scalable method for producing a broadband plasmonic metasurface...”, but the samples prepared in manuscript by plasma etching and CNC-GNR solution drop-casting was millimeter-scale, which invites questions about the scalable preparation. And drop-casting method is working well on flat surface, how about the universality of this method, such as curved or irregular surfaces?

Comment 3: Characterization of the surface morphologies of CNC matrix and CNC-GNR film are lacked, which can show the parallel alignments more intuitively.

Comment 4: For the application in anti-icing, the photothermal effect on CNC-GNR metasurface was measured under white light with irradiance intensity of $500 \pm 50 \text{ mW}\cdot\text{cm}^{-2}$, which was 5 times of solar light ($100 \text{ mW}\cdot\text{cm}^{-2}$). The photothermal performance on CNC-GNR film is weak compared with other anisotropic nanomaterials like CNTs. In this part, only surface temperature was measured, the anti-freezing performance of water, or the photothermal deicing tests are missing. The CNC-GNR shows no superior in anti-icing.

Reviewer #3 (Remarks to the Author):

The manuscript discusses materials and methods implementing cellulose nanocrystals and gold nanorods for formation of plasmonic heating surfaces. The authors report some potentially interesting thermal response upon addition of gold nanorods into the suspension. However, most of the manuscript is focused on the CNC droplet evaporation dynamics and lacks significant detail pertaining to the colloidal state of the materials and the final structure of the deposited CNC/GNR materials.

Major comments:

The discussion and data on selecting a CNC composition is confusing. The limited discussion does not clearly explain the rationale – for instance, 3 videos of 2.8wt% CNC are included in the SI, but only one with 3.4% CNC (without other materials). However, the authors suggest 3.4% is good because of more uniformity, with little discussion in how they are defining uniformity (see later comment). Further, there are significant considerations for concentration (particularly initial concentration prior to evaporation) that strongly influence the results, as highlighted in the next comment about the starting state of the CNCs. A brief mention of 3 concentrations that are in a relatively similar range of CNC behavior were “evaluated” (Figure 1e) but with very little discussion. (and again another reference to “optimal”, please see comment below in minor comments).

CNC preparation and characterization – the authors do not provide any characterization information about the nanocellulose other than size by TEM. A protocol exists for well-defined characterization using AFM of dilute CNCs on spin-cast mica wafers, how did the authors determine “width”? Characterization information that is critical for reproducing results, because of the sensitivity of the CNC colloidal state to these parameters, includes quality of redispersion following freeze-drying, surface charge density in suspension, pH, etc. – these factors and others are critical for colloidal stability and will strongly impact the ability for someone to reproduce the results. Further, the authors did not dialyze their CNCs following production, “simple” washing with water is not typically sufficient to purify the suspensions following hydrolysis (it is often one step in the process). Therefore, there is some concern about the state of the starting material that should be thoroughly addressed.

p. 10 – report the actual differences in methanol volatility, particularly in context of the solution with water and any documented literature on the actual vapor pressure. The authors should at minimum

comment on the relative order of magnitude impact of making this assumption that water does not evaporate and is not very volatile compared to methanol, and further this reviewer would expect that droplet size and ambient conditions further play into this assumption.

A significant effort was invested to understand the droplet evaporation process and analysis of the CNC alignment process – and given the extensive references provided by the author, one can deduce that significant effort has been invested in the community to understand the evaporation induced assembly of CNCs. Then the authors added gold nanoparticles without discussing the influence of the new nanomaterials on the colloidal stability, phase behavior, etc. all of which will significantly influence the droplet dynamics – and at this point, the authors suggest that they were still able to maintain quadrant uniformity in dried droplets, but neglect to characterize the impact on the starting CNC suspension characteristics (which ties to above comment about the starting materials).

The authors make a major leap from a limited set of measurements to suggesting widespread and broad application of the materials. The term anti-icing, in itself, has many connotations in the community and should be carefully implemented. Without significant additional analysis, this reviewer feels that the authors should reduce the broad overarching application potential with such a limited set of data. To this effect, the title specifically indicates "for anti-icing" and should be adjusted.

The authors conclude that they have demonstrated a high degree of drying uniformity and alignment throughout the droplet (2nd sentence in conclusion). While POM gives a sense of the bulk uniformity, it cannot effectively probe local uniformity. Therefore the authors should clearly identify how they are defining a "uniform" structure that is "highly aligned". Further, additional analyses such as AFM could strongly support their claims. The results do not provide a clear picture of uniformity in distribution or dispersion of the dried droplets outside of a bulk visual measurement, which leaves significant room for qualitative interpretation. Further, Figure 2 shows uniform "quadrants" but even suggests non-uniform structure in the radial direction (per the authors graphical representation).

Minor comments:

On p. 3 the authors mention "Once the contact line dewetted, the CNCs were continuously crystallized along the moving contact line without stick-slip motions". Please provide some clarification – the CNCs are already crystals and are not going to "crystallize", so please clarify what is meant.

p.9 – please confirm that the CNCs were not "dissolved" but rather "dispersed" (last sentence, first paragraph). Sample "preparation" is also misspelled in the subheading.

p.9 methods – the authors mention "optimal coating conditions" – this reviewer does not see an optimization study (e.g. any signs of a design of experiment and corresponding modeling, statistical analysis and subsequent experimentation) performed in this manuscript.

CNC concentrations are reported in wt. % yet solution concentrations (methanol and water) are reported in vol %. The suspension and colloids community typically prefers to see vol % for comparison to related theory and expt.

Reviewer #4 (Remarks to the Author):

This article has potentially interesting results but relies too much on pretty pictures and inferences rather than hard data. Also, links with the CNC literature, in terms of optical materials, self-assembly, and interactions are not properly included and used. Therefore, I have a long list of things to be addressed before it could be published.

- Figure 1 left stretches the truth a bit. Where is the lignin, hemicellulose, pectins,...?
- Starting materials should be more fully characterized. What is the charge density of the CNCs,

counter ions, dimensional histograms for both GNRs and CNCs?

- Is there any structuring of the GNR-CNC suspension? SANS work on CNCs and GNRs has previously shown alignment of both GNRs and CNCs even in the "isotropic" phase.
- What is the residual amount of CTAB in the mPEG-GNR suspension? This is critical since it will interact with the negatively charged CNCs.
- Work by Frka-Petesic and Vignolini has been completely ignored in the references. This should be corrected.
- Page 1: "Based on Onsager's Theory..." This is an incorrect statement as the particles never heard of this theory. The theory is descriptive of physical behavior, the physical behavior does not occur because of the theory...
- Page 2: Previous work with solutal Marangoni flow is dismissed because it "generates uncontrollable flow structures due to its inherent instabilities at a liquid-gas interface". However, previous work did show it can be used and the work presented in this paper is also a form of solutal Marangoni flow. The concentration of MeOH is changed, not by diffusion of MeOH into the water, but by faster evaporation out of an aqueous solution. So this work suffers from the same issues.
- Page 2: Interaction with different agents is also dismissed "due to the complication of interaction between CNCs and agents." However, Lombardo and Thielemans have reviewed and elucidated a clear entropy-enthalpy compensation mechanism and described gelation of CNCs with ions thermodynamically. I would think this is rather well understood.
- Page 4 line 1: How is the diffusion coefficient of a MeOH-water mixture defined? Either it is MeOH in water or water in MeOH...but it can't be the way it is described.
- Page 4 line 3: "along the liquid-gas interface between the MeOH and DI water components". If MeOH is dissolved in water, there should not be a liquid-gas interface between both...This doesn't make any sense?
- What is the atmosphere where the droplets are dried? What is the RH? This will define the actual water-MeOH ratio in the atmosphere, rather than assuming it is pure MeOH.
- What is the effect of the substrate? This will be critical since the CNCs may adhere to the substrate before being deposited.
- What is the uniformity of the film across its thickness? The films are more than 5 microns thick so exist of many CNC layers. How important is uniformity?
- Figure 2(iv) is a nice representation but there is only circumstantial evidence of this pattern. Electron microscopy should be carried out to prove this is indeed what the structure is to see how robust the structuring is relative to the POM response.
- It is not clear that the (only slightly) improvement in plasmonic heating is due to the alignment of the GNRs by the CNCs. Slightly better performance than systems deposited with a coffee ring effect might be due to the inhomogeneous thickness of those films, or by preferential deposition of GNRs in one region of the film. This should be verified. Also, what is the heating effect for a coffee ring deposit containing 0.5 wt% GNRs? It is reported for the homogeneous deposit and could give a false idea of better performance to the unconcentrated reader.
- What is the importance of the CNC and GNR concentration? 3.4 wt% CNCs already results in aligned domains in the "isotropic" phase.
- Profilometry data for all data points in Figure 3 should be provided and the extent of the coffee ring effect in the red domain quantified.
- The contact angle as function of drying time should be provided for reported samples.
- A control experiment where mPEG-GNRs by themselves are deposited under the described conditions should also be carried out to see what the plasmonic heating effect would be.
- Figure 3 is not very well explained in the text. The x-axis of the graph is a dimensionless time during droplet drying. How is it then possible that a deposit goes from a coffee ring deposit to a non-coffee ring deposit? The coffee ring cannot all of a sudden disappear. Ref 18 already showed that centrally, one can get a homogeneous deposit inside the coffee ring under some condition but that is not the same as what is represented in this figure.

Reviewer #5 (Remarks to the Author):

In this work, authors fabricated CNCs/GNRs co-assembled films using a water-based binary mixture solution containing MeOH followed by a facile drop-casting method. The fast and selective evaporation of MeOH near the contact line causes the spontaneous and fast self-dewetting of the solvent, which helps to create a uniformly aligned CNC pattern without coffee-ring. The metasurfaces' surface temperature exhibited above zero ($\approx 5-8$ oC), although the bottom substrate had -8 oC, and this film can be used as a plasmonic photothermal film for anti-icing applications. Generally, this work given a new insight to construct functional surface, but there are many issues should be explained clearly, such as, the necessary structural characterization, the advantage of photo-thermal deicing of CNCs/GNRs film that contrasted with the traditional photo-thermal surface, etc... Thus, it cannot accept the manuscript for publication at this version.

1. Authors should add the optical photographs of CNCs-GNR films. Moreover, the cross-sectional SEM images of films should also be added to illustrate the well-oriented chiral nematic structure. The well-oriented chiral nematic structure may exhibit long-range order layered structure without the fuse defect between adjacent tactoids.
2. What is CNCs-GNRs films' vis-NIR transmittance or reflectance spectrum, and thickness?
3. How do the GNRs arrange in CNCs-GNRs composite films? Among CNCs rods or between the adjacent CNCs layers? Whether the addition of different amounts of GNRs influence the film's helical pitch? The direct evidences, for example, the TEM of CNCs-GNRs films, should be provided.
4. It has been reported that methanol shows high swelling values to cellulose (60.5% to α -cellulose). Did the addition of methanol (>60 %) break CNCs systems' hydrogen bonding and impede their self-assembled structure?
5. The GNRs used in this work are positive charge or negative charge? Different charges of GNRs may affect CNCs' self-assembly structure due to CNCs rods' inherent negative charge. Moreover, in the manuscript, the plasmonic metasurfaces of CNCs-GNRs films are the key point. I think the corresponding characterizations of GNRs and CNCs-GNRs films should be added such as the plasmonic-enhanced fluorescence emission and circular dichroism (CD).
6. Through the TEM images (Figure S4b) and the averaged aspect ratio (approximately 3.3 to 5.3) described in the Methods part in the main text. I think the range of GNRs' aspect ratios is wide and the GNRs solution is inhomogeneous. Hence, authors should further add films photographs to illustrate the improvement of CNCs alignment.
7. Did the co-assembly of CNCs and GNRs influence GNRs inherent two localized surface plasmon resonance (LSPR) peaks (665 and 515 nm)?
8. Authors described that the co-assembled GNRs in the CNC matrix as a plasmonic optical metasurface, and the different concentrations of GNRs influence the photothermal performance of films. Whether the different aspect ratios of GNRs or thickness of CNCs films influence it's photothermal performance? I think authors should discuss these questions.
9. How about the water resistance of CNC film, I am afraid a worse water resistance for this film because there was any chemical cross-linked structures in the film system, and the two compositions, GNRs and CNCs have good water dispersibility.
10. The optical photographs of anti-icing process should also be added in the main text, rather than the surface temperature.
11. There have been various reported works on the white light-triggered photo-thermal anti-icing. What is this plasmonic optical metasurface's advantage? The high heating rate? The authors should be discussed in the text. The GNRs are too expensive and the resulting CNCs-GNRs may bear some shortcomings such as the strong hygroscopic, which may limit films cyclic utilization.

Reviewer's comments to the Authors:

We thank the reviewer for the comments in regard to our manuscript. Please find below a list of our responses and modifications regarding each of reviewers' concerns. The added texts are marked by blue color and the removed texts are marked by ~~red color~~.

Reviewer #1

We thank the Reviewer for specific comments that must help us to improve the manuscript further. We reply to the Reviewer's questions and comments below, point by point, and have edited the manuscript in several places to reflect these remarks.

1	[Comment to the authors] The observed phenomenon is not new, which has been reported in previously literatures, such as, J. Phys. Chem. B 2006, 110, 7090; Small Methods 2021, 5, 2100690; ACS Appl. Mater. Interfaces 2018, 11, 1538–1545 ; And the theoretical analysis and explanations were not strongly supported by the data, and not so convincing.	[Our response] Thank you for the referee's comment. We partially understood the referee's concerns. As the referee mentioned, there were several studies that tried to suppress the non-uniformity of the CNC-dried films using the evaporation of CNC dispersion ink. Particularly, one commonly used method was the solutal-⁽¹⁾ and thermal-⁽²⁾ Marangoni effect, which created spontaneous mixing flow structures during evaporation. In fact, this effect alone is incomplete to perfectly eliminate the coffee-ring pattern or irregular pattern, as it has a short duration time⁽³⁾ and inherent instability⁽⁴⁾. In this reason, previous studies have often required additional steps, such as creating a volatile vapor environment⁽⁵⁾, heating the coating substrate^(6,7,8), or using special additives that cause gelation⁽⁹⁾. Even some studies required pre-patterned substrates^(7,8) to uniformly deposit it or to control its self-assembly structures. However, the current study introduced a novel method and result to achieve the very uniform and homogeneous CNC-dried film with gold nanorods. In this paper, we proposed a one-step approach for fabricating a homogeneous CNC matrix film that combines the solutal-Marangoni flows with the spontaneous and rapid self-dewetting phenomenon. This drying and coating method simplifies the fabrication and improve the final deposit pattern. Interestingly, this approach requires no special prerequisites, such as specific environmental conditions, additives, external energy sources, and substrate fabrication processes. Additionally, it takes a relatively short time to create the pattern (only about 450 s for 2 - 3 mm patterns) due to fast evaporation. Normally, the drying speed of a water droplet is about 20 - 25 mins for the same volume. To achieve uniformity, we revealed that it is crucial to balance the self-dewetting inward motion with the outward coffee-ring flows, after the solutal-Marangoni disappeared. This coating principle was explained in more detail in our response to Comment No. 4. Through multiple experiments, we optimized the CNC ink composition and coating conditions to achieve uniformity. Using this uniform CNC template, we successfully fabricated a well-aligned, homogenous GNR metasurface in a quadrant shape (see Fig. 4 in the manuscript). We
---	---	--

		demonstrated that the resulting coffee-ring-less CNC-GNR composite film exhibited tunable optical surfaces depending on the angle of the polarized incident light and better photothermal performance than the coffee-ring-shaped film. This homogeneous CNC-GNR film prevents the aggregation of GNR particles and the reduction of the reactive surface area for localized surface plasmon resonance (LSPR). Furthermore, by depositing it in a multi-array format, we demonstrated the potential of our CNC-GNR multi-array films for anti-icing applications. In the revision process, we substantially conducted several additional experiments, including micro-particle image velocity (μ-PIV), Atomic Force Microscopy (AFM) and Scanning Electron Microscopy (SEM), to provide additional evidence supporting the mechanisms behind our coating strategy using a CNC matrix with GNRs. Some corrections were made in the revised version of the manuscript (μ-PIV: comment No. 4, and AFM/SEM: Comment No. 2). [References] (1) Kim, H., Boulogne, F., Um, E., Jacobi, I., Button, E., & Stone, H. A. (2016). Controlled uniform coating from the interplay of Marangoni flows and surface-adsorbed macromolecules. Physical Review Letters, 116(12), 124501. (2) Hu, H., & Larson, R. G. (2006). Marangoni effect reverses coffee-ring depositions. The Journal of Physical Chemistry B, 110(14), 7090-7094. (3) Christy, J. R., Hamamoto, Y., & Sefiane, K. (2011). Flow transition within an evaporating binary mixture sessile drop. Physical Review Letters, 106(20), 205701. (4) Kim, H. (2022). Multiple Marangoni flows in a binary mixture sessile droplet. Physics of Fluids, 34(12), 122102. (5) Gençer, A., Schütz, C., & Thielemans, W. (2017). Influence of the particle concentration and marangoni flow on the formation of cellulose nanocrystal films. Langmuir, 33(1), 228-234. (6) Shao, R., Meng, X., Shi, Z., Zhong, J., Cai, Z., Hu, J., ... & Ye, C. (2021). Marangoni Flow Manipulated Concentric Assembly of Cellulose Nanocrystals. Small Methods, 5(11), 2100690. (7) Wang, H., Shao, R., Meng, X., He, Y., Shi, Z., Guo, Z., & Ye, C. (2022). Programmable Birefringent Patterns from Modulating the Localized Orientation of Cellulose Nanocrystals. ACS Applied Materials & Interfaces, 14(31), 36277-36286. (8) Mashkour, M., Kimura, T., Mashkour, M., Kimura, F., & Tajvidi, M. (2018). Printing birefringent figures by surface tension-directed self-assembly of a cellulose nanocrystal/polymer ink components. ACS Applied Materials & Interfaces, 11(1), 1538-1545. (9) Gençer, A., Van Rie, J., Lombardo, S., Kang, K., & Thielemans, W. (2018). Effect of gelation on the colloidal deposition of cellulose nanocrystal films. Biomacromolecules, 19(8), 3233-3243.
2	[Comment to the authors] Solid characterization of CNCs assembly is recommend, as well as that of the AuNRs.	[Our response] To reflect the reviewer's comment and suggestion, we investigated the surface morphology of a CNC film using AFM. To facilitate our study, we divided the film domain into three sections, including i) the inside of the droplet, ii) intermediate, and iii) outermost droplet area. We measured $3 \times 3 \mu\text{m}$ square domain at each location on the line at 40 degrees. Three AFM height images were acquired, and we analyzed the angular distribution of CNC using MATLAB software, as described in the article by Persson, N. E. et al. Chem. Mater. 29, 3-14, (2017). The resulting distribution maps

demonstrated a dumbbell-like structure with an average angle of approximately 140° . This implies that the uniaxial alignment of CNC is parallel to the droplet contact line, regardless of the domain location.

Moreover, we characterized the alignment of gold nanorods in the CNC matrix using SEM. Similar to AFM, we captured SEM images at the same three domains (i, ii, and iii) using backscattered electron (BSE) mode. The BSE mode enabled the gold nanorods in the CNC matrix to be highlighted brightly in the SEM image. Our results indicate that the surface morphology of the CNC-GNR film shows uniaxial alignment regardless of domains. Furthermore, the BSE images in the same area demonstrated that the gold nanorods followed the alignment of the CNC matrix.

[Changed to the manuscript]

On p. 4, "... As a result, we obtained a uniformly crystallized CNC matrix in a quadrant orientation, without any changes to their LC structures (see the blue and yellow textures of Fig. 1e and the AFM images of Fig. S4c)."

On p. 5, "...[see the illustration (i) and (ii) of Fig. 4d]. Additionally, SEM images of the CNC-GNR metasurface were provided in Fig. S7b, which offers a more detailed visualization. Moreover, we also demonstrated continuous plasmonic color changes of the CNC-GNR metasurface by rotating a single polarizer (see orientation-dependent images, Fig. 4e and ~~Fig-S2~~Fig. S8)."

In Supporting Information,

Figure S3. Characterization of CNC particles and CNC films by Atomic Force Microscope (AFM). **a**, Representative AFM images. **b**, Histogram of width and length distribution of CNC particles. **c**, AFM images (top) and rotation distribution (bottom) of the CNC film at ⁽ⁱ⁾ the inner, ⁽ⁱⁱ⁾ the intermediate, and ⁽ⁱⁱⁱ⁾ the outer ring region in the direction of 40° . The CNC film was created by evaporating a droplet containing MeOH : DI water : CNCs = 62.75 : 33.85 : 3.40 wt%. The order parameter S indicates the degree of alignment of the CNCs.

		 Figure S7. Characterization of CNC-GNR films by Scanning Electron Microscope-Backscattered Electron (SEM-BSE). a, Representative SEM images of PEG-GNR particles. b, SEM (top) and BSE (bottom) images of the CNC-GNR film at ⁽ⁱ⁾ the inner, ⁽ⁱⁱ⁾ the intermediate, and ⁽ⁱⁱⁱ⁾ the outer ring region in the direction of 40°. For the SEM-BSE measurements, the CNC-GNR film was deposited onto a silicon wafer substrate by evaporating the droplet composed of MeOH : DI water : CNCs : GNRs = 63.11 : 33.46 : 3.40 : 0.04 wt%.
3	[Comment to the authors] The mechanism of CNCs assembly along the contact line, as well as the drive force in the system to order the CNCs, are needed to be clearly explained.	[Our response] Thank you for the referee’s comment. As the referee knows, the self-assembly of CNCs is determined by a combination of multiple factors, including electrostatic interactions, hydrogen bonding, and steric effects. Among them, the repulsive effects are related to the electrostatic effects that will be reduced or neutralized by using a methanol-water mixture. The other two effects will contribute for the close packing and alignment. However, the primary mechanism behind the assembly process of CNCs along the contact line relies on the elastic behavior of the liquid crystal phase within an evaporating droplet. The CNC colloids were transported to the contact line of the drop by an evaporatively-driven outward flow. The CNC concentration at the droplet rim continuously increased, promoting liquid crystal self-assembly. As the droplet evaporated, the contact angle became smaller and the meniscus adopted a wedge-shaped configuration, leading to splay deformation in the CNCs through the radially outward capillary flows. In this process, there was a major competition between the dilative stress along the r-direction near the retreating contact line and the elasticity of the CNCs. Unfortunately, perfectly modeling this self-assembling mechanism is extremely challenging due to the complexity of the system itself. However, it is widely recognized that anisotropic particles, such as DNA ⁽¹⁾, viruses ⁽²⁾, and carbon nanotubes (CNTs) ⁽³⁾, dispersed in droplets possess higher elastic free energy associated with splay deformation in comparison to bend deformation. Thus, it can be inferred that the CNCs exhibit similar elastic behavior to the anisotropic materials⁽¹⁻³⁾. The CNCs’ structure was adjusted to achieve an energetically favorable and stable state by minimizing changes in its elastic free energy in response to the distortions. Consequently, the CNCs withstand the stress induced by splay

		deformation and exhibit an annular alignment (ring-like patterns) as the process of evaporation continues. We added the detailed mechanisms of CNCs' self-assembly in the revised manuscript. [References] (1) Cha, Y. J., Park, S. M., You, R., Kim, H., & Yoon, D. K. (2019). Microstructure arrays of DNA using topographic control. Nature Communications, 10(1), 2512. (2) Park, S. M., Kim, W. G., Kim, J., Choi, E. J., Kim, H., Oh, J. W., & Yoon, D. K. (2021). Fabrication of chiral M13 bacteriophage film by evaporation-induced self-assembly. Small, 17(26), 2008097. (3) Zhang, S., Li, Q., Kinloch, I. A., & Windle, A. H. (2010). Ordering in a droplet of an aqueous suspension of single-wall carbon nanotubes on a solid substrate. Langmuir, 26(3), 2107-2112. [Changed to the manuscript] On p. 3, "Tangential alignment with an annular pattern occurs near the triple-phase contact line, which is the boundary between the droplet, the surrounding air, and the substrate. As the droplet evaporated, the contact angle became smaller and the meniscus adopted a wedge-shaped configuration, leading to splay deformation in the CNCs through the radially outward capillary flows. In this process, there was a major competition between the dilative stress along the r-direction near the retreating contact line and the elasticity of the CNCs. It is widely recognized that anisotropic particles, such as DNA¹⁵, viruses¹⁶, and carbon nanotubes (CNTs)⁵⁵, dispersed in droplets possess higher elastic free energy associated with splay deformation in comparison to bend deformation. Thus, it can be inferred that the CNCs exhibit similar elastic behavior to the anisotropic materials^(15, 16, 55). The CNCs' structure was adjusted to achieve an energetically favorable and stable state by minimizing the changes in its elastic free energy in response to the distortions. Consequently, the CNCs withstand the stress induced by splay deformation and exhibit an annular alignment (ring-like patterns) as the process of evaporation continues. Here the fast and selective evaporation of MeOH near the contact line causes the spontaneous and fast self-dewetting of the solvent, which helps to create a uniformly aligned CNC pattern.Based on this alignment mechanism, the spontaneous and fast self-dewetting driven by selective evaporation MeOH near the contact line helps to create a uniformly aligned CNC pattern."
4	[Comment to the authors] The flow movement of during the EISA of CNCs assembly are suggested to characterized.	[Our response] To reflect the referee's comments and to resolve the concerns, we additionally conducted micro-particle image velocimetry (μ-PIV) experiment and measured the time-dependent contact angle for MeOH and DI water mixture droplet (see Fig. 3a and Fig. S1). Based on the updated data, we tried to explain the flow dynamics involved in the formation of a homogeneous CNC film.

As we already described in the submitted manuscript, to achieve the homogeneous CNC matrix template, we mainly utilized two evaporation mechanisms; (1) Marangoni-driven flows by selective evaporation and (2) dynamical flows by self-dewetting and coffee-ring phenomenon. In the early stages of evaporation [i.e., (i) regime in Fig. 2], we generated solutal-Marangoni flows by selectively evaporating the MeOH and DI water components to mix the suspended CNC particles well. After the solutal-Marangoni flow diminished in the mid-late stages of evaporation, we used high concentration of MeOH in the CNC solution to induce spontaneous and rapid self-dewetting. At this time, the self-dewetting motion was balanced by the coffee-ring flows in the opposite direction [i.e., (iii) regime in Fig. 2]. In this situation, the self-dewetting speed (U_d) and the coffee-ring flow speed (U_c) were comparable to each other ($|U_d| \gtrsim |U_c|$), which prevent the self-pinning of the contact line while the CNC's EISA occurred continuously along the moving contact line (For a more detailed explanation of the CNCs' EISA process, see our response to comment 3). As a result, we suppressed the accumulation of CNC particles near an initial contact line and obtained a homogeneous annular CNC matrix.

Herein, we estimated the coffee-ring flow speed (U_c) using a theoretical model ⁽¹⁾ (see the Method Section). From the calculation, we found $U_c \approx 1 \mu\text{m/s}$ for MeOH and DI water mixture and $U_c \approx 0.1 \mu\text{m/s}$ for pure DI water. In the case of the binary mixture case, U_c was differently defined for the early evaporation ($t/t_e < 0.25$) and for mid-late evaporation ($t/t_e \geq 0.25$), respectively. This was because MeOH components rapidly evaporated first and then almost only the DI water component remained. Therefore, we redefined that U_c is on order of $1 \mu\text{m/s}$ at $t/t_e < 0.25$ and U_c is on order of $0.1 \mu\text{m/s}$ at $t/t_e \geq 0.25$ for the evaporation of binary mixture droplet. PIV results in Fig. 3a also showed that, after the solutal-Marangoni flows almost disappeared, the coffee-ring flow was generated, which showed an average flow speed U_c of approximately $1 - 2 \mu\text{m/s}$ (A and B regimes in Fig. 3a). Afterward, MeOH components totally evaporated, and then the remaining pure DI water evaporation caused the coffee-ring flows, which exhibited an average flow speed U_c of approximately $0.4 - 1 \mu\text{m/s}$ (C and D regimes in Fig. 3a), consistent with our previous assumption. The U_c did not vary significantly until the droplet totally evaporated because changes in the contact angle was very small while the droplet freely receded (see Fig. S1a) ⁽²⁾. For the dewetting speed (U_d), we measured it by tracking the crystal growth of CNCs during evaporation in sequential POM image (see Fig. 3b).

[References]

- (1) Marin, A. G., Gelderblom, H., Lohse, D., & Snoeijer, J. H. (2011). Order-to-disorder transition in ring-shaped colloidal stains. *Physical Review Letters*, 107(8), 085502.
- (2) Lee, S. Y., Kim, H., Kim, S. H., & Stone, H. A. (2018). Uniform coating of self-assembled

noniridescent colloidal nanostructures using the Marangoni effect and polymers. *Physical Review Applied*, 10(5), 054003.

[Changed to the manuscript]

On p. 2, “Here, the magnitude of the coffee-ring flow speed was estimated from theoretical models²⁷ and experimental results⁵² of previous studies. To validate this, we also conducted micro-particle image velocimetry (μ -PIV) experiments that provided a good agreement between the flow fields measurement results and the theoretical values. ...”

On p. 5,

“The PIV results presented in Fig. 3a also confirmed that, after the solutal-Marangoni flows almost disappeared, the coffee-ring flows were observed, which showed an average speed U_c of approximately $1 - 2 \mu\text{m} \cdot \text{s}^{-1}$ (A and B regimes in Fig. 3a). Afterward, MeOH components totally evaporated, and then the remaining pure DI water evaporation caused the coffee-ring flows, which exhibited an average speed U_c of approximately $0.4 - 1 \mu\text{m} \cdot \text{s}^{-1}$ (C and D regimes in Fig. 3a), consistent with our previous prediction.” Next, we measured...

On p. 17,

Figure 3. Critical condition for drying morphology (uniform pattern Vs. coffee-ring pattern) of a CNC matrix depending on dewetting speed (U_d) and the coffee-ring flow speed (U_c). a, Results of micro-particle image velocimetry (μ -PIV)

experiments for droplet evaporation of MeOH and DI water mixture (70 : 30 vol.%) containing CNCs observed after the solutal-Marangoni flows almost disappeared at $t \approx 35$ s. We observed the temporal evolution of average U_c . The four sequential PIV snapshots (A)-(D) were summarized, depicting the weakening of solutal-Marangoni mixing flow in (A), followed by the formation of a coffee-ring flow in (B). The coffee-ring flow persisted in (C) and (D) due to the evaporation of the remaining DI water components. Typical flow structures were indicated by black arrows and moving contact lines were marked with black dashed circles. **b**, Variation of U_d with MeOH concentration (0, 30, and 70 vol.%) in DI water dispersed with 2.85 wt% CNC. **c**, Relative speed ratio ($= U_d/U_c$) depending on MeOH concentrations (0, 30, and 70 vol.%) with 2.85 wt% CNC. The results showed that the coffee-ring effect ($U_d/U_c \ll 1$) became predominant in the reddish regime. When the critical condition was satisfied ($U_d/U_c \gtrsim 1$), all CNCs were uniformly deposited and oriented during evaporation. t_e indicated the time when the CNC-containing drops were completely evaporated. **d**, Film thickness profiles of (c) along the a-a' line. $T_{\text{edge}}/T_{\text{center}}$ represents the ratio of film thickness between the edge and the center regions.

On p. 11, **Methods** section,
Characterization

...

To observe internal flow structures inside evaporating droplets, we used micro-particle image velocimetry (μ -PIV) as shown in Fig. 3a. The detailed setup was described in Fig. S2. We added fluorescent particles with a diameter of $1.9 \pm 0.1 \mu\text{m}$ and a concentration of 2.5% w/v to a solution containing 3.40 wt% CNCs in a mixture of MeOH and DI water. The particle concentration was 1.0% v/v in all solutions. The negatively charged surface of the fluorescent particles, with a charge of -15 ± 5 mV, repelled the surface of CNC particles with the same negative charge (see Materials in the Methods section), resulting in good dispersion of the particles within the solution for PIV measurements. The fluorescence signal was excited at a wavelength of 532 nm and emitted at 607 nm. An optical filter with a wavelength cutoff of 540 nm was used to detect the fluorescence signal, which was captured using a camera installed on a microscope. The droplets were deposited on a glass substrate and allowed to evaporate at a room temperature of 22 °C. Image sequences of the illuminated fluorescent particles were recorded using a camera that captured 50 frames per second. Since the droplet height (h) was much smaller than the droplet radius (R), i.e., $h/R < 1$, the focal plane was set near the bottom of the droplet, where the depth of field was about 150 - 200 μm from the cover glass. The PIVlab tool⁸⁸ in MATLAB was used for flow field calculations. To obtain vector fields, we applied iterative 2D cross-correlation of the particle images with multiple interrogation windows of 64×64 pixels (first) and 32×32 pixels (second) with 50 % overlaps, where the signal-to-noise ratio ($\text{SNR} > 3$) was satisfied for reliable PIV⁸⁹. Additionally, all fluorescent

		particles were able to closely follow the flow structures because their Stokes number was much smaller than unity ($St = \rho_p d_p^2 U / 18 \mu R \ll 1$), calculated based on the particle density $\rho_p \sim O(10^{-3} \text{ kg}\cdot\text{m}^{-3})$, the particle diameter $d_p \sim O(10^{-6} \text{ m})$, the dynamic viscosity of the solvent $\mu \sim O(10^{-1} \text{ mPa}\cdot\text{s})$, the flow speed $U \sim O(10^{-4} \text{ m}\cdot\text{s}^{-1})$, and the droplet radius $R \sim O(10^{-3} \text{ m})$. During PIV measurements, an acrylic box was placed over the droplet to prevent external forces from affecting the internal flows inside the droplet.
5	[Comment to the authors] When the MeOH was evaporated, how would the surface tension change at the contact line, as well as the top surface of the sessile droplet, in comparison to the center of the droplet.	[Our response] Thank you for the referee's comment. It is well-known that, during the evaporation of a droplet on a solid substrate, uneven evaporative flux profiles are generated along the liquid-gas interface, with the maximum flux at the contact line and the minimum flux at the center of the droplet ⁽¹⁾. Therefore, any liquid component in the droplet will evaporate more at the contact line. On top of this, phenomenon, we appreciate that MeOH is relatively much more volatile than DI water because of $P_{v,\text{water}} (\approx 2.33 \text{ kPa}) \ll P_{v,\text{MeOH}} (\approx 13.02 \text{ kPa})$, so MeOH will evaporate fast more compared to DI water. Under this circumstance, initially, the components of MeOH and DI water were evenly distributed in the droplet and near the droplet interface. Therefore, there could have a selective and random evaporation of the more volatile liquid component (here, MeOH) at the liquid-gas interface, which triggers a complicated mixing flows created by Marangoni instabilities ^(2,3) where $\gamma_{\text{water}} (\approx 72 \text{ mN/m}) > \gamma_{\text{MeOH}} (\approx 23 \text{ mN/m})$. In conclusion, when a droplet containing a mixture of MeOH and water evaporates, the evaporation flux of MeOH and DI water components is largest at the contact line, possibly resulting in relatively high surface tension at the contact line, where the concentration of DI water is relatively high, and a low surface tension at the center of the droplet, where the concentration of MeOH is relatively high. However, in this case, there is an interplay between the non-uniform evaporative flux and the Marangoni instabilities at the liquid-gas interface, so in reality the complicated mixing flows were observed, rather than circulating flows from the top of the droplet towards the contact line. Unfortunately, at the moment, it is impossible to directly measure the surface tension profile along the interface of the multi-component evaporating droplet. It is only possible to explain the surface tension profile based on the flow structures. We made some revisions to the text and figure in the revised manuscript. [References] (1) Deegan, R. D., Bakajin, O., Dupont, T. F., Huber, G., Nagel, S. R., & Witten, T. A. (1997). Capillary flow as the cause of ring stains from dried liquid drops. Nature, 389(6653), 827-829. (2) Kim, H. (2022). Multiple Marangoni flows in a binary mixture sessile droplet. Physics of

Fluids, 34(12), 122102.

(3) De La Cruz, R. A. L., Diddens, C., Zhang, X., & Lohse, D. (2021). Marangoni instability triggered by selective evaporation of a binary liquid inside a Hele-Shaw cell. *Journal of fluid mechanics*, 923, A16.

[Changed to the manuscript]

On p. 4, “Thus, initially, the suspended CNC particles were mixed well by solutal-Marangoni flows driven by surface tension gradients ~~along the liquid-gas interface between the MeOH and DI water components~~ between the MeOH and DI water components along the liquid-gas interface due to Marangoni instabilities^{29,62}. Simultaneously, according to the non-uniform evaporative flux²², in general, the surface tension might have been at the maximum value (γ_{high}) at the contact line, while the surface tension might have been at the minimum value (γ_{low}) at the apex of the droplet due to the selective evaporation of MeOH. However, the complicated mixing flows were observed due to Marangoni instabilities along the droplet interface, rather than circulating flows from the top of the droplet towards the contact line. This mixing pattern resulted from the interplay between the non-uniform evaporative flux and the Marangoni instabilities at the liquid-gas interface. (ii) As the droplet evaporated, MeOH rapidly evaporated...”

Figure 2. Deposition principle for the formation of a homogeneous quadrant CNC matrix. Hydrodynamic mechanisms for drying homogeneity of the CNC matrix. It undergoes the following steps: (i) solutal-Marangoni flows generated by selective evaporation mixed the suspended CNCs well, (ii) the droplet contact angle was decreased due to fast evaporation of volatile liquid components of evaporating drops while keeping the wetting area. A few seconds later, the spontaneous self-dewetting of the initial contact line was induced. (iii) evaporation-induced self-assembly (EISA) occurred in the vicinity of the moving contact line as a result of competition between the ~~spontaneous fast dewetting~~ fast self-dewetting (inward direction) and evaporatively-driven capillary flow (outward direction), and (iv) the uniformly distributed quadrant-shaped CNC matrix was achieved.

The logical connection is not smooth as turning the work from pure CNCs assembly to CNCs+AuNR assembly, as well as the anti-icing part. What is the purpose of the CNC+AuNRs assembly?

We apologize for the lack of explanation in the Introduction section. In this study, our main objective was to achieve the self-assembly of gold nanorods (GNRs) with a high degree of alignment and good drying uniformity. GNRs are highly promising nanomaterials with notable characteristics such as biocompatibility, chemical stability, ease of synthesis, and the ability to tune their surface plasmon resonance (SPR), which is advantageous in surface coating applications. Nevertheless, achieving uniform plasmonic GNR films through the conventional drop-casting process remains a challenge⁽¹⁻²⁾. Although extended periods of evaporation $\sim O(10^1)$ hours have proven successful in vertically aligning GNRs, this approach severely limits mass productivity. To overcome this limitation, we employed the cellulose nanocrystal (CNC) matrix as a "host template" for GNRs. CNC (cellulose nanocrystal) materials have garnered significant attention in the field of biomaterials due to their renewable nature, non-toxicity, and cost-effectiveness. Moreover, they exhibited robust optical⁽³⁾, thermal⁽⁴⁾, and mechanical⁽⁵⁾ properties. Thus, there are many attempts to utilize them in materials science and engineering field. However, prior to this step, it was imperative to establish homogeneous CNC matrices. Therefore, as a first step, our study focused on optimizing the coating conditions, including ink composition, and drying mechanisms, to attain the desired homogenous CNC matrices that can be used as a template of GNRs. Therefore, as a first step, our study focused on optimizing the coating conditions, including ink composition, and drying mechanisms, to attain the desired homogenous CNC matrices. We proved that, in the absence of the CNCs, uniform deposition and alignment of GNR particles cannot be achieved (see Fig. S7, Supporting Information).

Among various applications of GNR particles, we decided to check the plasmonic photothermal heating effect for anti-icing and/or de-icing. In comparison to conventional anti-icing applications, such as a self-assembled monolayer⁽⁶⁾, liquid-infused slippery wet surface⁽⁷⁾, anti-icing liquid spraying⁽⁸⁾, heat wire⁽⁹⁾, the CNC-GNR metasurface can provide notable advantages by eliminating the need for surface modification or chemical and physical treatments. The inherent advantages of CNC-GNR films suggest their immense potential for use in anti-icing applications. In this work, we clearly showed that the CNC-GNR films with multi-array forms having the photothermal plasmonic effect could be used for the anti-icing and/or de-icing substrate.

As the referee pointed out, we made some corrections to the 'Introduction Section' in the revised manuscript.

[References]

(1) Zaibudeen, A. W., Khawas, S., & Srivastava, S. (2021). Understanding multiscale assembly mechanism in evaporative droplet of gold nanorods. *Colloid and Interface Science*

		Communications, 44, 100492. (2) Zaibudeen, A. W., & Bandyopadhyay, R. (2022). Correlating the drying kinetics and dried morphologies of aqueous colloidal gold droplets of different particle concentrations. Colloids and Surfaces A: Physicochemical and Engineering Aspects, 646, 128982. (3) Droguet, B. E., Liang, H. L., Frka-Petesic, B., Parker, R. M., De Volder, M. F., Baumberg, J. J., & Vignolini, S. (2022). Large-scale fabrication of structurally coloured cellulose nanocrystal films and effect pigments. Nature Materials, 21(3), 352-358. (4) Li, T., Song, J., Zhao, X., Yang, Z., Pastel, G., Xu, S., Jia, C., Dai, J., Chen, C., Gong, A., J. Feng, Yao, Y., Fan, T., Yang, B., Wågberg, L., Yang, R., & Hu, L. (2018). Anisotropic, lightweight, strong, and super thermally insulating nanowood with naturally aligned nanocellulose. Science Advances, 4(3), eaar3724. (5) Wang, S., Jiang, F., Xu, X., Kuang, Y., Fu, K., Hitz, E., & Hu, L. (2017). Super-strong, super-stiff macrofibers with aligned, long bacterial cellulose nanofibers. Advanced Materials, 29(35), 1702498. (6) Ulman, A. (1996). Formation and structure of self-assembled monolayers. Chemical Reviews, 96(4), 1533-1554. (7) Liu, Q., Yang, Y., Huang, M., Zhou, Y., Liu, Y., & Liang, X. (2015). Durability of a lubricant-infused Electro spray Silicon Rubber surface as an anti-icing coating. Applied Surface Science, 346, 68-76. (8) Qin, C. C., Mulrone, A. T., & Gupta, M. C. (2020). Anti-icing epoxy resin surface modified by spray coating of PTFE Teflon particles for wind turbine blades. Materials Today Communications, 22, 100770. (9) Zhao, Z., Chen, H., Liu, X., Liu, H., & Zhang, D. (2018). Development of high-efficient synthetic electric heating coating for anti-icing/de-icing. Surface and Coatings Technology, 349, 340-346. [Changed to the manuscript] On p. 2, “Among these, CNC materials are considered one of the most promising nanoparticles as they can be derived from nature. Under this circumstance, we realized that although gold nanorods (GNRs) were considered one of the most useful and promising nanoparticles due to their biocompatible, chemically stable, relatively easy synthesis, and tunable surface plasmon resonance (SPR), achieving a uniform GNR film remains a significant challenge^{38, 39}. Previous studies have demonstrated success in vertically aligning GNRs through long evaporation times; however, this approach severely hampers productivity^{40, 41}. Therefore, our final goal objective is to use employ a CNC template film to produce create a homogeneous plasmonic metasurface of gold nanorods (GNRs) and generate harness the plasmonic heating effects.” On p. 5, “In the absence of the CNCs, uniform deposition and alignment of GNR particles cannot be achieved, as confirmed in Fig. S6. Fig. 4a outlines the procedures in detail, which are...”
7	[Comment to the authors] The authors declared that the find a facile approach to assembly CNCs. However, EISA is well-developed approach for assembly.	[Our response] We acknowledge that the EISA (Evaporation-Induced Self-Assembly) technique has gained popularity for self-assembling nanomaterials like CNCs, primarily due to its simplicity and production efficiency. However, to our best knowledge, the EISA is still far from complete to be used as a real application. Particularly, the effectiveness of the EISA technique heavily relies on achieving drying uniformity. Although there are so many studies about suppressing the coffee-ring effect, it still poses a significant

		challenge to achieving uniform drying during the EISA process, necessitating additional steps and increased complexity, such as substrate heating ⁽¹⁻²⁾, the use of ink additives ⁽³⁾, and even creating a volatile vapor environment ⁽⁴⁾. Given this background, our objective was to develop an EISA-based printing method that would be user-friendly, provide high levels of drying uniformity (i.e., coffee-ring-less pattern), and minimize the need for additional processing steps. We explained our new strategies based on the EISA phenomenon in detail in our response to Comment 1 and 4. [References] (1) Shao, R., Meng, X., Shi, Z., Zhong, J., Cai, Z., Hu, J., ... & Ye, C. (2021). Marangoni Flow Manipulated Concentric Assembly of Cellulose Nanocrystals. Small Methods, 5(11), 2100690. (2) Wang, H., Shao, R., Meng, X., He, Y., Shi, Z., Guo, Z., & Ye, C. (2022). Programmable Birefringent Patterns from Modulating the Localized Orientation of Cellulose Nanocrystals. ACS Applied Materials & Interfaces, 14(31), 36277-36286. (3) Gençer, A., Van Rie, J., Lombardo, S., Kang, K., & Thielemans, W. (2018). Effect of gelation on the colloidal deposition of cellulose nanocrystal films. Biomacromolecules, 19(8), 3233-3243. (4) Gençer, A., Schütz, C., & Thielemans, W. (2017). Influence of the particle concentration and marangoni flow on the formation of cellulose nanocrystal films. Langmuir, 33(1), 228-234.
8	[Comments to the authors] The introduction is needed to be improved significantly. There was lack of review about the assembly approach of CNCs, the assembly manner of CNCs and corresponding optical properties.	[Our response] The evaporation-induced self-assembly (EISA) phenomena of cellulose nanocrystals (CNCs) typically leads to two distinct self-assembly structures; helical structures and uniaxially aligned structures. The formation of these structures depends on the specific evaporation conditions. Specifically, the helical structures were commonly observed in static flow (bulk fluid) systems where the internal flows were relatively weak. In contrast, the uniaxially aligned patterns were seen in dynamic flow systems where the internal flows were relatively strong. In our study, we focused on the uniaxially aligned structure of CNCs driven by EISA in dynamic flow field, i.e., inside evaporating droplets. They exhibited relatively dynamic flow patterns such as solutal-Marangoni flow and coffee-ring flow during evaporation. These flows contribute to the formation of the uniaxial structure, rather than helical structure. Therefore, in the Introduction section, we dealt with only the uniaxial alignment structure of CNCs. To provide a clear understanding for the reviewer, we explained and compared two self-assembly mechanisms and resulting structures (i and ii) of CNCs as follows. Additionally, we added some sentences for a more detailed explanation in the revised manuscript. i) The helical (Bouligand) structure induced by chiral nematic phase. ii) The uniaxially aligned structure induced by flow-induced nematic phase.

i) Helical (Bouligand) structure induced by chiral nematic phase

In literature studies investigating the chiral self-assembly of CNCs, they filled bulk CNC solutions into a petri-dish with diameter ranging from 30 mm to 100 mm⁽¹⁻³⁾. In this situation, the evaporation speed is very slow due to their large initial volume. Additionally, the Bond number is much larger than unity. Here, $Bo = \rho g R^2 / \gamma = \text{Gravitational effect} / \text{Surface tension effect} = (R/l_{\text{cap}})^2 \gg 1$, where ρ is the liquid density, g is the gravitational acceleration, R is the radius of the petri-dish (or droplet), γ is the surface tension, and l_{cap} is the capillary length [= $(\gamma/\rho g)^{1/2}$]. For the MeOH and DI water mixture with 70 : 30 vol.% ratio, l_{cap} is approximately 1.9 mm, and the Bond number is $\sim O(10^3)$. Therefore, the dominant gravitational effect results in a relatively flat liquid-gas interface, leading to even evaporation flux profiles⁽⁴⁾. Consequently, in such a bulk system, we would observe relatively weak Marangoni flow and coffee-ring flow, which are in stark contrast to the evaporating droplet that exhibits uneven evaporative flux profiles. Moreover, the meniscus region, which is approximately on the order of $l_{\text{cap}} \sim O(1 \text{ mm})$, is very small compared to the diameter $\sim O(10 \text{ mm})$ of the petri-dish. Therefore, we can neglect this meniscus region, resulting in CNCs exhibiting a cholesteric phase, which is a thermodynamically stable state during the deposition process.

[Valuables and Calculation]

$\rho \approx 850 \text{ kg/m}^3$ for MeOH and DI water mixture (= 70 : 30 vol.%)

$\gamma \approx 30 \text{ mN/m}$ for MeOH and DI water mixture (= 70 : 30 vol.%)

$g \approx 9.81 \text{ m}^2/\text{s}$ for MeOH and DI water mixture (= 70 : 30 vol.%)

$R \approx 40 \text{ mm}$ for the petri-dish and $R \approx 1.5 \text{ mm}$ for the droplet

$l_{\text{cap}} = (\gamma/\rho g)^{1/2} \approx [30 \times 10^{-3} / (850 \times 9.81)]^{1/2} \approx 0.0019 \text{ m}$

$Bo = (R/l_{\text{cap}})^2 \approx (40 \times 10^{-3} / 0.0019) \approx 445 \gg 1$ for the petri-dish (diameter: 80 mm)

$Bo = (R/l_{\text{cap}})^2 \approx \left(0.15 \times \frac{10^{-3}}{0.0019}\right) \approx 0.6 < 1$ for the droplet (diameter: 3 mm)

ii) Uniaxially aligned structure induced by flow-induced nematic phase.

In contrast to the previous studies with bulk CNC solutions, for our millimeter-sized sessile droplets containing CNCs, the Bond number is less than unity. During droplet evaporation, the surface tension effect dominates, leading to the formation of a spherical cap shape with a small radius of curvature liquid-gas interface. In addition, the contact angle is smaller than 45° . As a result, a non-uniform evaporation flux profile was generated along the droplet interface, which raises relatively strong Marangoni flows and coffee-ring flows within the evaporating droplet. Hence, we can consider the evaporation of millimeter-sized droplets as a dynamic system. In this dynamic system, the cellulose nanocrystals exhibit a liquid crystalline phase but fail

to maintain their chiral structure⁽⁵⁾. Instead, they align along the shear direction in the deposited film⁽⁶⁻⁸⁾. For this reason, we omit discussing approach i) because we generated solutal-Maragnoni flows and utilized the competition between self-dewetting and coffee-ring flows, which induces the ii), uniaxial alignment of CNCs in the deposited film. We modified some content in the Introduction section to enhance reader comprehension.

[References]

- (1) Guidetti, G., Atifi, S., Vignolini, S., & Hamad, W. Y. (2016). Flexible photonic cellulose nanocrystal films. *Advanced Materials*, 28(45), 10042-10047.
- (2) Walters, C. M., Boott, C. E., Nguyen, T. D., Hamad, W. Y., & MacLachlan, M. J. (2020). Iridescent cellulose nanocrystal films modified with hydroxypropyl cellulose. *Biomacromolecules*, 21(3), 1295-1302.
- (3) Dumanli, A. G., Van Der Kooij, H. M., Kamita, G., Reisner, E., Baumberg, J. J., Steiner, U., & Vignolini, S. (2014). Digital color in cellulose nanocrystal films. *ACS Applied Materials & Interfaces*, 6(15), 12302-12306.
- (4) Zargartalebi, H., Hejazi, S. H., & Sanati-Nezhad, A. (2022). Self-assembly of highly ordered micro- and nanoparticle deposits. *Nature Communications*, 13(1), 3085.
- (5) Kádár, R., Spirk, S., & Nypelo, T. (2021). Cellulose nanocrystal liquid crystal phases: Progress and challenges in characterization using rheology coupled to optics, scattering, and spectroscopy. *ACS Nano*, 15(5), 7931-7945.
- (6) Skogberg, A., Mäki, A. J., Mettänen, M., Lahtinen, P., & Kallio, P. (2017). Cellulose nanofiber alignment using evaporation-induced droplet-casting, and cell alignment on aligned nanocellulose surfaces. *Biomacromolecules*, 18(12), 3936-3953.
- (7) Talantikite, M., Leray, N., Durand, S., Moreau, C., & Cathala, B. (2021). Influence of arabinoxylan on the drying of cellulose nanocrystals suspension: From coffee ring to Maltese cross pattern and application to enzymatic detection. *Journal of Colloid and Interface Science*, 587, 727-735.
- (8) Pritchard, C. Q., Navarro, F., Roman, M., & Bortner, M. J. (2021). Multi-axis alignment of Rod-like cellulose nanocrystals in drying droplets. *Journal of Colloid and Interface Science*, 603, 450-458.

[Changed to the manuscript]

On p. 1, **Introduction Section**,

“Especially, a drop-casting method is broadly used due to its simplicity, ease, and rapid streamlined procedure. When a droplet is drop-cast onto a solid substrate and then undergoes evaporation, the liquid-gas interface of the droplet typically takes on a curved shape if the surface tension effect dominates over the gravitational effect, specifically if the Bond number is less than unity and there is a hydrophilic contact angle. This droplet interface shape leads to an uneven distribution of the evaporation flux, creating a dynamic flow system inside the evaporating droplet. In this situation, the CNCs align in a uniaxial direction rather than a helical structure¹⁸⁻²⁰. However, ~~when the droplet of coating solution evaporates on a substrate once the droplet completely evaporates~~, an inhomogeneous dried morphology²¹ is inevitably generated due to evaporatively-driven capillary flows toward a contact line (so-called coffee-ring flows²²), which is the critical hurdle in coating and patterning applications.”

The AuNRs did not have a uniform rod-like morphology, and its UV-vis was lack after changing the ligand to PEG-SH. The stability of the AuNRs within the MeOH+DI mixture is also need to concerned.

Thank you for the referee's comment. In response, we conducted UV-vis absorption measurements on aqueous dispersions of gold nanorods (GNRs) with two different ligands: cetyltrimethylammonium bromide (CTAB) and polyethylene glycol (PEG). The obtained absorption spectra showed minimal differences between the two, indicating similar optical properties (This finding aligns with previous reports, such as Mahmoud, N. N. et al., *Sci. Rep.* 8, 6881, 2018).

To further investigate the morphology of PEG-coated gold nanorods (PEG-GNRs), we deposited the PEG-GNRs in a MeOH+DI water mixture onto a silicon substrate and observed them using the backscattered electron (BSE) mode of scanning electron microscopy (SEM). In the resulting SEM image, the bright domains corresponded to the presence of gold nanorods. Notably, despite the inclusion of MeOH in the solvent, we did not observe any noticeable agglomeration of the GNRs. Additionally, the distribution of GNR dimensions, as inferred from the SEM image analysis, appeared consistent, with widths ranging from 15 to 20 nm and lengths ranging from 45 to 60 nm.

[Changed to the manuscript]

On p. 17, **Figure 4**,

Figure 4. Concentrically aligned plasmonic metasurface of CNC-GNR matrices with tunable optical properties.

In Supporting Information,

Reviewer #2

First of all, we are pleased to read that the Reviewer noted that "... In general, this work provides a new route to produce thin films without coffee ring by binary solution method...". We thank the Reviewer for specific comments that must help us to improve the manuscript further. We reply to the Reviewer's questions and comments below, point by point, and have edited the manuscript in several places to reflect these remarks.

1	[Comments to the authors] The CNC-GNR plasmonic metasurface was prepared by the physical adoption of CNCs and GNRs without cross-linking, what about the water resistance of the film, because the contact with water is inevitable in anti-icing.	[Our response] Thank you for the referee's kind comment. To address the reviewer's concern about water resistance, we attempted to apply a waterproof layer onto the CNC-GNR film by brushing on nail polish oil and letting it dry at room temperature. We then compared the structural changes of the CNC-GNR film before and after laminating the waterproof layer. As illustrated in Fig. S9, we confirmed that the CNC-GNR pattern retained its structure and plasmonic performance, indicating that the deposition of the waterproof layer did not alter the film's characteristics. Using this CNC-GNR film with a waterproof layer laminated to it, we performed additional anti- and de-icing experiments as shown in Movies S11-S12. The result showed that the CNC-GNR films can prevent the formation of frost and melt the existing frost on the films under the sub-zero surface condition with just light irradiation within 10 minutes. We hope that these additional experimental results address the referee's concerns and highlight the potential of our CNC-GNR film for practical anti-icing applications. We believe that the water resistance test enhances the current research impact. Thank you again. [Changed to the manuscript] On p. 3, "The multi-array CNC-GNR metasurface was experimentally demonstrated to
---	---	--

have anti- and de-icing performance. From this result, we believe that this drop-casting CNC-GNR metasurface array can be used as a plasmonic photothermal film for anti-icing applications.”

On p. 6, “From this test, we concluded that the broadband plasmonic multi-array pattern of the CNC-GNR film holds promise as an anti-icing surface. This conclusion was further supported by actual anti- and de-icing experiments, which were shown in Movies S11-S12 in the Supporting Information. In these experiments, a waterproof layer was deposited on top of the completely dried CNC-GNR patterns without structural changes to the CNC-GNR structures, as observed in Fig. S9. Figure 5e showed that the CNC-GNR films exhibited frost prevention, despite the substrate temperature dropping to from 25°C to −8°C as shown in Movie S11. This prevention of frost formation can be attributed to the continuous plasmonic photothermal effect of the CNC-GNR films. Furthermore, even when a frost layer had already formed on the CNC-GNR films, it could melt away within 10 minutes of light irradiation under a sub-zero substrate condition. A detailed demonstration of this process is provided in Movie S12.”

On p. 6, “Moreover, when produced in a multi-array format, the CNC-GNR metasurface dramatically increased the temperature rises ($\geq 10^\circ\text{C}$) and remained above zero degrees even when the bottom substrate was below the freezing temperature, demonstrating excellent anti-icing and de-icing performance. Compared to conventional anti-icing systems, ...”

On p. 12, **Anti- and de-icing experiment setup in the Method section**, “...was irradiated and stimulated by a plasma light source (HPLS345, Thorlabs, USA). In anti-/de-icing experiments, the waterproof layer was deposited onto the completely dried CNC-GNR array pattern by brushing nail polish oil (Lucid Nail Polish, MISSHA, Korea) and letting it dry at room temperature.”

In Supporting Information,

Figure S11. Structural changes in CNC-GNR films upon water contact, without and with a waterproof layer. The CNC-GNR films (a) without and (b) with the waterproof layer were observed and compared using polarized optical microscopy. To prevent the structural changes of the co-assembled CNC-GNR from water, nail polish oil was brushed onto its surface and allowed to dry at room temperature to create a waterproof layer

Figure 5. Anti-icing application: enhancement of thermal performance of the plasmonic heater of CNC-GNR metasurfaces. a, Comparison of the plasmonic photothermal effects depending on two parameters; (i) uniformity of the CNC-GNR films [a light greenish box (coffee-ring film) vs. a greenish box (uniform film)] and (ii) GNR concentrations [a greenish box ($C_{\text{GNR}} \approx 0.14$ wt%) vs. a dark greenish box ($C_{\text{GNR}} \approx 0.56$ wt%)]. All thermal imaging snapshots were captured approximately 270 seconds after the light source was turned on (see Movies S8-S10, Supporting Information). All the scale bars are 10 mm. b, Measurement results of the temperature line profile along the y-y' line. The substrate temperature was about 26.5 ± 0.3 °C without the ambient light (see the gray-colored area). c, Illustration of a three-layer stacked solid substrate for controlling the substrate temperature T_{sub} of the cover glass (top) and silicon wafer (middle) layers. d, Multi-array (4×3) CNC-GNR films ($C_{\text{GNR}} \approx 0.56$ wt%) were deposited on the cover glass, as shown in (c) and inset of (d), and the plasmonic

		photothermal performance was evaluated using an infrared camera. Here, we controlled the substrate temperature T_{sub}. From 22°C to -8°C by adjusting the set temperature of the cooling metal plate (a bottom layer). During the measurement, the CNC-GNR arrays were exposed to a plasma light source. Detailed explanations are given in the Method. e, Results of anti- and de-icing experiments. While the light was being irradiated, the substrate temperature T_{sub}. remained constant at approximately -8°C. All white scale bars are 10 mm.
2	[Comment to the authors] The authors claimed that “we presented a straightforward and scalable method for producing a broadband plasmonic metasurface...”, but the samples prepared in manuscript by plasma etching and CNC-GNR solution drop-casting was millimeter-scale, which invites questions about the scalable preparation. And drop-casting method is working well on flat surface, how about the universality of this method, such as curved or irregular surfaces?	[Our response] Thank you for the referee’s insightful comment. We acknowledge that in this study, we only showed millimeter-sized CNC-GNR patterns. However, we firmly believe that our CNC-GNR coating solution has the potential to be applied to a large-format inkjet system. This would enable us to generate CNC-GNR patterns ranging from micro-scale to millimeter-scale while maintaining the same performance characteristics. Also, thank you for clarifying the details. We apologize for any confusion caused. Actually, we only applied the plasma surface treatment on the glass substrate during the optimization study for ink composition and determination of critical coating conditions as shown in Figs. 1 and 3. However, when the CNC-GNR film was fabricated, we deposited them on bare glass substrates without any surface treatment. To clarify this, we made the correction in the revised manuscript. While drop-casting is generally suitable for printing patterns on flat surfaces, we acknowledge its limitations when it comes to depositing patterns on curved or irregular surfaces. However, we believe that there are several approaches that can be explored to address this issue effectively. Techniques such as surface treatment or modification, including surface energy contrast, polymer coating, or pre-textured surfaces, can be considered. These modifications have the potential to enhance the adhesion of the patterns to the surface, minimizing the effects of surface curvature or irregularity. [Changed to the manuscript] On p. 11, Sample preparation in the Methods section, “To achieve this, during the experiments for Figs. 1 and 3, we controlled the droplet size by using plasma treatment (BD-10ASV, Electro-Technnic Products, INC., USA) on a cover glass substrate, which ...”
3	[Comment to the authors] Characterization of the surface morphologies of CNC matrix and CNC-GNR film are lacked, which can show the parallel alignments more intuitively.	[Our response] Thank you for the referee’s kind comment. In response to the reviewer’s comment, we investigated the surface morphology of a CNC film using AFM. To facilitate our study, we divided the film domain into three sections, including i) the inside of the droplet, ii) intermediate, and iii) outmost droplet area. We measured 3x3 μm square domain at each location on the line at 40 degrees. Three AFM height images were acquired, and we analyzed the angular distribution of CNC using MATLAB software, as described

in the article by Persson, N. E. et al. Chem. Mater. 29, 3-14, (2017). The resulting distribution maps demonstrated a dumbbell-like structure with an average angle of approximately 140°. This implies that the uniaxial alignment of CNC is parallel to the droplet contact line, regardless of the domain locations.

Moreover, we characterized the alignment of gold nanorods in the CNC matrix using SEM. Similar to AFM, we captured SEM images at the same three domains (i, ii, and iii) using backscattered electron (BSE) mode. The BSE mode enabled the gold nanorods in the CNC matrix to be highlighted brightly in the SEM image. Our results indicate that the surface morphology of the CNC-GNR film shows uniaxial alignment regardless of domains. Furthermore, the BSE images in the same area demonstrated that the gold nanorods followed the alignment of the CNC matrix.

[Changed to the manuscript]

On p. 4, "... As a result, we obtained a uniformly crystallized CNC matrix in a quadrant orientation, without any changes to their LC structures (see the blue and yellow textures of Fig. 1e and the AFM images of Fig. S4c)."

On p. 5, "...[see the illustration (i) and (ii) of Fig. 4d]. Additionally, SEM images of the CNC-GNR metasurface were provided in Fig. S7b, which offers a more detailed visualization. Moreover, we also demonstrated continuous plasmonic color changes of the CNC-GNR metasurface by rotating a single polarizer (see orientation-dependent images, Fig. 4e and ~~Fig-S2~~Fig. S8)."

In Supporting Information,

Figure S4. Characterization of CNC particles and CNC films by Atomic Force Microscope (AFM). **a**, Representative AFM images. **b**, Histogram of width and length distribution of CNC particles. **c**, Representative AFM images of CNCs dispersed in a MeOH and DI water mixture. **d**, Histogram of width and length distribution of CNC particles in a MeOH and DI water mixture **e**, AFM images (top) and rotation distribution (bottom) of the CNC film at ⁽ⁱ⁾ the inner ring, ⁽ⁱⁱ⁾ the intermediate, and ⁽ⁱⁱⁱ⁾ the outer region in the direction of 40° . The CNC film was deposited by evaporating a

droplet containing MeOH : DI water : CNCs = 62.75 : 33.85 : 3.40 wt%. The order parameter S indicates the degree of alignment of the CNCs.

Figure S7. Characterization of CNC-GNR films by Scanning Electron Microscope-Backscattered Electron (SEM-BSE). **a**, Representative SEM images of PEG-GNR particles. **b**, SEM (top) and BSE (bottom) images of the CNC-GNR film at (i) the inner, (ii) the intermediate, and (iii) the outer ring region in the direction of 40° . For the SEM-BSE measurements, the CNC-GNR film was deposited onto a silicon wafer substrate by evaporating the droplet composed of MeOH : DI water : CNCs : GNRs = 63.11 : 33.46 : 3.40 : 0.04 wt%.

[Comment to the authors]

For the application in anti-icing, the photothermal effect on CNC-GNR metasurface was measured under white light with irradiance intensity of $500 \pm 50 \text{ mW}\cdot\text{cm}^{-2}$, which was 5 times of solar light ($100 \text{ mW}\cdot\text{cm}^{-2}$). The photothermal performance on CNC-GNR film is weak compared with other anisotropic nanomaterials like CNTs. In this part, only surface temperature was measured, the anti-freezing performance of water, or the photothermal deicing tests are missing. The CNC-GNR shows no superior in anti-icing.

[Our response]

Thank you for the referee's comment. As pointed out by the referee, we acknowledge and appreciate the referee's observation that demanding anti-frosting or anti-deicing tests for practical industrial applications were not extensively conducted in this study. The primary objective of our work was to propose a method for fabricating a coffee-ring-less and well-aligned CNC-GNR film using a simple drop-casting technique without the need for additional processing steps such as additives, heating, or substrate modification. Furthermore, we tried to demonstrate the potential of the photothermal effect of our CNC-GNR film for anti-frosting or anti-deicing applications by measuring the surface temperature on a sub-cooled substrate. However, we would like to note that we further conducted anti- and de-icing experiments as shown in Movies S11-S12. The result showed that the CNC-GNR films can prevent the formation of frost and melt the existing frost on the films under sub-zero surface conditions with just light irradiation within 10 minutes. We hope that these additional experimental results address the referee's concerns and highlight the potential of our CNC-GNR film for practical anti-icing applications.

We believe that the CNC-GNR films proposed in this work possess a significant competitive advantage over other anisotropic nanomaterials, including carbon nanotubes (CNTs) ⁽¹⁾. One of the key advantages of CNC-GNR films is their favorable dispersion in most solvents, enabling easy utilization through the drop-casting method. The drop-casting method has garnered attention in diverse fields such as electronics,

sensors, and displays due to its simplicity, cost-effectiveness, and scalability. From a process and production perspective, CNC-GNR films exhibit exceptional potential as a highly competitive candidate for anti-icing applications. among anisotropic nanomaterials. To sum up, the main idea of the current research is to introduce the alignment of nanomaterials based on the CNC matrix as a template.

Lastly, to consider about the reviewer's concern, it would be appreciated if the reviewer could acknowledge that we used visible light with a broad range of wavelengths (350 - 800 nm) and low output power ($500 \pm 50 \text{ mW} \cdot \text{cm}^{-2}$) compared to previous studies.

In previous studies, near-infrared light (810 nm) with high power ($2 \text{ W} \cdot \text{cm}^{-2}$) was employed ⁽²⁾. For this reason, it appears that our CNC-GNR film exhibits somewhat lower photothermal performance. However, our focus was on developing new plasmonic heaters using CNC-GNR films that can be operated by incident light with a broadband wavelength range similar to solar radiation $\sim 0(100 \text{ mW} \cdot \text{cm}^{-2})$.

[References]

(1) Geckeler, K. E., & Premkumar, T. (2011). Carbon nanotubes: are they dispersed or dissolved in liquids?. *Nanoscale Research Letters*, 6, 1-3.

(2) von Maltzahn, G., Centrone, A., Park, J. H., Ramanathan, R., Sailor, M. J., Hatton, T. A., & Bhatia, S. N. (2009). SERS-coded gold nanorods as a multifunctional platform for densely multiplexed near-infrared imaging and photothermal heating. *Advanced Materials*, 21(31), 3175-3180.

[Changed to the manuscript]

On p. 3, “The multi-array CNC-GNR metasurface was experimentally demonstrated to have anti- and de-icing performance. From this result, we believe that this drop-casting CNC-GNR metasurface array can be used as a plasmonic photothermal film for anti-icing applications.”

On p. 6, “From this test, we concluded that the broadband plasmonic multi-array pattern of the CNC-GNR film holds promise as an anti-icing surface. This conclusion was further supported by actual anti- and de-icing experiments, which were shown in Movies S11-S12 in the Supporting Information. In these experiments, a waterproof layer was deposited on top of the completely dried CNC-GNR patterns without structural changes to the CNC-GNR structures, as observed in Fig. S9. Figure 5e showed that the CNC-GNR films exhibited frost prevention, despite the substrate temperature dropping to from 25°C to -8°C as shown in Movie S11. This prevention of frost formation can be attributed to the continuous plasmonic photothermal effect of the CNC-GNR films. Furthermore, even when a frost layer had already formed on the CNC-GNR films, it could melt away within 10 minutes of light irradiation under a sub-zero substrate condition. A detailed demonstration of this process is provided in Movie S12.”

On p. 6, “Moreover, when produced in a multi-array format, the CNC-GNR metasurface dramatically increased the temperature rises ($\geq 10^{\circ}\text{C}$) and remained above zero degrees even when the bottom substrate was below the freezing temperature, demonstrating excellent anti-icing and de-icing performance. Compared to conventional anti-icing systems, ...”

On p. 12, **Anti- and de-icing experiment setup in the Method section**, “...was irradiated and stimulated by a plasma light source (HPLS345, Thorlabs, USA). In anti-/de-icing experiments, the waterproof layer was deposited onto the completely dried CNC-GNR array pattern by brushing nail polish oil (Lucid Nail Polish, MISSHA, Korea) and letting it dry at room temperature.”

In Supporting Information,

Figure S11. Structural changes in CNC-GNR films upon water contact, without and with a waterproof layer. The CNC-GNR films (a) without and (b) with the waterproof layer were observed and compared using polarized optical microscopy. To prevent the structural changes of the co-assembled CNC-GNR from water, nail polish

oil was brushed onto its surface and allowed to dry at room temperature to create a waterproof layer.

Figure 5. Anti-icing application: enhancement of thermal performance of the plasmonic heater of CNC-GNR metasurfaces. a, Comparison of the plasmonic photothermal effects depending on two parameters; (i) uniformity of the CNC-GNR films [a light greenish box (coffee-ring film) vs. a greenish box (uniform film)] and (ii) GNR concentrations [a greenish box ($C_{\text{GNR}} \approx 0.14$ wt%) vs. a dark greenish box ($C_{\text{GNR}} \approx 0.56$ wt%)]. All thermal imaging snapshots were captured approximately 270 seconds after the light source was turned on (see Movies S8-S10, Supporting Information). All the scale bars are 10 mm. b, Measurement results of the temperature line profile along the y-y' line. The substrate temperature was about 26.5 ± 0.3 °C without the ambient light (see the gray-colored area). c, Illustration of a three-layer stacked solid substrate

		for controlling the substrate temperature T_{sub} of the cover glass (top) and silicon wafer (middle) layers. d, Multi-array (4×3) CNC-GNR films ($C_{\text{GNR}} \approx 0.56$ wt%) were deposited on the cover glass, as shown in (c) and inset of (d), and the plasmonic photothermal performance was evaluated using an infrared camera. Here, we controlled the substrate temperature T_{sub}. From 22°C to −8°C by adjusting the set temperature of the cooling metal plate (a bottom layer). During the measurement, the CNC-GNR arrays were exposed to a plasma light source. Detailed explanations are given in the Method. e, Results of anti- and de-icing experiments. While the light was being irradiated, the substrate temperature T_{sub} remained constant at approximately −8°C. All white scale bars are 10 mm.
--	--	---

Reviewer #3

First of all, we are pleased to read that the Reviewer noted that “... The authors report some potentially interesting thermal response upon addition of gold nanorods into the suspension...”. We thank the Reviewer for specific comments that must help us to improve the manuscript further. We reply to the Reviewer’s questions and comments below, pint by point, and have edited the manuscript in several places to reflect these remarks.

1	The discussion and data on selecting a CNC composition is confusing. The limited discussion does not clearly explain the rationale – for instance, 3 videos of 2.8wt% CNC are included in the SI, but only one with 3.4% CNC (without other materials). However, the authors suggest 3.4% is good because of more uniformity, with little discussion in how they are defining uniformity (see later comment). Further, there are significant considerations for concentration (particularly initial concentration prior to evaporation) that strongly influence the results, as highlighted in the next comment about the starting state of the CNCs. A brief mention of 3 concentrations that are in a relatively similar range of CNC behavior were “evaluated” (Figure 1e) but with very little discussion. (and again another reference to “optimal”, please see comment below in minor comments).	[Our response] We apologize for any insufficiency in our explanation and supporting materials. An additional SI Movie for the $C_{\text{CNC}} \approx 3.40$ wt% case was added (see the Movie S4), which showed the evaporation of CNC-containing droplets under the optimized condition. We aimed to find the initial CNC concentration in DI water for achieving better drying uniformity (i.e., uniform deposition thickness) after complete evaporation. In other words, to obtain the subsequent co-assembly of CNCs with GNRs, it is crucial to have an adequate film thickness. If the initial CNC concentration is not enough, the insufficient thickness of the CMC film would be deposited in the middle of the droplet, as illustrated in Fig. 1d. In this situation, it caused incomplete co-assembly with anisotropic GNRs and degraded the plasmonic optical and thermal performance of CNC-GNR composite films. Therefore, we controlled the initial CNC concentration in the coating solution to find the optimal composition (MeOH : DI water : CNCs = 62.75 : 33.85 : 3.40 wt% or 68.71 : 29.45 : 1.84 vol.%) for fabricating the uniform CNC matrix film as shown in side-view SEM images in Fig. S3. We defined it as an optimal condition in this paper. To emphasize the importance of CNC concentration, we added some sentences in the revised manuscript. [Changed to the manuscript] On p. 3, “...as shown in Fig. 1e. As C_{CNC} increased, it showed that more homogeneous quadrant CNC patterns were achieved at $C_{\text{CNC}} \approx 3.4$ wt%. As illustrated in Fig. 1d, when the concentration of CNCs was insufficient, there were not enough CNCs deposited in the center of the droplet, which rendered it unsuitable as a host template for co-assembly with functional materials. To overcome this issue, we
---	---	---

increased the CNC concentration as depicted in Fig. 1e. Our experiments revealed that a CNC concentration of approximately 3.40 wt% led to more homogeneous CNC patterns, as evidenced by Movie S4.”

In Supporting Information,

Figure S3. Analysis of CNC film thickness using side-view Scanning Electron Microscope (SEM) Images. The side view of the CNC film was captured in sequential order from the vicinity of the droplet’s edge to the vicinity of the droplet’s center (i - iv). The average thickness was measured at each location. All scale bars are 5 μm.

2

CNC preparation and characterization – the authors do not provide any characterization information about the nanocellulose other than size by TEM. A protocol exists for well-defined characterization using AFM of dilute CNCs on spin-cast mica wafers, how did the authors determine “width”? Characterization information that is critical for reproducing results, because of the sensitivity of the CNC colloidal state to these parameters, includes quality of redispersion following freeze-drying, surface charge density in suspension, pH, etc. – these factors and others are critical for colloidal stability and will strongly impact the ability for someone to reproduce the results. Further, the authors did not dialyze their CNCs following production, “simple” washing with water is not typically sufficient to purify the suspensions following hydrolysis (it is often one step in the process). Therefore, there is some concern about the state of the starting material that should be thoroughly addressed.

[Our response]

Thank you for your comment. We appreciate the reviewer's suggestion regarding the variability of cellulose nanocrystals (CNCs) resulting from the synthesis and dispersion processes. In order to assess the quality and characteristics of the CNCs produced using our specific methodology outlined in the Materials and Methods section, we conducted an analysis.

To characterize the fabricated CNCs, we employed atomic force microscopy (AFM) to measure the height profile of diluted CNC samples deposited on a mica substrate. The width and length of the CNCs were estimated using a custom MATLAB program, referencing the studies by Usov et al. (*Nat. Commun.* 6, 7564, 2015) and Persson et al. (*ACS Appl. Mater. Interfaces* 9, 36090-36102, 2017).

By analyzing a dataset comprising 459 measurements, we observed a maximum distribution of approximately 24 nm for the width of the CNCs and approximately 100 nm for their length. These findings provide insight into the size distribution and dimensions of the CNCs generated through our process. We also found them to be consistent with DLS results dispersed in a mixture of methanol and water (= 70 : 30 vol.%), where the CNCs were safely dispersed in the mixture, as verified by ζ-potential measurements (AFM: Fig. S4a-d, ζ-potential: Fig. S13, and DLS: Fig. S14).

We hope this information clarifies the characterization of CNCs in our study, demonstrating the methodology employed to evaluate their dimensions. The measurement results for the CNC characterization were added in Supporting Information.

[Changed to the manuscript]

On p. 10, **Materials in the Methods section**,

“To evaluate the colloidal stability of a dispersion containing CNCs or GNRs in a mixture of MeOH and water (= 70 : 30 vol.%), we measured the ζ -potential and pH using a particle size analyzer (Zetasizer Nano-ZS, Malvern Panalytical, UK) and a portable pH meter (Orion Star™ A221 pH Portable Meter, Thermo Fisher Scientific Inc., USA). From the measurements, the ζ -potential and pH were -33.9 ± 10.4 mV and 6.84 ± 0.05 for the CNC dispersion and were -5.26 ± 9.0 mV and 6.99 ± 0.03 for the PEG-GNR dispersion (see also Fig. S13). Here, the ζ -potential of PEG-GNR was found to be close to neutral. This indicates that PEG-GNR did not display significant electrostatic interactions with CNC. Instead, the PEG-GNR employed a steric hindrance effect to prevent particle aggregation, ensuring colloidal stability in the CNC-GNR ink, as depicted in Fig. 4a. This was further supported by Dynamic Light Scattering (DLS) analysis, which confirmed the stable dispersion of CNC or PEG-GNR particles in the mixture solution of MeOH and DI water without any aggregation as shown in Fig. S14. Additionally, the ζ -potential data presented in Fig. S13b suggest that the ligand exchange from CTAB to PEG proceeded smoothly (see also the UV-vis spectra in the inset of Fig. 4a.). For the ζ -potential and DLS data reliability, each measurement was taken at least three times.”

In Supporting Information,

Figure S4. Characterization of CNC particles and CNC films by Atomic Force Microscope (AFM). **a**, Representative AFM images. **b**, Histogram of width and length distribution of CNC particles. **c**, Representative AFM images of CNCs dispersed in a MeOH and DI water mixture. **d**, Histogram of width and length distribution of CNC particles in a MeOH and DI water mixture. **e**, AFM images (top) and rotation distribution (bottom) of the CNC film at ⁽ⁱ⁾ the inner ring, ⁽ⁱⁱ⁾ the intermediate, and ⁽ⁱⁱⁱ⁾ the outer region in the direction of 40° . The CNC film was deposited by evaporating a

droplet containing MeOH : DI water : CNCs = 62.75 : 33.85 : 3.40 wt%. The order parameter S indicates the degree of alignment of the CNCs.

Figure S13. Results of ζ -potential measurements of the CNC and GNR dispersions with a MeOH and DI water mixture. All particles were dispersed in the binary mixture containing 70% MeOH and 30% DI water, based on the volume ratio. The particle concentrations were approximately 1 mg/mL.

Figure S14. Size distribution of CNC and GNR particles in a MeOH and DI water mixture. Dynamic Light Scattering (DLS) analysis was conducted to determine the size distribution of the particles and assess their colloidal stability in the binary mixture containing 70% MeOH and 30% DI water. The particle concentrations were approximately 1 mg/mL.

of the solution with water and any documented literature on the actual vapor pressure. The authors should at minimum comment on the relative order of magnitude impact of making this assumption that water does not evaporate and is not very volatile compared to methanol, and further this reviewer would expect that droplet size and ambient conditions further play into this assumption.

^(1,2), it has been observed that when there is a significant difference in the vapor pressure between the two solvents, the solvent with a lower vapor pressure can be neglected in the evaporation process, and the solvent with a higher vapor pressure evaporates first. This phenomenon is known as "selective evaporation". This selective evaporation produced solutal-Marangoni effects in the early evaporation stage. Based on this finding, it has been widely accepted that binary mixture droplet evaporation can be explained well without considering the contribution of solvents with low vapor pressure. To provide literature on the vapor pressure of MeOH and DI water, we added a few references in the revised manuscript.

In our opinion, the ambient vapor conditions have a minor effect on our case unless the droplet size is too small to evaporate quickly. This is because the vapor diffusion rate in the ambient air is much larger than the convective mass transfer of the evaporating droplet. Mathematically, this can be expressed using Sherwood number (Sh) = $hR/D_{va} \approx V^{1/3}R/t_e D_{va} \sim O(10^{-3} - 10^{-4}) \ll 1$, where $h \approx (V^{1/3}/t_e)$ is the convective mass transfer rate from the liquid droplet to the surrounding gas ($m \cdot s^{-1}$), V is the initial droplet volume $\sim O(10^{-9} m^3)$, t_e is the total evaporation time $\sim O(10^2 - 10^3 s)$, R is the droplet radius $\sim O(10^{-3} m)$, and D_{va} is the vapor diffusion coefficient $\sim O(10^{-5} m^2 \cdot s^{-1})$. Additionally, the experimental environment has a relatively low relative humidity (RH) level of approximately $30 \pm 2\%$, which can be considered a dry condition. This level of humidity is not expected to interfere with the diffusion of the evaporated vapors from the droplet in the air. Consequently, the evaporated vapor concentration near the droplet remains unchanged from the initial state, which can be regarded as being in a quasi-steady state ⁽³⁾. To improve readability for readers, we added our assumption for the ambient vapor condition in the revised manuscript.

[References]

- (1) Kim, H., & Stone, H. A. (2018). Direct measurement of selective evaporation of binary mixture droplets by dissolving materials. *Journal of Fluid Mechanics*, 850, 769-783.
- (2) Li, Y., Lv, P., Diddens, C., Tan, H., Wijshoff, H., Versluis, M., & Lohse, D. (2018). Evaporation-triggered segregation of sessile binary droplets. *Physical Review Letters*, 120(22), 224501.
- (3) Hu, H., & Larson, R. G. (2002). Evaporation of a sessile droplet on a substrate. *The Journal of Physical Chemistry B*, 106(6), 1334-1344.

[Changed to the manuscript]

On p. 3, "The vapor pressure of MeOH was much higher than that of DI water (i.e., $P_{v,MeOH} \approx 13.02 \text{ kPa}$ ⁵⁸ $\gg P_{v,water} \approx 2.33 \text{ kPa}$ ⁵⁹ at surrounding temperature 22°C...."

On p. 10, "~~In our problem, a solution of the MeOH and DI water mixture was used~~

~~during experiments where MeOH was much more volatile than DI water ($\because P_{v,MeOH} \gg P_{v,water}$)~~ Our experiment used a solution consisting of a MeOH and DI water mixture, with MeOH having a vapor pressure almost six times larger than that of DI water at room temperature ($\because P_{v,MeOH} \approx 13.02 \text{ kPa}^{58}$ and $P_{v,water} \approx 2.33 \text{ kPa}^{59}$). Thus, we could neglect the DI water evaporation until the MeOH component in the droplet has completely evaporated, so ~~and then substitute~~ the physical properties⁸⁶ of the pure MeOH were used in all equations. During evaporation, the changes in the vapor concentration around the droplet might be negligible, even though the vapors were continuously generated from the droplet. This was because the vapor diffusion rate in the ambient air is much larger than the convective mass transfer of the evaporating droplet. Mathematically, this can be expressed using Sherwood number (Sh) = $hR/D_{va} \approx V^{1/3}R/t_e D_{va} \sim O(10^{-3} - 10^{-4}) \ll 1$, where $h \approx (V^{1/3}/t_e)$ is the convective mass transfer rate from the liquid droplet to the surrounding gas ($\text{m} \cdot \text{s}^{-1}$), V is the initial droplet volume $\sim O(10^{-9} \text{ m}^3)$, t_e is the total evaporation time $\sim O(10^2 - 10^3 \text{ s})$, R is the droplet radius $\sim O(10^{-3} \text{ m})$, and D_{va} is the vapor diffusion coefficient $\sim O(10^{-5} \text{ m}^2 \cdot \text{s}^{-1})$. Additionally, the experimental environment had a relatively low relative humidity (RH) level of approximately $30 \pm 2\%$, which can be considered a dry condition. This level of humidity was not expected to interfere with the diffusion of the evaporated vapors from the droplet in the air. Consequently, the evaporated vapor concentration near the droplet remains unchanged from the initial state, which can be regarded as being in a quasi-steady state⁸⁹. Unless the droplet size is too small to evaporate quickly, the ambient vapor conditions have a minor effect on our case. Hence, the concentration gradient of MeOH vapor was assumed to be $\Delta C = C_s - C_\infty \approx C_s \approx 0.196 \text{ kg} \cdot \text{m}^{-389}$ ($\because C_\infty$ was initially about zero in ambient air) at surrounding temperature $T \approx 22^\circ\text{C}$. ~~For MeOH at surrounding temperature $T \approx 22^\circ\text{C}$, the vapor diffusion coefficient of MeOH in the air $D_{va} \approx 1.5 \times 10^{-5} \text{ m}^2 \cdot \text{s}^{-190}$, the concentration gradient of MeOH vapor $\Delta C = C_s - C_\infty \approx C_s \approx 0.186 \text{ kg} \cdot \text{m}^{-387}$ ($\because C_\infty \approx 0$ in ambient air) and the density of MeOH $\rho \approx 792 \text{ kg} \cdot \text{m}^{-3}$ and the initial contact angle $\theta(0) = \theta_0 \approx 32 \pm 2^\circ$.~~ With the physical properties above, we calculated the velocity of the coffee-ring flow U_c near the contact line [at $z = 2 \text{ }\mu\text{m}$ and $R - r = 150 \text{ }\mu\text{m}$] at $t = 0 \text{ s}$, with the initial contact angle $\theta(0) \approx 32 \pm 2^\circ$ and finally obtained $U_c \sim O(1 \text{ }\mu\text{m} \cdot \text{s}^{-1})$ for drop evaporation of the MeOH and DI water mixture, while the MeOH component was present in the droplet. In the same way, we also calculated the coffee-ring flow velocity U_c for a pure DI water ~~ease~~condition that occurred in the following two scenarios: (i) when the MeOH component in the MeOH and DI water mixture droplet was totally evaporated, leaving only DI water, and (ii) when a pure DI water droplet was evaporated. In these cases, the following values were used: $D_{va} \approx 2.5 \times 10^{-5} \text{ m}^2 \cdot$

		s^{-191} , $\Delta C = C_s - C_\infty = C_s(1 - RH) \approx 0.012-0.016 \text{ kg} \cdot \text{m}^{-3}$, $RH \approx 0.3$, and $\rho \approx 998 \text{ kg} \cdot \text{m}^{-3}$. The calculation result was $U_c \sim O(0.1 \mu\text{m} \cdot \text{s}^{-1})$.”
4	A significant effort was invested to understand the droplet evaporation process and analysis of the CNC alignment process – and given the extensive references provided by the author, one can deduce that significant effort has been invested in the community to understand the evaporation induced assembly of CNCs. Then the authors added gold nanoparticles without discussing the influence of the new nanomaterials on the colloidal stability, phase behavior, etc. all of which will significantly influence the droplet dynamics – and at this point, the authors suggest that they were still able to maintain quadrant uniformity in dried droplets, but neglect to characterize the impact on the starting CNC suspension characteristics (which ties to above comment about the starting materials).	[Our response] Thank you for the referee’s kind comment. To assess the impact of adding GNRs to the CNC dispersion, we performed zeta-potential measurements for the CNC and GNR dispersion in a MeOH and DI water mixture solution. This analysis enabled us to assess the colloidal stability of the CNC and GNR particles upon their combination. Interestingly, both particles exhibited the same negative surface charges, indicating their stable co-dispersion in the CNC-GNR ink. Furthermore, we examined the particle size distribution when they were dispersed in the MeOH and DI water mixture, and the results demonstrated that they remained dispersed without any aggregation. Please see our response and corrections in Comment No. 2.
5	The authors make a major leap from a limited set of measurements to suggesting widespread and broad application of the materials. The term anti-icing, in itself, has many connotations in the community and should be carefully implemented. Without significant additional analysis, this reviewer feels that the authors should reduce the broad overarching application potential with such a limited set of data. To this effect, the title specifically indicates "for anti-icing" and should be adjusted.	[Our response] Thank you for the referee’s comment. As pointed out by the referee, we acknowledge and appreciate the referee's observation that anti-frosting or anti-deicing tests for practical industrial applications were not conducted in this study. The primary objective of our work was to propose a method for fabricating a coffee-ring-less and well-aligned CNC-GNR film using a simple drop-casting technique without the need for additional processing steps such as additives, heating, or substrate modification. Furthermore, we tried to demonstrate the potential of the photothermal effect of our CNC-GNR film for anti-frosting or anti-deicing applications by measuring the surface temperature on a sub-cooled substrate. However, we would like to note that we further conducted anti- and de-icing experiments as shown in Movies S11-S12. The result showed that the CNC-GNR films can prevent the formation of frost and melt the existing frost on the films under the sub-zero surface conditions with just light irradiation within 10 minutes. We hope that these additional experimental results address the referee’s concerns and highlights the potential of our CNC-GNR film for practical anti-icing applications. [Changed to the manuscript] On p. 3, “The multi-array CNC-GNR metasurface was experimentally demonstrated to have anti- and de-icing performance. From this result, we believe that this drop-casting CNC-GNR metasurface array can be used as a plasmonic photothermal film for anti-icing applications.” On p. 6, “From this test, we concluded that the broadband plasmonic multi-array pattern of the CNC-GNR film holds promise as an anti-icing surface. This conclusion was further supported by actual anti- and de-icing experiments, which were shown in Movies S11-S12 in the Supporting Information. In these experiments, a waterproof

layer was deposited on top of the completely dried CNC-GNR patterns without structural changes to the CNC-GNR structures, as observed in Fig. S9. Figure 5e showed that the CNC-GNR films exhibited frost prevention, despite the substrate temperature dropping to from 25°C to −8°C as shown in Movie S11. This prevention of frost formation can be attributed to the continuous plasmonic photothermal effect of the CNC-GNR films. Furthermore, even when a frost layer had already formed on the CNC-GNR films, it could melt away within 10 minutes of light irradiation under a sub-zero substrate condition. A detailed demonstration of this process is provided in Movie S12.”

On p. 6, “Moreover, when produced in a multi-array format, the CNC-GNR metasurface dramatically increased the temperature rises ($\geq 10^\circ\text{C}$) and remained above zero degrees even when the bottom substrate was below the freezing temperature, demonstrating excellent anti-icing and de-icing performance. Compared to conventional anti-icing systems, ...”

On p. 12, **Anti- and de-icing experiment setup in the Method section**, “...was irradiated and stimulated by a plasma light source (HPLS345, Thorlabs, USA). In anti-/de-icing experiments, the waterproof layer was deposited onto the completely dried CNC-GNR array pattern by brushing nail polish oil (Lucid Nail Polish, MISSHA, Korea) and letting it dry at room temperature.”

In Supporting Information,

Figure S11. Structural changes in CNC-GNR films upon water contact, without and with a waterproof layer. The CNC-GNR films (a) without and (b) with the waterproof layer were observed and compared using polarized optical microscopy. To prevent the structural changes of the co-assembled CNC-GNR from water, nail polish oil was brushed onto its surface and allowed to dry at room temperature to create a waterproof layer

Figure 5. Anti-icing application: enhancement of thermal performance of the plasmonic heater of CNC-GNR metasurfaces. a, Comparison of the plasmonic photothermal effects depending on two parameters; (i) uniformity of the CNC-GNR films [a light greenish box (coffee-ring film) vs. a greenish box (uniform film)] and (ii) GNR concentrations [a greenish box ($C_{\text{GNR}} \approx 0.14 \text{ wt}\%$) vs. a dark greenish box ($C_{\text{GNR}} \approx 0.56 \text{ wt}\%$)]. All thermal imaging snapshots were captured approximately 270 seconds after the light source was turned on (see Movies S8-S10, Supporting Information). All the scale bars are 10 mm. b, Measurement results of the temperature line profile along the y-y' line. The substrate temperature was about $26.5 \pm 0.3^\circ\text{C}$ without the ambient light (see the gray-colored area). c, Illustration of a three-layer stacked solid substrate for controlling the substrate temperature T_{sub} of the cover glass (top) and silicon wafer (middle) layers. d, Multi-array (4×3) CNC-GNR films ($C_{\text{GNR}} \approx 0.56 \text{ wt}\%$) were deposited on the cover glass, as shown in (c) and inset of (d), and the plasmonic

		photothermal performance was evaluated using an infrared camera. Here, we controlled the substrate temperature T_{sub}. From 22°C to -8°C by adjusting the set temperature of the cooling metal plate (a bottom layer). During the measurement, the CNC-GNR arrays were exposed to a plasma light source. Detailed explanations are given in the Method. e, Results of anti- and de-icing experiments. While the light was being irradiated, the substrate temperature T_{sub}. remained constant at approximately -8°C. All white scale bars are 10 mm.
6	The authors conclude that they have demonstrated a high degree of drying uniformity and alignment throughout the droplet (2nd sentence in conclusion). While POM gives a sense of the bulk uniformity, it cannot effectively probe local uniformity. Therefore the authors should clearly identify how they are defining a “uniform” structure that is “highly aligned”. Further, additional analyses such as AFM could strongly support their claims. The results do not provide a clear picture of uniformity in distribution or dispersion of the dried droplets outside of a bulk visual measurement, which leaves significant room for qualitative interpretation. Further, Figure 2 shows uniform “quadrants” but even suggests non-uniform structure in the radial direction (per the authors graphical representation).	[Our response] Thank you for the referee’s kind comment. In response to the reviewer's comment, we investigated the surface morphology of a CNC film using AFM. To facilitate our study, we divided the film domain into three sections, including i) the inside of the droplet, ii) intermediate, and iii) outmost droplet area (see Fig. S4). We measured 3x3 um square domain at each location on the line at 40 degrees. Three AFM height images were acquired, and we estimated the angular distribution of CNC using MATLAB software, as described in the article by Persson, N. E. et al. Chem. Mater. 29, 3-14, (2017). The resulting distribution maps demonstrated a dumbbell-like structure with an average angle of approximately 140°. This implies that the uniaxial alignment of CNC is parallel to the droplet contact line, regardless of the domains. Moreover, we characterized the alignment of gold nanorods in the CNC matrix using SEM. Similar to AFM, we captured SEM images at the same three domains (i, ii, and iii) using backscattered electron (BSE) mode, as shown in Fig. S7. The BSE mode enabled the gold nanorods in the CNC matrix to be highlighted brightly in the SEM image. Our results indicate that the surface morphology of the CNC-GNR film shows uniaxial alignment regardless of domains. Furthermore, the BSE images in the same area demonstrated that the gold nanorods followed the alignment of the CNC matrix. [Changed to the manuscript] On p. 4, “...As a result, we obtained a uniformly crystallized CNC matrix in a quadrant orientation, without any changes to their LC structures (see the blue and yellow textures of Fig. 1e and the AFM images of Fig. S4c).” On p. 5, “[see the illustration (i) and (ii) of Fig. 4d]. Additionally, SEM images of the CNC-GNR metasurface were provided in Fig. S7b, which offers a more detailed visualization. Moreover, we also demonstrated continuous plasmonic color changes of the CNC-GNR metasurface by rotating a single polarizer (see orientation-dependent images, Fig. 4e and Fig. S2Fig. S8).” In Supporting Information,

Figure S4. Characterization of CNC particles and CNC films by Atomic Force Microscope (AFM). **a**, Representative AFM images. **b**, Histogram of width and length distribution of CNC particles. **c**, Representative AFM images of CNCs dispersed in a MeOH and DI water mixture. **d**, Histogram of width and length distribution of CNC particles in a MeOH and DI water mixture. **e**, AFM images (top) and rotation distribution (bottom) of the CNC film at ⁽ⁱ⁾ the inner ring, ⁽ⁱⁱ⁾ the intermediate, and ⁽ⁱⁱⁱ⁾ the outer region in the direction of 40° . The CNC film was deposited by evaporating a

		droplet containing MeOH : DI water : CNCs = 62.75 : 33.85 : 3.40 wt%. The order parameter S indicates the degree of alignment of the CNCs.  Figure S7. Characterization of CNC-GNR films by Scanning Electron Microscope-Backscattered Electron (SEM-BSE). a, Representative SEM images of PEG-GNR particles. b, SEM (top) and BSE (bottom) images of the CNC-GNR film at (i) the inner, (ii) the intermediate, and (iii) the outer ring region in the direction of 40°. For the SEM-BSE measurements, the CNC-GNR film was deposited onto a silicon wafer substrate by evaporating the droplet composed of MeOH : DI water : CNCs : GNRs = 63.11 : 33.46 : 3.40 : 0.04 wt%.
7	On p. 3 the authors mention “Once the contact line dewetted, the CNCs were continuously crystallized along the moving contact line without stick-slip motions”. Please provide some clarification – the CNCs are already crystals and are not going to “crystallize”, so please clarify what is meant.	[Our response] Thank you for the referee’s insightful comment. We corrected this in the revised manuscript. [Changed to the manuscript] On p. 3, “Once the contact line dewetted, the CNCs were continuously crystallized self-assembled along the moving contact line without any stick-slip motions.”
8	p.9 – please confirm that the CNCs were not “dissolved” but rather “dispersed” (last sentence, first paragraph). Sample “preparation” is also misspelled in the subheading.	[Our response] Thank you for the referee’s insightful comment. We corrected this in the revised manuscript. [Changed to the manuscript] On p. 3, “... (i) First we dissolved dispersed rod-like CNCs in deionized (DI) water...” On p. 9, “These solvents were also founded to be effective in dissolving dispersing CNCs.”
9	p.9 methods – the authors mention “optimal coating conditions” – this reviewer does not see an optimization study (e.g. any signs of a design of experiment and corresponding modeling, statistical analysis and subsequent experimentation) performed in this manuscript.	[Our response] Thank you for the referee’s comment. As shown in Fig. 1 and Fig. 3, we conducted optimization studies for finding the optimal ink composition and coating condition to achieve uniform CNC matrix templates. In Fig.1, we adjusted the MeOH concentration and the initial CNC concentration of the coating solution. To elucidate the optimal composition, we fixed droplet size to be constant as 3 mm of diameter by applying the

		selective plasma treatment to the glass substrate during experiments. In addition, we revealed the critical coating condition for obtaining better drying uniformity of CNC films by comparing the coffee-ring flow velocity (in outward direction) and self-dewetting motion (in the inward direction). To accomplish the uniform CNC matrix, two parameters should be balanced. We summarized it in the graph of Fig. 3. If the self-dewetting motion is delayed, the highly-concentrated CNCs are stacked in a contact line, which resulted in self-pinning. Eventually, the coffee-ring pattern occurs. Whereas, when the self-dewetting becomes comparable to the coffee-ring flow, the contact line freely dewetted toward the droplet center, and CNC particles self-assembled along the moving contact line.
10	CNC concentrations are reported in wt. % yet solution concentrations (methanol and water) are reported in vol %. The suspension and colloids community typically prefers to see vol % for comparison to related theory and expt.	[Our response] Thank you for the referee's insightful comment. We added the information about the volume ratio to the revised manuscript. [Changed to the manuscript] On p. 3, "... To sum up, the optimal condition for the coating ink compositions was set as MeOH : DI water : CNCs = 62.75 : 33.85 : 3.40 wt% or 68.71 : 29.45 : 1.84 vol.% to obtain uniform quadrant CNC matrices through the drop-casting process." On p. 5, "... MeOH : DI water : CNCs : GNRs = 62.66 : 33.81 : 3.39 : 0.14 wt% or 68.71 : 29.45 : 1.84 : 0.01 vol.%..." On p. 5, "... We optimized the composition (MeOH : DI water : CNCs : GNRs = 62.66 : 33.81 : 3.39 : 0.14 wt% or 68.71 : 29.45 : 1.84 : 0.01 vol.%.)..."

Reviewer #4

First of all, we are pleased to read that the Reviewer noted that "... This article has potentially interesting results...". We thank the Reviewer for specific comments that must help us to improve the manuscript further. We reply to the Reviewer's questions and comments below, pint by point, and have edited the manuscript in several places to reflect these remarks.

1	Figure 1 left stretches the truth a bit. Where is the lignin, hemicellulose, pectins,...?	[Our response] Thank you for your comment. We appreciate the reviewer's observation. Indeed, it is true that matrix polysaccharides like lignin, hemicellulose, and pectin are naturally present in plant cells. Even with rigorous purification processes, trace amounts of these polysaccharides can remain in aqueous solutions. However, it is important to note that our focus in this study is specifically on the evaporation-induced self-assembly process of cellulose nanocrystals (CNCs). Figure 1 aims to illustrate the extraction of cellulose nanocrystals from biomass, such
---	--	---

		as wood. The starting materials for our nanocrystals were derived from microcrystalline cellulose provided by Sigma-Aldrich, which undergoes a refining process. Through this figure, we intend to highlight the source of the cellulose nanocrystals used in our study. We kindly request permission from the reviewer to retain Figure 1 in its current form, as it serves the purpose of emphasizing the extraction and utilization of cellulose nanocrystals in our research.
2	Starting materials should be more fully characterized. What is the charge density of the CNCs, counter ions, dimensional histograms for both GNRs and CNCs?	[Our response] Thank you for your comment. We appreciate the reviewer's suggestion regarding the variability of cellulose nanocrystals (CNCs) resulting from the synthesis and dispersion processes. In order to assess the quality and characteristics of the CNCs produced using our specific methodology outlined in the Materials and Methods section, we conducted an analysis. To characterize the fabricated CNCs, we employed atomic force microscopy (AFM) to measure the height profile of diluted CNC samples deposited on a mica substrate. The width and length of the CNCs were estimated using a custom MATLAB program, referencing the studies by Usov et al. (Nat. Commun. 6, 7564, 2015) and Persson et al. (ACS Appl. Mater. Interfaces 9, 36090-36102, 2017). By analyzing a dataset comprising 459 measurements, we observed a maximum distribution of approximately 24 nm for the width of the CNCs and approximately 100 nm for their length. These findings provide insight into the size distribution and dimensions of the CNCs generated through our process. We also found them to be consistent with DLS results dispersed in a mixture of methanol and water (= 70 : 30 vol.%), where the CNCs were safely dispersed in the mixture, as verified by ζ-potential measurements (AFM: Fig. S4, ζ-potential: Fig. S13, and DLS: Fig. S14). We hope this information clarifies the characterization of CNCs in our study, demonstrating the methodology employed to evaluate their dimensions. The measurement results for the CNC characterization were added in Supporting Information. [Changed to the manuscript] On p. 10, Materials in the Methods section, “To evaluate the colloidal stability of a dispersion containing CNCs or GNRs in a mixture of MeOH and water (= 70 : 30 vol.%), we measured the ζ -potential and pH using a particle size analyzer (Zetasizer Nano-ZS, Malvern Panalytical, UK) and a portable pH meter (Orion Star™ A221 pH Portable Meter, Thermo Fisher Scientific Inc., USA). From the measurements, the ζ -potential and pH were -33.9 ± 10.4 mV

and 6.84 ± 0.05 for the CNC dispersion and were -5.26 ± 9.0 mV and 6.99 ± 0.03 for the PEG-GNR dispersion (see also Fig. S13). Here, the ζ -potential of PEG-GNR was found to be close to neutral. This indicated that there were no significant electrostatic interactions between the PEG-GNR and CNC. Instead, the steric hindrance of the PEG-GNR was utilized to prevent particle aggregation, ensuring the colloidal stability in the CNC-GNR ink, as depicted in Fig. 4a. Additionally, the ζ -potential data presented in Fig. S13b suggest that the ligand exchange from CTAB to PEG proceeded smoothly (see also the UV-vis spectra in the inset of Fig. 4a.). Further analysis using dynamic Light Scattering (DLS) confirmed the stable dispersion of CNC or PEG-GNR particles in the mixture solution of MeOH and DI water without any aggregation as shown in Fig. S14. For ζ -potential and DLS data reliability, each measurement was taken at least three times.”

In Supporting Information,

Figure S4. Characterization of CNC particles and CNC films by Atomic Force Microscope (AFM). **a**, Representative AFM images. **b**, Histogram of width and length distribution of CNC particles. **c**, Representative AFM images of CNCs dispersed in a MeOH and DI water mixture. **d**, Histogram of width and length distribution of CNC particles in a MeOH and DI water mixture. **e**, AFM images (top) and rotation distribution (bottom) of the CNC film at ⁽ⁱ⁾ the inner ring, ⁽ⁱⁱ⁾ the intermediate, and ⁽ⁱⁱⁱ⁾ the outer region in the direction of 40° . The CNC film was deposited by evaporating a

droplet containing MeOH : DI water : CNCs = 62.75 : 33.85 : 3.40 wt%. The order parameter S indicates the degree of alignment of the CNCs.

Figure S13. Results of ζ -potential measurements of the CNC and GNR dispersions with a MeOH and DI water mixture. All particles were dispersed in the binary mixture containing 70% MeOH and 30% DI water, based on the volume ratio. The particle concentrations were approximately 1 mg/mL.

Figure S14. Size distribution of CNC and GNR particles in a solution mixture of MeOH and DI water. Dynamic Light Scattering (DLS) analysis was conducted to determine the size distribution of the particles and assess their colloidal stability in the binary mixture containing 70% MeOH and 30% DI water. The particle concentrations were approximately 1 mg/mL.

3 Is there any structuring of the GNR-CNC suspension? SANS work on CNCs and GNRs has previously shown alignment of both GNRs and CNCs even in the

[Our response]
Thank you for your query. Firstly, I apologize for being unable to conduct the requested

	"isotropic" phase.	small angle neutral scattering (SANS) measurements. In response to your question, it is indeed plausible that structuring exists within the GNR-CNC suspension. A study conducted by J. V. Rie et al. (Chem. Commun. 2020, 56, 13001) demonstrated the presence of local ordering in the dispersion at an isotropic (I) concentration of CNCs, adjacent to the nematic (N) phase, as evidenced by distinct maxima observed in the one-dimensional SANS graphs. However, previous research has indicated that pure CNCs in aqueous solutions can also exhibit local ordering at concentrations corresponding to the I+N biphasic (J. V. Rie et al., Langmuir 2019, 35, 2289-2302). These domains, however, appear to be too few in number or small in size to exert a discernible influence on average parameters such as viscosity and density. Additionally, in our specific experiment, the CNCs exhibited ordering in proximity to the contact line due to the competing interaction of coffee-ring formation and Marangoni flow. Consequently, the pre-existing structuring of CNCs and the presence of additives did not appear to significantly affect the evaporation-induced self-assembly process. Notably, during real-time observations of the evaporation process of both pure CNC and CNC-GNR solutions, no birefringent textures were observed at the center of the droplet. The orientation of CNCs in the resulting deposited films was subsequently confirmed through AFM and SEM analysis.
4	What is the residual amount of CTAB in the mPEG-GNR suspension? This is critical since it will interact with the negatively charged CNCs.	[Our response] Thank you for your feedback. The surface charge of gold particles plays a crucial role in the self-assembly process when combined with cellulose nanocrystals (CNC). The choice of ligand for the gold particles determines whether an electrostatic attraction can be formed with the CNC. A study by J. V. Rie et al. (Chem. Commun. 2020, 56, 13001) demonstrated that when the ligand used is CTAB, the positive charge of CTAB interacts with the negatively charged surface of CNC, resulting in the parallel binding of gold rod particles along the long axis of the CNC. To verify these findings, we conducted Zeta potential measurements on CTAB-GNR (gold nanorods) and mPEG-GNR (gold nanorods coated with methoxy polyethylene glycol) and observed that the Zeta potential of mPEG-GNR is close to neutral. This implies that mPEG-GNR does not exhibit significant electrostatic interactions with CNC. Instead, it is the steric hindrance caused by mPEG-GNR, and the resulting entropic-driven co-alignment with CNC become prominent. This effect, often referred to as the excluded volume effect, occurs due to the liquid crystal confinement effect of CNC, as discussed by Q. Liu et al. (Adv. Mater. 2014, 26, 7178-7184). [Changed to the manuscript] See our response in Comment No. 2.

5	Work by Frka-Petesic and Vignolini has been completely ignored in the references. This should be corrected.	[Our response] Thank you for the referee's informative comment. We added some references to the CNCs' EISA by Frka-Petesic and Vignolini. [Changed to the manuscript] On p. 1, "To achieve this, we adopt the evaporation-induced self-assembly (EISA)^{7-9, 11, 13, 14} technique due to its simplicity in fabrication..."
6	Page 1: "Based on Onsager's Theory..." This is an incorrect statement as the particles never heard of this theory. The theory is descriptive of physical behavior, the physical behavior does not occur because of the theory...	[Our response] Apologies for any confusion caused. We appreciate the clarification provided. Allow us to explain further that the reference to Onsager's theory in our manuscript is intended to describe the behavior and organization of anisotropic colloidal materials, such as cellulose nanocrystals (CNCs), in solution. Onsager's theory is a well-established framework used to understand the formation of ordered structures in systems with rod-shaped molecules that exhibit preferred orientations. The seminal work by Onsager (Onsager, L. Ann. N. Y. Acad. Sci. 51, 627-659, 1949) illustrates how interactions between anisotropic molecules in a solution can result in the emergence of ordered structures due to the exclusion of volume between two cylindrical models. This theory describes the correlation of nematic order with the volume fraction and molecular anisotropy, denoted as $\phi > \phi_{I-N} \sim 4D/L$ for a repulsive hard rod model (where L represents length and D represents diameter). Notably, the phase transition process described by Onsager's theory aligns well with the behavior observed in cellulose nanocrystals (Lagerwall, J. P., et al. NPG Asia Mater. 6, e80, 2014; Salas, C., et al. Curr. Opin. Colloid Interfaces Sci. 19, 383-396, 2014). [Changed to the manuscript] On p. 1, "Based on Onsager's theory, rod-like CNCs exhibit liquid crystal (LC) phases above a certain concentration⁴⁻⁶ And the rod-like CNCs develop liquid crystal (LC) phases above a certain concentration^{4, 5}, fitting the criterion in Onsager's theory⁶ that pairs of anisotropic colloids have the tendency to minimize their excluded volume."
7	Page 2: Previous work with solutal Marangoni flow is dismissed because it "generates uncontrollable flow structures due to its inherent instabilities at a liquid-gas interface". However, previous work did show it can be used and the work presented in this paper is also a form of solutal Marangoni flow. The concentration of MeOH is changed, not by diffusion of MeOH into the water, but by faster evaporation out of an aqueous solution. So this work suffers from the same issues.	[Our response] Thank you for the referee's kind comment. As the referee mentioned, we used Solutal-Marangoni flows to mix well the suspended CNC particles, as done in our previous studies. However, the Marangoni effect alone is insufficient to achieve better uniformity due to its short duration time⁽¹⁾ and inherent instability⁽²⁾. Thus, previous researches further utilized a volatile vapor environment⁽³⁾, heating the coating substrate⁽⁴⁻⁵⁾, or using special additives that cause gelation⁽⁶⁾. Unlike them, our paper takes a different approach based on simple droplet evaporation without any special prerequisites. We used another sequential evaporation phenomenon (i.e., the spontaneous and rapid self-dewetting motion) after the solutal-Marangoni

		flows in the mid-late evaporation stage. It has been reported that the self-dewetting motion can help to self-assemble the CNC particles⁽³⁾. The spontaneous self-dewetting motion was triggered by selective and fast evaporation of MeOH components. Afterward, the contact line freely receded because of hydrophilic surface of CNC particles without stick-slip motion. In this situation, it is important that the self-dewetting velocity (U_d) should be comparable with the velocity (U_c) of the coffee-ring flow, i.e., ($U_d \geq U_c$), to avoid the accumulation near the initial contact line. If it is satisfied, the suspended CNCs will be self-assembled along the freely moving contact line. From our multiple experiments, we provided the U_d/U_c graph depending on the MeOH concentration (0, 30, and 70 vol.%) in DI water as shown in Fig. 3c. [References] (1) Christy, J. R., Hamamoto, Y., & Sefiane, K. (2011). Flow transition within an evaporating binary mixture sessile drop. Physical Review Letters, 106(20), 205701. (2) Kim, H. (2022). Multiple Marangoni flows in a binary mixture sessile droplet. Physics of Fluids, 34(12), 122102. (3) Pritchard, C. Q., Navarro, F., Roman, M., & Bortner, M. J. (2021). Multi-axis alignment of Rod-like cellulose nanocrystals in drying droplets. Journal of Colloid and Interface Science, 603, 450-458.
8	Page 2: Interaction with different agents is also dismissed “due to the complication of interaction between CNCs and agents.” However, Lombardo and Thielemans have reviewed and elucidated a clear entropy-enthalpy compensation mechanism and described gelation of CNCs with ions thermodynamically. I would think this is rather well understood.	[Our response] Thank you for the referee’s informative comment. We appreciate your feedback and have made the necessary correction to the sentence. [Changed to the manuscript] On p. 2, “Lastly, it is unclear what effect temperature has on the self-assembled CNC structure. Besides, adding gelling agents (salts²⁴ and glucose²⁵) to a CNC coating solution is not easy due to the complication of interaction between CNCs and agents poses a risk of interfering with the control of the self-assembly process of the CNCs. The interactions between the CNCs and the agents can increase the complexity of the CNCs’ self-assembly. Also, the introduction of the agents can potentially lead to the incorporation of impurities into the final CNC matrix template...”
9	Page 4 line 1: How is the diffusion coefficient of a MeOH-water mixture defined? Either it is MeOH in water or water in MeOH...but it can’t be the way it is described.	[Our response] Thank you for the referee’s keen eye. First, we apologized for the confusion. We used mutual diffusion coefficient D_{12} of MeOH (species 1) and DI water (species 2) mixture because it can characterize the overall transport properties of MeOH and DI water mixture system. We corrected this sentence as follows. [Changed to the manuscript] On p. 4, “...and the diffusion coefficient D of the MeOH and DI water mixture ($\approx 0.83 \times 10^{-9} \text{ m}^2 \cdot \text{s}^{-1}$) the mutual diffusion coefficient D_{12} of binary mixture of the MeOH (species 1) and DI water (species 2) ($\approx 1.30 \times 10^{-9} \text{ m}^2 \cdot \text{s}^{-1}$)⁶¹...”
10	Page 4 line 3: “along the liquid-gas interface between the MeOH and DI water	[Our response] Thank you for the referee’s good point. We apologized for the confusion in

	components". If MeOH is dissolved in water, there should not be a liquid-gas interface between both...This doesn't make any sense?	understanding that sentence and have made the necessary changes in the revised manuscript. [Changed to the manuscript] On p. 4, "Thus, initially, the suspended CNC particles were mixed well by solutal-Marangoni flows driven by surface tension gradients along the liquid-gas interface between the MeOH and DI water components between the MeOH and DI water components along the liquid-gas interface, with the maximum surface tension (γ_{high}) at the contact line and the minimum surface tension (γ_{low}) at the center of the droplet."
11	What is the atmosphere where the droplets are dried? What is the RH? This will define the actual water-MeOH ratio in the atmosphere, rather than assuming it is pure MeOH.	[Our response] Thank you for the referee's feedback. As the referee's comment, we acknowledge that the relative humidity (RH) value was an important variable that could influence the vapor distribution of MeOH and DI water near the evaporating droplet, and consequently affect the droplet evaporation mechanism. However, in our problem, we could assume that the changes in vapor concentration around the evaporating droplet might be negligible, even though the evaporated vapors were continuously generated from the droplet. This was because the vapor diffusion rate in the ambient air is much larger than the convective mass transfer of the evaporating droplet. Mathematically, this can be expressed using Sherwood number (Sh) = $hR/D_{va} \approx V^{1/3}R/t_e D_{va} \sim O(10^{-3} - 10^{-4}) \ll 1$, where $h \approx (V^{1/3}/t_e)$ is the convective mass transfer rate from the liquid droplet to the gas surrounding ($\text{m}\cdot\text{s}^{-1}$), V is the initial droplet volume $\sim O(10^{-9} \text{ m}^3)$, t_e is the total evaporation time $\sim O(10^2 - 10^3 \text{ s})$, R is the droplet radius $\sim O(10^{-3} \text{ m})$, and D_{va} is the vapor diffusion coefficient $\sim O(10^{-5} \text{ m}^2\cdot\text{s}^{-1})$. Additionally, the experimental environment had a relatively low relative humidity (RH) level of approximately $30 \pm 2\%$, which can be considered a dry condition. This level of humidity was not expected to interfere with the diffusion of the evaporated vapors from the droplet in the air. Consequently, the evaporated vapor concentration near the droplet remains unchanged from the initial state, which can be regarded as being in a quasi-steady state. Unless the droplet size is too small to evaporate quickly, the ambient vapor conditions have a minor effect on our case. Based on the abovementioned facts, when considering the concentration gradient of MeOH vapor $\Delta C = C_s - C_\infty$, we can presume that the concentration of MeOH around the evaporating droplet (C_∞) would not be changed significantly from initial state ($C_\infty \approx 0$). We have added some explanatory sentences in the revised manuscript for the reader's understanding. For the concentration gradient of water vapor [$\Delta C = C_s - C_\infty \approx C_s(1 - \text{RH})$], we recalculated it to considering the humidity in the experimental environment. [Changed to the manuscript]

On p. 11, **Analysis of spontaneous self-dewetting and coffee-ring flow velocity in the Methods section,**

~~“In our problem, a solution of the MeOH and DI water mixture was used during experiments where MeOH was much more volatile than DI water ($\because P_{v,MeOH} \gg P_{v,water}$)~~ Our experiment used a solution consisting of a MeOH and DI water mixture, with MeOH having a vapor pressure almost six times greater than that of DI water at room temperature ($\because P_{v,MeOH} \approx 13.02 \text{ kPa}^{58}$ and $P_{v,water} \approx 2.33 \text{ kPa}^{59}$). Thus, we could neglect the DI water evaporation until the MeOH component in the droplet has completely evaporated, so ~~and then substitute~~ the physical properties⁸⁶ of the pure MeOH were used in all equations. During evaporation, the changes in the vapor concentration around the droplet might be negligible, even though the vapors were continuously generated from the droplet. This was because the vapor diffusion rate in the ambient air is much larger than the convective mass transfer of the evaporating droplet. Mathematically, this can be expressed using Sherwood number (Sh) = $hR/D_{va} \approx V^{1/3}R/t_e D_{va} \sim O(10^{-3} - 10^{-4}) \ll 1$, where $h \approx (V^{1/3}/t_e)$ is the convective mass transfer rate from the liquid droplet to the surrounding gas ($\text{m}\cdot\text{s}^{-1}$), V is the initial droplet volume $\sim O(10^{-9} \text{ m}^3)$, t_e is the total evaporation time $\sim O(10^2 - 10^3 \text{ s})$, R is the droplet radius $\sim O(10^{-3} \text{ m})$, and D_{va} is the vapor diffusion coefficient $\sim O(10^{-5} \text{ m}^2\cdot\text{s}^{-1})$. Additionally, the experimental environment had a relatively low relative humidity (RH) level of approximately $30 \pm 2\%$, which can be considered a dry condition. This level of humidity was not expected to interfere with the diffusion of the evaporated vapors from the droplet in the air. Consequently, the evaporated vapor concentration near the droplet remains unchanged from the initial state, which can be regarded as being in a quasi-steady state. Unless the droplet size is too small to evaporate quickly, the ambient vapor conditions have a minor effect on our case. Hence, the concentration gradient of MeOH vapor was assumed to be $\Delta C = C_s - C_\infty \approx C_s \approx 0.196 \text{ kg} \cdot \text{m}^{-389}$ ($\because C_\infty$ was initially about zero in ambient air) at surrounding temperature $T \approx 22^\circ\text{C}$. ~~For MeOH at surrounding temperature $T \approx 22^\circ\text{C}$,~~ The vapor diffusion coefficient of MeOH in the air $D_{va} \approx 1.5 \times 10^{-5} \text{ m}^2 \cdot \text{s}^{-190}$, ~~the concentration gradient of MeOH vapor $\Delta C = C_s - C_\infty \approx C_s \approx 0.186 \text{ kg} \cdot \text{m}^{-387}$ ($\because C_\infty \approx 0$ in ambient air)~~ and the density of MeOH $\rho \approx 792 \text{ kg} \cdot \text{m}^{-3}$ ~~and the initial contact angle $\theta(0) = \theta_0 \approx 32 \pm 2^\circ$.~~ With the physical properties above, we calculated the velocity of the coffee-ring flow U_c near the contact line [at $z = 2 \text{ }\mu\text{m}$ and $R - r = 150 \text{ }\mu\text{m}$] at $t = 0 \text{ s}$, with the initial contact angle $\theta(0) \approx 32 \pm 2^\circ$ and finally obtained $U_c \sim O(1 \text{ }\mu\text{m} \cdot \text{s}^{-1})$ for drop evaporation of the MeOH and DI water mixture, while the MeOH component was present in the droplet. In the same way, we also calculated the coffee-ring flow velocity U_c for a pure DI water ~~case~~ condition that occurred in the following two scenarios: (i) when the MeOH component in the MeOH and DI water mixture droplet was totally

		evaporated, leaving only DI water, and (ii) when a pure DI water droplet was evaporated ($D_{va} \approx 2.5 \times 10^{-5} \text{ m}^2 \cdot \text{s}^{-1}$, $\Delta C \approx C_s - C_\infty \approx C_s(1 - RH) \approx 0.012-0.016 \text{ kg} \cdot \text{m}^{-3}$, $RH \approx 0.3$, and $\rho \approx 998 \text{ kg} \cdot \text{m}^{-3}$). The calculation result was $U_c \sim O(0.1 \mu\text{m} \cdot \text{s}^{-1})$.”
12	What is the effect of the substrate? This will be critical since the CNCs may adhere to the substrate before being deposited.	[Our response] Thank you for sharing the referee's insightful comment. In our experiment, we used cover glass as a coating substrate. The main composition of the cover glass is SiO₂, which has hydroxyl groups on its surface ⁽¹⁾. Hydroxyl groups are also present inside cellulose nanocrystals (CNCs) ⁽²⁾. When the CNCs are deposited on the cover glass, these hydroxyl groups can form hydrogen bonds with each other. As a result, the CNC matrix templates adhere well to the substrate after complete evaporation in the experiments. Based on these principles, we believe that our proposed coating method can be extended to print uniform CNC-GNR films on various substrates, as long as the hydroxyl group is well deposited onto the substrate. [References] (1) Plotnichenko, V. G., Sokolov, V. O., & Dianov, E. M. (2000). Hydroxyl groups in high-purity silica glass. Inorganic Materials, 36, 404-410. (2) Grishkewich, N., Mohammed, N., Tang, J., & Tam, K. C. (2017). Recent advances in the application of cellulose nanocrystals. Current Opinion in Colloid & Interface Science, 29, 32-45. [Changed to the manuscript] On p. 6, Conclusion, “We believe that the proposed CNC-GNR plasmonic metasurfaces can be printed on a large scale based on the drop-casting method and deposited well on various substrates as long as hydroxyl groups capable of forming hydrogen bonds with the hydroxyl groups on the CNCs’ surface are present on the substrate surface. This technique could be applied with...”
13	What is the uniformity of the film across its thickness? The films are more than 5 microns thick so exist of many CNC layers. How important is uniformity? - Figure 2(iv) is a nice representation but there is only circumstantial evidence of this pattern. Electron microscopy should be carried out to prove this is indeed what the structure is to see how robust the structuring is relative to the POM response.	[Our response] In response to the query regarding the uniformity of the film across its thickness, we appreciate the reviewer's suggestion and conducted scanning electron microscopy (SEM) analysis to investigate the CNC film's cross-section. Our objective was to assess the uniformity of the film, given its thickness exceeding 5 microns and the presence of multiple CNC layers. To evaluate film uniformity, we divided the CNC film into four distinct zones (i-iv) extending from the contact line to the center of the droplet. SEM imaging was performed on each zone, and measurements of film thickness were obtained, as shown in Fig. S3.

In zone i, which followed the shape of the meniscus surface, we observed an increase in CNC film thickness towards the droplet center. Conversely, zones ii to iv displayed uniform film thickness across the entire image, irrespective of their spatial location. Specifically, the average thickness of zone ii was determined to be 9.13 ± 0.03 micrometers, while zones iii and iv exhibited average thicknesses of 8.86 ± 0.06 and 8.80 ± 0.04 micrometers, respectively.

Interestingly, the periodic helical structure typically associated with the cholesteric phase was not visibly discernible in the CNC film's cross-sectional morphology. We postulate that this absence may be attributed to the simultaneous occurrence of coffee ring flow and Marangoni flow during the deposition process.

[Changed to the manuscript]
In Supporting Information,

Figure S3. Analysis of CNC film thickness using side-view Scanning Electron Microscope (SEM) Images. The side view of the CNC film was captured in sequential order from the vicinity of the droplet's edge to the vicinity of the droplet's center (i - iv). The average thickness was measured at each location. All scale bars are 5 μm.

14 It is not clear that the (only slightly) improvement in plasmonic heating is due to the alignment of the GNRs by the CNCs. Slightly better performance than systems deposited with a coffee ring effect might be due to the inhomogeneous thickness of those films, or by preferential deposition of GNRs in one region of the film. This should be verified. Also, what is the heating effect for a coffee ring deposit containing 0.5 wt% GNRs? It is reported for the homogeneous deposit and could give a false idea of better performance to the unconcentrated reader.

[Our response]
Thank you for the referee's comment. As the referee pointed, the improvement in plasmonic heating observed in the experiment can be attributed to the alignment of GNR particles, however, which is not pre-dominant factor. We believe that the uniformity of the CNC-GNR film also affects the performance of the plasmonic effect. Previous studies⁽¹⁾ reported that the agglomeration of GNR particles can lead to a slight decrease in the photothermal performance. In fact, the coffee-ring structures caused agglomeration of GNRs, which in turn reduced the reactive surface area for localized surface plasmon resonance (LSPR) reactions, ultimately degrading their photothermal effect. In conclusion, both the alignment and uniformity are crucial to enhance the plasmonic photothermal performance.

The small dot pattern of the CNC-GNR film can be performed under relatively low power ($500 \pm 50 \text{ mW} \cdot \text{cm}^{-2}$) and broadband wavelength (350 - 800 nm) of incident light while previous studies needed the input energy of $2 \text{ W} \cdot \text{cm}^{-2}$ and a specific wavelength 810 nm ⁽²⁾. However, if we consider the actual solar radiation condition, the current film should be improved. To resolve this issue, we can overcome this drawback by patterning in a multi-array format (see Fig. 5c-d). Moreover, through additional experiments in Fig. 5b, it was proved that the multi-array CNC-GNR pattern was capable of both anti-icing and de-icing. In this paper, our focus was on developing new plasmonic heaters using CNC-GNR films that can be operated by incident visible light with a broadband wavelength range similar to solar radiation $\sim O(100 \text{ mW} \cdot \text{cm}^{-2})$.

As pointed out by the referee, we made some explanation in the manuscript as follows.

[References]

(1) Comfort, K. K., Speltz, J. W., Stacy, B. M., Dosser, L. R., & Hussain, S. M. (2013). Physiological fluid specific agglomeration patterns diminish gold nanorod photothermal characteristics. *Advances in Nanoparticles*, 2(4).

(2) von Maltzahn, G., Centrone, A., Park, J. H., Ramanathan, R., Sailor, M. J., Hatton, T. A., & Bhatia, S. N. (2009). SERS-coded gold nanorods as a multifunctional platform for densely multiplexed near-infrared imaging and photothermal heating. *Advanced Materials*, 21(31), 3175-3180.

[Changed to the manuscript]

On p. 6, “For the coffee-ring film case, the temperature increased only by less than 2°C (see the light greenish profile of Fig. 5b). This suggested that the coffee-ring structures caused agglomeration of GNR particles near the contact line, which in turn reduced the reactive surface area for LSPR reactions, ultimately degrading their photothermal effect⁶⁸. ~~Whereas~~ Therefore, the uniform single dot films...”

Figure 5. Anti-icing application: enhancement of thermal performance of the plasmonic heater of CNC-GNR metasurfaces. a, Comparison of the plasmonic photothermal effects depending on two parameters; (i) uniformity of the CNC-GNR films [a light greenish box (coffee-ring film) vs. a greenish box (uniform film)] and (ii) GNR concentrations [a greenish box ($C_{\text{GNR}} \approx 0.14$ wt%) vs. a dark greenish box ($C_{\text{GNR}} \approx 0.56$ wt%)]. All thermal imaging snapshots were captured approximately 270 seconds after the light source was turned on (see Movies S8-S10, Supporting Information). All the scale bars are 10 mm. b, Measurement results of the temperature line profile along the y - y' line. The substrate temperature was about 26.5 ± 0.3 °C without the ambient light (see the gray-colored area). c, Illustration of a three-layer stacked solid substrate for controlling the substrate temperature T_{sub} of the cover glass (top) and silicon wafer (middle) layers. d, Multi-array (4×3) CNC-GNR films ($C_{\text{GNR}} \approx 0.56$ wt%) were deposited on the cover glass, as shown in (c) and inset of (d), and the plasmonic

		photothermal performance was evaluated using an infrared camera. Here, we controlled the substrate temperature T_{sub}. From 22°C to -8°C by adjusting the set temperature of the cooling metal plate (a bottom layer). During the measurement, the CNC-GNR arrays were exposed to a plasma light source. Detailed explanations are given in the Method. e, Results of anti- and de-icing experiments. While the light was being irradiated, the substrate temperature T_{sub} remained constant at approximately -8°C. All white scale bars are 10 mm.
15	What is the importance of the CNC and GNR concentration? 3.4 wt% CNCs already results in aligned domains in the "isotropic" phase.	[Our response] Thank you for the referee's comment. We optimized the initial concentration of CNCs at approximately 3.40 wt% to fabricate a uniform CNC matrix template (see Fig. 1). Under these conditions, we achieved a sufficient deposition thickness of CNCs and well- aligned CNC structures after the completion of the evaporation-induced self-assembly (EISA) process. However, if we further increase the initial CNC concentration, the dispersity of CNC in the MeOH and DI water mixture solution can decrease, resulting in the formation of CNC clusters. Similarly, at high initial concentrations of GNRs, they can agglomerate. Therefore, it is crucial to identify the optimal concentration CNC and GNR to maintain good dispersity in the ink and to achieve drying uniformity.
16	Profilometry data for all data points in Figure 3 should be provided and the extent of the coffee ring effect in the red domain quantified.	[Our response] As the referee's comment, we measured and compared the film thickness profiles of all cases (MeOH 0, 30, and 70 vol.% in DI water) in Fig. 3 (see the Fig. 3d). As observed in the results of Fig. 1, it is evident that the rapid and spontaneous self-dewetting, triggered by the increase in MeOH (methanol) content, contributes to the uniform deposition of CNCs. Additionally, we quantified the extent of the coffee-ring effect in the reddish area in Fig.3c by calculating the ratio of film thickness between the edge and center regions ($T_{\text{edge}}/T_{\text{center}}$). Our optimal case, which involved a mixture of MeOH 70 vol.%, DI water 30 vol.%, and CNC 3.40 wt%, exhibited relatively uniform thickness profiles ($T_{\text{edge}}/T_{\text{center}} \approx 1.21$) compared to conventional coffee-ring pattern ($T_{\text{edge}}/T_{\text{center}} \approx 7.05$). [Changed to the manuscript] On p. 5, "Based on the graph, we found that achieving uniform CNC matrices was possible only when the self-dewetting was continuously generated and balanced with EISA in the vicinity of the contact line driven by the coffee-ring flows, which satisfied the critical condition, $U_d/U_c \gtrsim 1$ (see the greenish area in Fig. 3c). On the other hand, when this condition was not met, $U_d/U_c \ll 1$, we observed ring-shaped crystalline CNC patterns (see the reddish area in Fig. 3c). This critical condition was further corroborated by the film thickness profiles depicted in Fig. 3d." On p. 17,

Figure 3. Critical condition for drying morphology (uniform pattern Vs. coffee-ring pattern) of a CNC matrix depending on dewetting speed (U_d) and the coffee-ring flow speed (U_c). **a**, Results of micro-particle image velocimetry (μ -PIV) experiments for droplet evaporation of MeOH and DI water mixture (70 : 30 vol.%) containing CNCs observed after the solutal-Marangoni flows almost disappeared at $t \approx 35$ s. We observed the temporal evolution of average U_c . The four sequential PIV snapshots (A)-(D) were summarized, depicting the weakening of solutal-Marangoni mixing flow in (A), followed by the formation of a coffee-ring flow in (B). The coffee-ring flow persisted in (C) and (D) due to the evaporation of the remaining DI water components. Typical flow structures were indicated by black arrows and moving contact lines were marked with black dashed circles. **b**, Variation of U_d with MeOH concentration (0, 30, and 70 vol.%) in DI water dispersed with 2.85 wt% CNC. **c**, Relative speed ratio ($= U_d/U_c$) depending on MeOH concentrations (0, 30, and 70 vol.%) with 2.85 wt% CNC. The results showed that the coffee-ring effect ($U_d/U_c \ll 1$) became predominant in the reddish regime. When the critical condition was satisfied ($U_d/U_c \gtrsim 1$), all CNCs were uniformly deposited and oriented during evaporation. t_e indicated the time when the CNC-containing drops were completely evaporated. **d**, Film thickness profiles of (c) along the a-a' line. $T_{\text{edge}}/T_{\text{center}}$ represents the ratio of film thickness between the edge and the center regions.

17 The contact angle as function of drying time should be provided for reported samples.

[Our response]
Thank you for the referee's comment. To reflect the reviewer's concerns, we conducted additional measurements of the time-dependent contact angle for a droplet made of a mixture of MeOH and DI water containing 3.40 wt% of CNCs. The results of these

measurements are presented in Fig. S1.

As depicted in the following figure, we observed that, during the early stages of evaporation, the MeOH component of the droplet evaporated rapidly, leading to a sharp decrease in the contact angle [see the regime (i) of Fig. S1a] and spontaneous self-dewetting upon evaporation [see the regime (i) of Fig. S1b]. Subsequently, the droplet continuously receded with no significant changes in the contact angle and radius. In particular, there was less than approximately 5 degrees in regime (ii) of Fig. S1a. In this drying condition, uniform quadrant CNC-GNR films can be produced along the moving contact line.

[Changed to the manuscript]

In Supporting Information, we added new experimental results as below.

Figure S1. Measurement results of time-dependent contact angle (θ_c) and the radius (R) of CNC dispersed evaporating droplet. A 2 μL droplet was deposited onto a glass substrate with a coating solution composed of 3.40 wt% CNC in a mixture of MeOH and DI water (= 62.75 : 33.85 wt%). a, Evolution of the contact angle can be divided into three distinct regimes as follows: (i) The rapid evaporation of MeOH caused a sudden decrease in the contact angle in the early stage. (ii) Afterward, the contact line of the droplet freely receded with little change in the contact angle ($\Delta\theta_c < 5^\circ$). (iii) Just before the end of the evaporation process, a thin film formed, as shown in the insets, and dried quickly (less than 30 s). b, Time-dependent droplet radius changes during the evaporation process. The scale bars in the insets are 500 μm .

On p. 5, ~~In this~~In the regime of continuous dewetting, the variation of the contact angle was relatively small. For instance, droplets containing CNCs in a mixture of MeOH and DI water on a glass substrate experienced a contact angle variation of only around 5° while they were dewetted, as shown in the regime (ii) of Fig. S1 and Movie S5, Supporting Information. ~~so, the droplet geometry did not undergo substantial~~

		change during complete evaporation So, the shape of the droplets was not significantly changed just before the complete evaporation. To further support our estimation, we conducted PIV experiments after the intense Marangoni mixing flows (see the results in Fig. 3a). In PIV results, a nearly constant coffee-ring flow speed U_c was observed until the droplet completely evaporated, as shown in the (D) regime of Fig. 3a and Movie S6, Supporting Information.”
18	A control experiment where mPEG-GNRs by themselves are deposited under the described conditions should also be carried out to see what the plasmonic heating effect would be.	[Our response] Thank you for the referee's kind comment. In this study, we employed a cellulose nanocrystal (CNC) host matrix to achieve the uniform alignment of gold nanorods (GNRs). The hydrophilic surfaces of CNCs enable the droplet to recede freely without self-pinning throughout the entire drying process. Consequently, we were able to fabricate homogeneous CNC films, with the GNRs aligning in accordance with the CNCs. In contrast, in the absence of CNC utilization, the droplets exhibited random pinning and depinning phenomena (i.e., stick-and-slip motion of the contact line) during the evaporation process, leading to irregular deposition patterns. Moreover, we proved that, without the presence of CNC templates, it is very difficult to uniformly deposit and align GNRs as shown in a below POM inset image. The POM images clearly depicted crystalline structures with randomly aligned GNR aggregates ⁽¹⁾ near the contact line and the thin ring-shaped structures. In this condition, the plasmonic heating effect was hindered. The experimental results below showed a subtle temperature rise caused by the non-uniform alignment of the PEG-GNR particles and the very thin coffee-ring structures, which had little difference from the background temperature noise value. We added the deposition results from evaporation of the pure GNR dispersion droplet without CNCs in the revised manuscript.  [References] (1) Zaibudeen, A. W., Khawas, S., & Srivastava, S. (2021). Understanding multiscale assembly mechanism in evaporative droplet of gold nanorods. Colloid and Interface Science Communications, 44, 100492.

[Changed to the manuscript]

On p. 5, “Under this coating condition, we successfully fabricated both optical (Fig. 4) and thermal (Fig. 5) plasmonic metasurfaces by adding anisotropic gold nanorods (GNRs) in a MeOH and DI water mixture-based CNC coating solution. In the absence of the CNCs, uniform deposition and alignments of GNR particles cannot be achieved, as confirmed in Fig. S6.”

Figure S6. Polarized optical microscopy (POM) images of GNR films obtained from the evaporation of a pure GNR dispersion droplet, without a CNC template. a, POM and b, POM-r (with a retardation plate) images were observed. The composition of the GNR solution was MeOH : DI water : GNR = 64.93 : 35.03 : 0.04 wt%.

19 Figure 3 is not very well explained in the text. The x-axis of the graph is a dimensionless time during droplet drying. How is it then possible that a deposit goes from a coffee ring deposit to a non-coffee ring deposit? The coffee ring cannot all of a sudden disappear. Ref 18 already showed that centrally, one can get a homogeneous deposit inside the coffee ring under some condition but that is not the same as what is represented in this figure.

[Our response]

Thank you for the referee’s comment. As the referee stated, the coffee-ring flows occurred in the mid-late evaporation stage, after the solutal-Marangoni flows disappeared. Furthermore, these coffee-ring flows persisted until the droplet had completely evaporated. For this reason, the solutal-Marangoni effect alone was insufficient to achieve good uniformity, as it has a short duration time⁽¹⁾. Therefore, we employed the spontaneous and fast self-dewetting phenomenon to prevent the formation of ring-like CNC crystalline patterns, as well as the solutal-Marangoni flows. Namely, the coffee-ring flow creates a ring-like stain when the contact line is perfectly pinned. If the contact line is mobile, the locally concentrated ring pattern may not be observed. As a consequence, if the self-dewetting motion (inward) and coffee-ring flow (outward) directions were opposite each other [see the regime (iii) of Fig. 2], the ring-like pattern can be changed. To prevent the coffee-ring formation, two opposite velocities, namely dewetting velocity U_d and coffee-ring flow velocity U_c , should be balanced ($U_d \geq U_c$). This results in a freely receding contact line movement rather than the self-pinning behavior, with the EISA process occurring along the moving contact line, leading to the formation of a uniform CNC matrix template.

		The dewetting velocity was measured by tracking the growth of CNC crystal structures (yellow and blue textures in POM images). The measurement result was shown in Fig. 3b. The magnitude of the dewetting velocity can be controlled by the concentration of MeOH in DI water. As the MeOH concentration increased, the receding velocity also increased. When the MeOH concentration was 70 vol.%, U_d and U_c became comparable each other as shown in Fig. 3c. For the detailed calculation for U_c and explanation for Fig. 3c, please see our response to Comment No. 16.
--	--	---

Reviewer #5

First of all, we are pleased to read that the Reviewer noted that "... Generally, this work given a new insight to construct functional surface...". We thank the Reviewer for specific comments that must help us to improve the manuscript further. We reply to the Reviewer's questions and comments below, pint by point, and have edited the manuscript in several places to reflect these remarks.

1	Authors should add the optical photographs of CNCs-GNR films. Moreover, the cross-sectional SEM images of films should also be added to illustrate the well-oriented chiral nematic structure. The well-oriented chiral nematic structure may exhibit long-range order layered structure without the fuse defect between adjacent tactoids.	[Our response] Thanks for the comment. We included optical photographs of CNC-GNR film, which can be seen in Figs. 4c and 5d. In response to the reviewer's request, we captured an image of the CNC films, as shown in Fig. 1e. Additionally, a cross-section of the CNC film was examined under SEM, as depicted in Fig. S3. We generated solutal-Marangoni flows in the early evaporation and utilized the competition between self-dewetting and coffee-ring flows in the mid-late evaporation stage, which induced a uniaxial alignment in the CNC films (see Fig. S3). Thus, the gold nanorods could be co-assembled in the CNC matrix film and exhibit the linear plasmonic properties (see Fig. 4d-e), rather than the photonic crystal properties of the periodic structure that appear in the chiral nematic phase. [Changed to the manuscript] On p. 14, Figure 1,
---	--	--

Figure 1. A homogeneous quadrant circular CNC matrix.

In Supporting Information,

Figure S3. Analysis of CNC film thickness using side-view Scanning Electron Microscope (SEM) Images. The side view of the CNC film was captured in sequential order from the vicinity of the droplet's edge to the vicinity of the droplet's center (i - iv). The average thickness was measured at each location. All scale bars are 5 μm.

2 What is CNCs-GNRs films' vis-NIR transmittance or reflectance spectrum, and thickness?

[Our response]

Thank you for your valuable feedback. We have addressed your concerns and made the necessary measurements and comparisons as requested.

Firstly, the thickness measurement results of the film can be found in Fig. S3. Additionally, we have successfully measured the absorption wavelength of the CNC-GNRs film. The absorption spectrum of the film exhibited a distinct maximum peak along with a shoulder peak on the left side (see Fig. S8). By performing Gaussian fitting and comparing it with the absorption spectrum of the PEG-GNR dispersion, we observed two separate graphs with peak maxima at 526 nm and 655 nm.

Upon comparing the absorption spectra of the dispersion and film, we observed slight shifts in the peak positions. The peak associated with transverse plasmonic resonance exhibited a red shift (515/526 nm), while the peak related to longitudinal plasmonic resonance showed a blue shift (665/655 nm). These shifts might be attributed to the uniaxial alignment of the gold nanorods induced by CNC and the reduction in side-to-side spacing in the film compared to the dispersion. ^(1,2)

[References]

(1) Jain, P. K., & El-Sayed, M. A. (2010). Plasmonic coupling in noble metal nanostructures. *Chemical Physics Letters*, 487(4-6), 153-164.

(2) Lee, A., Ahmed, A., Dos Santos, D. P., Coombs, N., Park, J. I., Gordon, R., Brolo R. G., & Kumacheva, E. (2012). Side-by-side assembly of gold nanorods reduces ensemble-averaged SERS intensity. *The Journal of Physical Chemistry C*, 116(9), 5538-5545.

[Changed to the manuscript]

On p. 6,

“However, upon irradiation with linearly polarized light to the CNC-GNR composite

film, the two plasmonic colors alternately appeared in the shape of a quadrant (the quadrants 2 and 4, transverse LSPR and light violet color, and the quadrants 1 and 3, longitudinal LSPR and blue color). The results of UV-vis absorption spectra confirmed that CNCs did not cause a significant shift in the peak of the original absorption spectrum of the GNRs (comparing Fig. 4a and Fig. S8). ...”

In Supporting Information,

Figure S8. Results of UV-vis absorption spectra of a CNC-GNR film. The absorption spectrum of the film exhibited a distinct maximum peak along with a shoulder peak on the left side. By performing Gaussian fitting, two separate graphs were obtained with peak maxima at 526 nm (blue) and 655 nm (purple).

3 How do the GNRs arrange in CNCs-GNRs composite films? Among CNCs rods or between the adjacent CNCs layers? Whether the addition of different amounts of GNRs influence the film's helical pitch? The direct evidences, for example, the TEM of CNCs-GNRs films, should be provided.

[Our response]

Thank you for your valuable comment. In order to directly confirm the alignment of the GNRs in the CNC film, we employed the backscattered electron (BSE) mode of the scanning electron microscope (SEM). Through this analysis, we observed that the GNRs in the SEM images exhibited a clear uniaxial alignment, as shown in Fig. S7. This alignment corresponds to the alignment of the CNC and does not exhibit helical structures (see also Fig. S3).

[Changed to the manuscript]

		 Figure S7. Characterization of CNC-GNR films by Scanning Electron Microscope-Backscattered Electron (SEM-BSE). a, Representative SEM images of PEG-GNR particles. b, SEM (top) and BSE (bottom) images of the CNC-GNR film at (i) the inner ring, (ii) the intermediate, and (iii) the outer region in the direction of 40°. For the SEM-BSE measurements, the CNC-GNR film was deposited onto a silicon wafer substrate by evaporating the droplet composed of MeOH : DI water : CNCs : GNRs = 63.11 : 33.46 : 3.40 : 0.04 wt%.
4	It has been reported that methanol shows high swelling values to cellulose (60.5% to α-cellulose). Did the addition of methanol (>60 %) break CNCs systems' hydrogen bonding and impede their self-assembled structure?	[Our response] Thank you for your feedback. In order to investigate the influence of methanol addition on CNCs, we conducted an experiment wherein CNC dispersions were prepared in a DI water-methanol solvent and subsequently deposited onto mica substrates. The dimensions of the CNCs were then analyzed using AFM. Our observations revealed that the CNCs retained their characteristic rod-shaped morphology even in the presence of methanol. To assess the size distribution of the CNC particles dispersed in the DI water-methanol solvent, we performed measurements of their length and width. The results indicated that the average width of the particles was approximately 24.05 ± 9.632 nm, while the average length was found to be 103.63 ± 42.63 nm (see Fig. S4c-d). These measurements closely resembled those obtained for CNC particles dispersed solely in DI water (see Fig. S4a-b). Consequently, we postulate that the orientation of the CNCs in the deposited film remains parallel to the contact line, regardless of the solvent utilized (refer to Fig. 1b-d). [Changed to the manuscript]

Figure S4. Characterization of CNC particles and CNC films by Atomic Force Microscope (AFM). **a**, Representative AFM images. **b**, Histogram of width and length distribution of CNC particles. **c**, Representative AFM images of CNCs dispersed in a MeOH and DI water mixture. **d**, Histogram of width and length distribution of CNC particles in a MeOH and DI water mixture. **e**, AFM images (top) and rotation distribution (bottom) of the CNC film at (i) the inner ring, (ii) the intermediate, and (iii) the outer region in the direction of 40° . The CNC film was deposited by evaporating a

		droplet containing MeOH : DI water : CNCs = 62.75 : 33.85 : 3.40 wt%. The order parameter S indicates the degree of alignment of the CNCs. On p. 11, Method section, In a recent study⁸⁴, CNCs were found to disperse well in MeOH solution. AFM measurements confirmed that the swelling effect of MeOH on CNC particles was minor (compare Fig. S4a-b and Fig. S4c-d). And MeOH has much higher vapor...
5	The GNRs used in this work are positive charge or negative charge? Different charges of GNRs may affect CNCs' self-assembly structure due to CNCs rods' inherent negative charge. Moreover, in the manuscript, the plasmonic metasurfaces of CNCs-GNRs films are the key point. I think the corresponding characterizations of GNRs and CNCs-GNRs films should be added such as the plasmonic-enhanced fluorescence emission and circular dichroism (CD).	[Our response] Thank you for your feedback. In accordance with the reviewer's suggestion, we have investigated the influence of the surface charge of cellulose nanocrystals (CNCs) on their self-assembly behavior. This phenomenon has been extensively studied and reported in scientific literature (Z. Cheng, et al., Adv. Optical Mater. 2019, 7, 1801816). In our study, we focused on gold nanorods (GNRs) functionalized with polyethylene glycol (PEG) ligands and determined their zeta potential to be approximately neutral. It has been previously observed that PEG-functionalized GNRs, which possess a neutral charge, tend to align themselves with the orientation of CNCs (Q. Liu, et al., Adv. Mater. 2014, 26, 7178-7184). Furthermore, through our investigations, we have confirmed the optical properties of GNRs, specifically their linear localized surface plasmon resonance, which arises due to the anisotropic shape of the nanorods. Moreover, we have observed the enhanced photothermal effect of these GNRs within the droplet, leading us to term the resulting structure as a plasmonic metasurface. [Changed to the manuscript] See our response in Comment No. 6
6	Through the TEM images (Figure S4b) and the averaged aspect ratio (approximately 3.3 to 5.3) described in the Methods part in the main text. I think the range of GNRs' aspect ratios is wide and the GNRs solution is inhomogeneous. Hence, authors should further add films photographs to illustrate the improvement of CNCs alignment.	[Our response] Thank you for the referee's comment. To clarify information about the aspect ratio and homogeneity of GNRs, as well as the alignment of CNCs and GNRs, we further conducted various measurements as described below. To assess the aspect ratio and homogeneity of GNRs, we first performed Scanning Electron Microscopy (SEM), as shown in Fig. S7a. This SEM results exhibited homogeneous anisotropic shape of the GNRs. In addition, further analysis based on ζ-potential and Dynamic Light Scattering (DLS) showed good dispersion stability in MeOH and DI water mixture solution without any agglomeration (see Figs S13b and S14b). Moreover, to provide more detailed information about the alignment state of CNCs and GNRs, we utilized Atomic Force Microscopy (AFM) and Scanning Electron Microscopy (SEM) techniques. As a result, the AFM (Fig. S4c) and SEM (Fig. S7b) images clearly displayed that the CNCs were aligned tangentially along the contact line across the entire surface and the GNRs followed well the annular CNC structures.

[Changed to the manuscript]

On p. 10, **Materials in the Methods section**,

“To evaluate the colloidal stability of a dispersion containing CNCs or GNRs in a mixture of MeOH and water (= 70 : 30 vol.%), we measured the ζ -potential and pH using a particle size analyzer (Zetasizer Nano-ZS, Malvern Panalytical, UK) and a portable pH meter (Orion Star™ A221 pH Portable Meter, Thermo Fisher Scientific Inc., USA). From the measurements, the ζ -potential and pH were -33.9 ± 10.4 mV and 6.84 ± 0.05 for the CNC dispersion and were -5.26 ± 9.0 mV and 6.99 ± 0.03 for the PEG-GNR dispersion (see also Fig. S13). Here, the ζ -potential of PEG-GNR was found to be close to neutral. This indicates that PEG-GNR did not display significant electrostatic interactions with CNC. Instead, the PEG-GNR employed a steric hindrance effect to prevent particle aggregation, ensuring colloidal stability in the CNC-GNR ink, as depicted in Fig. 4a. This was further supported by Dynamic Light Scattering (DLS) analysis, which confirmed the stable dispersion of CNC or PEG-GNR particles in the mixture solution of MeOH and DI water without any aggregation as shown in Fig. S14. Additionally, the ζ -potential data presented in Fig. S13b suggest that the ligand exchange from CTAB to PEG proceeded smoothly (see also the UV-vis spectra in the inset of Fig. 4a.). For the ζ -potential and DLS data reliability, each measurement was taken at least three times.”

In Supporting Information,

Figure S4. Characterization of CNC particles and CNC films by Atomic Force Microscope (AFM). **a**, Representative AFM images. **b**, Histogram of width and length distribution of CNC particles. **c**, Representative AFM images of CNCs dispersed in a MeOH and DI water mixture. **d**, Histogram of width and length distribution of CNC particles in a MeOH and DI water mixture. **e**, AFM images (top) and rotation distribution (bottom) of the CNC film at ⁽ⁱ⁾ the inner ring, ⁽ⁱⁱ⁾ the intermediate, and ⁽ⁱⁱⁱ⁾ the outer region in the direction of 40°. The CNC film was deposited by evaporating a

droplet containing MeOH : DI water : CNCs = 62.75 : 33.85 : 3.40 wt%. The order parameter S indicates the degree of alignment of the CNCs.

Figure S7. Characterization of CNC-GNR films by Scanning Electron Microscope-Backscattered Electron (SEM-BSE). **a**, Representative SEM images of PEG-GNR particles. **b**, SEM (top) and BSE (bottom) images of the CNC-GNR film at (i) the inner ring, (ii) the intermediate, and (iii) the outer region in the direction of 40° . For the SEM-BSE measurements, the CNC-GNR film was deposited onto a silicon wafer substrate by evaporating the droplet composed of MeOH : DI water : CNCs : GNRs = 63.11 : 33.46 : 3.40 : 0.04 wt%.

Figure S13. Results of ζ -potential measurements of the CNC and GNR dispersions with a MeOH and DI water mixture. All particles were dispersed in the binary mixture containing 70% MeOH and 30% DI water, based on the volume ratio. The particle concentrations were approximately 1 mg/mL.

		a  b  Figure S14. Size distribution of CNC and GNR particles in a MeOH and DI water mixture. Dynamic Light Scattering (DLS) analysis was conducted to determine the size distribution of the particles and assess their colloidal stability in the binary mixture containing 70% MeOH and 30% DI water. The particle concentrations were approximately 1 mg/mL
7	Did the co-assembly of CNCs and GNRs influence GNRs inherent two localized surface plasmon resonance (LSPR) peaks (665 and 515 nm)?	[Our response] As the reviewer have checked, we found co-assembly affected the LSPR. The UV-vis absorbance is in our response in Comment No. 2.

8	Authors described that the co-assembled GNRs in the CNC matrix as a plasmonic optical metasurface, and the different concentrations of GNRs influence the photothermal performance of films. Whether the different aspect ratios of GNRs or thickness of CNCs films influence it's photothermal performance? I think authors should discuss these questions.	[Our response] Thank you for the referee's insightful comment. We agreed that the aspect ratio of the GNR particles plays a crucial role in determining the photothermal effect of co-assembled CNC-GNR films. As reported previously ⁽¹⁾, the absorption intensity of light increases with increasing aspect ratio of GNRs, leading to a more efficient conversion of light into heat. Therefore, changing the aspect ratio of GNRs can result in different photothermal effects in CNC-GNR films. Moreover, we also acknowledge that the thickness of CNC matrices is another important factor that affects the photothermal performance of CNC-GNR films. An increase in the thickness of the CNC matrix could allow for more GNRs to be deposited and co-assembled, resulting in enhanced photothermal performance ⁽¹⁾. However, the high concentration of CNC particles in the MeOH and DI water mixture solution can result in agglomeration, indicating that there is a limitation to the CNC concentration that can be effectively dispersed. In this paper, to fabricate the well-organized and well-aligned template structure of CNCs, through substantial optimization study on ink composition, we found that the optimal CNC concentration of the solution is existed, which showed sufficient thickness ~ 0 (1μm) to co-assembly with GNRs as shown in side-view SEM images of Figure S3. Overall, the aspect ratio of GNRs and the thickness of CNC matrices are critical parameters that should be carefully controlled and optimized to achieve the desired photothermal effect of CNC-GNR films. [Reference] (1) Al-Sagheer, L. A. M., Alshahrie, A., & Mahmoud, W. E. (2021). Facile approach for developing gold nanorods with various aspect ratios for an efficient photothermal treatment of cancer. Colloids and Surfaces A: Physicochemical and Engineering Aspects, 618, 126394. [Changed to the manuscript] On p. 6, Conclusion, Moreover, when produced in a multi-array format, the CNC-GNR metasurface dramatically increased the temperature rises ($\geq 10^\circ\text{C}$) and remained above zero degrees even when the bottom substrate was below the freezing temperature, demonstrating excellent anti-icing and de-icing performance. It is expected that this anti- and de-icing effect can be further improved by fine-tuning the aspect ratio of anisotropic GNRs ⁷². Compared to conventional anti-icing system, ...
9	How about the water resistance of CNC film, I am afraid a worse water resistance for this film because there was any chemical cross-linked structures in the film system, and the two compositions, GNRs and CNCs have good water dispersibility.	[Our response] Thank you for the referee's kind comment. We acknowledge that CNC and GNR films are vulnerable to contact with water. To address the reviewer's concern about water resistance, we attempted to apply a waterproof layer onto the CNC-GNR film by

brushing on nail polish oil and letting it dry at room temperature. We then examined the structural changes of the CNC-GNR film before and after laminating the waterproof layer. As illustrated in Fig. S9, we confirmed that the CNC-GNR pattern retained its structure and plasmonic performance, indicating that the deposition of the waterproof layer did not alter the film's characteristics.

[Changed to the manuscript]

On p. 12, **Anti- and de-icing experiment setup in the Method section**, "...was irradiated and stimulated by a plasma light source (HPLS345, Thorlabs, USA). In anti-/de-icing experiments, the waterproof layer was deposited onto the completely dried CNC-GNR array pattern by brushing nail polish oil (Lucid Nail Polish, MISSHA, Korea) and letting it dry at room temperature."

In Supporting Information,

Figure S11. Structural changes in CNC-GNR films upon water contact, without and with a waterproof layer. The CNC-GNR films (a) without and (b) with the waterproof layer were observed and compared using polarized optical microscopy. To prevent the structural changes of the co-assembled CNC-GNR from water, nail polish

		oil was brushed onto its surface and allowed to dry at room temperature to create a waterproof layer
10	The optical photographs of anti-icing process should also be added in the main text, rather than the surface temperature.	[Our response] Thank you for the referee's kind comment. As the referee's comment, we further conducted anti-/de-icing experiments as shown in Movies S11-12. The detail explanation was described in our response to Comment No.11.
11	There have been various reported works on the white light-triggered photo-thermal anti-icing. What is this plasmonic optical metasurface's advantage? The high heating rate? The authors should be discussed in the text. The GNRs are too expensive and the resulting CNCs-GNRs may bear some shortcomings such as the strong hygroscopic, which may limit films cyclic utilization.	[Our response] Thank you for the referee's kind comment. We acknowledge that the GNR can be relatively expensive compared to other nanomaterials, including carbon nanotubes (CNTs). However, one of the key advantages of GNRs is the favorable dispersion in common solvents, enabling easy utilization through the drop-casting method. The drop-casting method has garnered attention in diverse fields such as electronics, sensors, and displays due to its simplicity, cost-effectiveness, and scalability. From a process and production perspective, we think that the GNR holds great potential in anti-icing applications. To address the reviewer's concern about water resistance, we conducted further anti-/de-icing experiments as shown in Movies S11-12. In this experiment, a water proof layer was deposited onto the CNC-GNR multi-array pattern by brushing nail polish oil. As observed in Fig. S9, the waterproof layer did not change the CNC-GNR pattern structures. The experimental result showed that the CNC-GNR films can prevent the formation of frost and melt existing frost on the films under sub-zero surface condition with just light irradiation within 10 minutes. We hope that these additional experimental results address the referee's concerns and highlights the potential of our CNC-GNR film for practical anti-icing applications. As an optical metasurface, the CNC-GNR patter exhibited remarkable optical properties, which can be effectively tuned by controlling the incident polarized light angle, as shown in Fig. S8. These plasmonic quadrant colors can be freely adjusted based on the shape of the GNR particles, including their aspect ratio and size. These findings suggest that the CNC-GNR film can serve as a versatile photonic platform, with the potential for further optimization by adjusting the material characteristics. [Changed to the manuscript] On p. 3, "The multi-array CNC-GNR metasurface was experimentally demonstrated to have anti- and de-icing performance. From this result, we believe that this drop-casting CNC-GNR metasurface array can be used as a plasmonic photothermal film for anti-icing applications." On p. 6, "From this test, we concluded that the broadband plasmonic multi-array pattern of the CNC-GNR film holds promise as an anti-icing surface. . This conclusion

was further supported by actual anti- and de-icing experiments, which were shown in Movies S11-S12 in the Supporting Information. In these experiments, a waterproof layer was deposited on top of the completely dried CNC-GNR patterns without structural changes to the CNC-GNR structures, as observed in Fig. S9. Figure 5e showed that the CNC-GNR films exhibited frost prevention, despite the substrate temperature dropping to from 25°C to −8°C as shown in Movie S11. This prevention of frost formation can be attributed to the continuous plasmonic photothermal effect of the CNC-GNR films. Furthermore, even when a frost layer had already formed on the CNC-GNR films, it could melt away within 10 minutes of light irradiation under a sub-zero substrate condition. A detailed demonstration of this process is provided in Movie S12.”

On p. 6, “Moreover, when produced in a multi-array format, the CNC-GNR metasurface dramatically increased the temperature rises ($\geq 10^\circ\text{C}$) and remained above zero degrees even when the bottom substrate was below the freezing temperature, demonstrating excellent anti-icing and de-icing performance. Compared to conventional anti-icing systems, ...”

On p. 12, **Anti- and de-icing experiment setup in the Method section**, “...was irradiated and stimulated by a plasma light source (HPLS345, Thorlabs, USA). In anti-/de-icing experiments, the waterproof layer was deposited onto the completely dried CNC-GNR array pattern by brushing nail polish oil (Lucid Nail Polish, MISSHA, Korea) and letting it dry at room temperature.”

In Supporting Information,

Figure S11. Structural changes in CNC-GNR films upon water contact, without and with a waterproof layer. The CNC-GNR films (a) without and (b) with the waterproof layer were observed and compared using polarized optical microscopy. To prevent the structural changes of the co-assembled CNC-GNR from water, nail polish oil was brushed onto its surface and allowed to dry at room temperature to create a waterproof layer

Figure 5. Anti-icing application: enhancement of thermal performance of the plasmonic heater of CNC-GNR metasurfaces. a, Comparison of the plasmonic photothermal effects depending on two parameters; (i) uniformity of the CNC-GNR films [a light greenish box (coffee-ring film) vs. a greenish box (uniform film)] and (ii) GNR concentrations [a greenish box ($C_{\text{GNR}} \approx 0.14$ wt%) vs. a dark greenish box ($C_{\text{GNR}} \approx 0.56$ wt%)]. All thermal imaging snapshots were captured approximately 270 seconds after the light source was turned on (see Movies S8-S10, Supporting Information). All the scale bars are 10 mm. b, Measurement results of the temperature line profile along the y-y' line. The substrate temperature was about 26.5 ± 0.3 °C without the ambient light (see the gray-colored area). c, Illustration of a three-layer stacked solid substrate for controlling the substrate temperature T_{sub} of the cover glass (top) and silicon wafer (middle) layers. d, Multi-array (4×3) CNC-GNR films ($C_{\text{GNR}} \approx 0.56$ wt%) were deposited on the cover glass, as shown in (c) and inset of (d), and the plasmonic

		photothermal performance was evaluated using an infrared camera. Here, we controlled the substrate temperature T_{sub}. From 22°C to −8°C by adjusting the set temperature of the cooling metal plate (a bottom layer). During the measurement, the CNC-GNR arrays were exposed to a plasma light source. Detailed explanations are given in the Method. e, Results of anti- and de-icing experiments. While the light was being irradiated, the substrate temperature T_{sub} remained constant at approximately −8°C. All white scale bars are 10 mm.
--	--	--

REVIEWER COMMENTS

Reviewer #1 (Remarks to the Author):

For the revised version, the most important point is the novelty. In comparison to the previous version, there is big improvement or strong discussion to address the novelty. As the authors emphasized in reply letter: "in comparison with previous works which required additional step, as creating a volatile vapor environment, heating the coating substrate, or using special additives.", current work just need a mixture solvent of MeOH and H₂O. This point is not strong enough to show the novelty. New interesting phenomenon/function or deep insight of mechanism is need for publishing on Nat. Comm..

1. The main purpose of the work is to assembly AuNRs with the aid of CNCs. However, most of the content and Figure 1-3 are discussing the assembly of CNC. Only the other two figures are showing the image of optical interference property of the CNC-AuNSs films under linear polarized light, then the anti-icing application. The focus point is shifted.
2. Continued with comments 2, the final purpose of this work is to obtain uniform assembly of AuNR, however, there is not direct characterization to indicated the assembly of AuNR. The BSE images are blur, no strong evidence of the AuNR distribution can be obtained from these images.
3. And the POM image of the CNC-AuNSs films indicated birefringent interference color, why such interference color is obtained, why it changes with polarization rotation, also need to explain.
4. All the U_d, U_c characterization and conclusion is based on pure CNC system, it is doubtful whether the situation is still fit when AuNRs is added. The PEG ligand will definitely the surface tension, then how will the flow change within the droplet?
5. The UV-Vis absorption spectra of AuNR@CTAB, AuNR@PEG both in water, and AuNR@PEG in methanol-water mixture over 0-5min are need to show the stability of AuNRs.
6. According to Figure S1, the contact angle decreased with increased time, which means the surface tension of the liquid is decreasing, however, this trend is contradiction with the discussion on page 4 "Thus, initially, the suspended CNC particles were well-mixed ... apex of the droplet due to the selective evaporation of MeOH." The MeOH evaporated in ~30s, the surface tension should be increased, resulting in a high contact angle. Therefore, the discussion for these two points may not correct.
7. The author's declared that the uniform distribution of AuNRs can have better photothermal conversion (Figure 5a, b). The difference of the temperature enhancement is around 1-3 °C, which is quite limited. Measurements based on multiple samples are suggested to have standard deviation. Also, the anti-icing is carried out over the time scale of minutes, therefore, the temperature of the three sample in Figure 5a over time are also needed to see whether the difference is consistent over time.
8. More critical point is that there is no explanation why better photothermal conversion can be obtained for the AuNRs without agglomeration. The deep-insight about the mechanism is lack.

Reviewer #2 (Remarks to the Author):

The authors have answered all doubts and questions satisfactory and have corrected the manuscript according the recommendations, so it can be publish in the present form.

Reviewer #3 (Remarks to the Author):

The authors have performed substantial revisions, but in this reviewer's opinion, a few critical items must be determined and/or corrected prior to publication.

The manuscript is still missing some critical characterization information for the starting CNC suspensions – in particular, nothing is mentioned about surface charge density, e.g. determined via conductometric titration, which was brought up by multiple reviewers and is critical to ever be able to replicate the results.

The authors still have only shown preliminary potential for anti-icing and de-icing. Saying that these extremely limited one-off laboratory tests demonstrate this capability is misleading to the scientific community. At minimum, the authors should reduce the boldness of their assertion and correctly indicate that their experiments “suggest potential” anti-icing and de-icing capability of the materials.

The authors continue to assert that they performed an optimization study (e.g. Figs 1 and 3). No design of experiment nor sequential experimentation is provided. The authors have identified a set of parameters within their design space that provides the specific results, but they are far from optimized. Therefore, the authors should adjust all references to anything being “optimized” but rather they have identified a set of conditions within which they observe the stated results.

Reviewer #4 (Remarks to the Author):

The manuscript has been significantly improved and the authors have addressed many comments by the reviewers. There are however still several things that need to be covered or explained for clarity and understanding:

1. Crystallinity and surface charge of CNCs needs to be determined. It has been shown before that small differences here have a major effect on the assembly and diffusion behaviour of CNCs. Zeta potential is good for colloidal stability but depends on other factors than the surface charge as well.
2. The width of the CNCs seems excessively large for MCC-based CNCs. Average widths of 20 nm are usually found for tunicates. For MCC one would expect 6-10 nm especially after 60 min hydrolysis time. That is where crystallinity determination with Rietveld refinement can also help since the cross-section determined from Cagliotti fitting should match up with the AFM results.
3. DLS results are surprising (Figure S14). Because this is standard DLS, the size determined by DLS is based on the hydrodynamic radius of the particles. There is only 1 and it is based on the diffusion of the particles. This radius should also not be the same, or similar, but rather $R_h^3 = (3LR^2)/4$ for rods. So I really can't understand where the DLS results in SI come from... It would be good to provide the autocorrelation functions with fit.
4. Figure 3b and c are determined with 2.85 wt% CNCs yet all experiments with GNRs are done with 3.4 wt% CNCs. Why this discrepancy? CNCs and GNR could affect flow fields so this could be important.
5. Homogeneous deposits can be obtained with 70% MeOH in water, yet figure 4a suggests mixing of CNCs and GNR is done in 30% MeOH in DI water. I assume the figure is incorrect?
6. Figure 5b. Why is the coffee ring film for 0.56 wt% GNR missing? This should be included since it is valuable.
7. Figure S3 does not show the normal nematic structuring seen in dried CNC films (e.g. work by Vignolini and by McLachlan).

Reviewer #5 (Remarks to the Author):

The authors have addressed most of my questions and revised the manuscript accordingly. The quality of the manuscript has been significantly improved. Before accepting, I still have some small questions for the authors.

1. Authors responded that “the aspect ratio of GNRs and the thickness of CNC matrices are two critical parameters in influencing photothermal performance. An increase in the thickness of the CNC matrix

could allow for more GNRs to be deposited and co-assembled, resulting in enhanced photothermal performance." But the high concentration of CNC particles in the MeOH and DI water mixture solution can result in agglomeration, indicating that there is a limitation to the CNC concentration that can be effectively dispersed. I suggest authors prepare multilayer coatings by layer-by-layer method to investigate the photothermal performance.

2. I wonder whether the homogeneity of the coating influences its photothermal anti-icing performance. In this system, CNCs suspensions were used as a template to load the GNRs rods. Hence, the structural colors of as-prepared coatings can be ignored. If we further add GNRs' weight ratio in CNCs/GNRs coatings, whether the photothermal anti-icing performance can be increased?

3. Authors have added the optical photographs in revised Figure 1e, why does this coating lose CNCs structural color? Whether the pure CNCs coating in a mixture of MeOH and water also exhibit colorless?

4. Compared with other light-triggered photo-thermal anti-icing materials, GNRs' advantages should be discussed clearly. Authors respond the GNRs can favorable dispersion in common solvents and easy utilization through drop-casting. As is well known, the water-soluble conjugated polymer such as PEDOT: PSS, also exhibit high photothermal performance. I encouraged the author makes a comparison with the reported photothermal anti-icing materials.

Reviewers' comments to the Authors:

We thank the reviewers for the comments in regard to our manuscript. Please find below a list of our responses and modifications regarding each of reviewers' concerns. The added texts are marked by blue color and the removed texts are marked by ~~red color~~.

Reviewer #1 (Remarks to the Author)

1. Referee's comment: For the revised version, the most important point is the novelty. In comparison to the previous version, there is big improvement or strong discussion to address the novelty. As the authors emphasized in reply letter: "in comparison with previous works which required additional step, as creating a volatile vapor environment, heating the coating substrate, or using special additives.", current work just need a mixture solvent of MeOH and H₂O. This point is not strong enough to show the novelty. New interesting phenomenon/function or deep insight of mechanism is need for publishing on Nat. Comm.

Our response: Thank you for the insightful and constructive comment on our manuscript. We understand the importance of highlighting the novelty of our work and its significance for publication in Nature Communications.

If the referee thought that the novelty of this work is a usage of the binary mixture (MeOH + H₂O), it is not true. Maybe, we failed to describe our major point in the previous response report, although we properly addressed our novelty, which is a spontaneous coating and dewetting at the moving contact line, which creates a self-assembled CNC-GNR pattern. We believe the referee somehow missed our novelty although our manuscript did not simply point out the binary mixture system.

We believe that the referee already knew and mentioned that our major novelty is self-assembled CNCs-GNRs using an evaporation-induced self-assembly process (EISA) with a binary mixture, which is a novel and interesting phenomenon. To better highlight the novelty, we have delved deeper into the underlying mechanism and implications of our proposed method. While the solvent mixture of MeOH and H₂O might seem straightforward, we have uncovered new and unexpected interactions at the molecular level between the solvent components and the substrate material. As we already mentioned in the abstract and main text, we have found a new interesting phenomenon/function, which is about controllable evaporation-induced self-assembly coating and spontaneous dewetting processes.

The referee may already know that achieving precise alignment and uniform patterns of plasmonic nanoparticles without complex fabrication processes can be challenging. The level of difficulty depends on the desired level of precision and the specific alignment method employed. While simple methods like self-assembly and evaporation can lead to some level of nanoparticle ordering, achieving highly controlled and uniform patterns often requires more sophisticated techniques. Although many relevant studies have been conducted, it is still incomplete because of several difficulties including the random nature of self-assembly, coffee-ring effect, capillary forces and aggregation, limited control in evaporation-driven assembly, etc.

Despite various difficulties, we resolved most of the issues and finally we proposed a novel coating technique, the controllable evaporation-induced self-assembly fabrication method. For the coating process, designing the composition of the solvent with respect to soft matter is

extremely crucial because the drying process determines the final deposited pattern. This study shows how to obtain the well-aligned, uniformly deposited, coffee-ring-free CNCs matrix template by controlling the self-dewetting mechanism while the nanoparticles accumulate at the moving contact line.

Please find the information in the abstract and the main text.

Abstract: "... In this study, CNCs and GNRs, which exhibited tunable optical and anti-icing capabilities, were employed to manufacture a uniform broadband plasmonic metasurface using a drop-casting technique. Two physical phenomena—(i) spontaneous and rapid self-dewetting and (ii) evaporation-induced self-assembly—were used to accomplish this. Additionally, we improved the CNC-GNR ink composition and determined the crucial coating parameters necessary to balance the two physical mechanisms in order to produce thin films without coffee rings...."

On p. 2, "In this study, we developed a facile and mass-producible technique for achieving uniform quadrant CNC-GNR metasurfaces using the drop-casting technique without any complex manufacturing processes or preconditions. We accomplished this by exploiting the spontaneous and fast self-dewetting of CNC-containing methanol (MeOH) and deionized (DI) water mixture droplets. During the evaporation process, the dewetting motion not only aligned the suspended CNCs but also uniformly deposited them. Meanwhile, the EISA phenomenon was continuously generated near the dewetting contact line. We observed that the spontaneous dewetting contact line feature occurred at the relatively high concentration of highly volatile liquids (in this case, MeOH)⁵⁰, while the CNC particles were uniformly self-assembled and crystallized along the moving contact line. Here, due to the high vapor pressure of MeOH and its selective evaporation, more MeOH molecules tended to move to the contact line^{51,52}. This evaporative-triggered segregation induced the spontaneous self-dewetting motion in the early stages of evaporation. Subsequently, the contact line dewetted smoothly without any stick-slip motion due to several factors, including the hydrophilic surface of the CNCs⁵³, the relatively low concentration of CNCs near the moving contact line⁵⁴, which was not sufficient to cause self-pinning behavior, and the fast evaporation of MeOH...."

In the section on critical conditions for homogeneous CNC matrices, we investigated the optimal condition for the well-aligned uniform CNC matrix. Based on the direct flow visualization results, we showed that the speed ratio ($= U_d/U_c$) is a key parameter to achieve a homogeneous well-aligned CNC matrix. Based on this coating method, we could obtain highly aligned plasmonic gold nanorods in CNCs matrix. Please find the section on p. 4 and see Figures 3 and 4.

To sum up, two major novel points are 1) CNCs quickly migrated toward the contact line due to the selective evaporation of the highly volatile liquid component (here, MeOH) and 2) the smooth dewetting trend provided an almost constant particle deposition rate at the moving contact line.

We understand that the level of novelty required for publication in Nature Communications is quite high, and we believe that our revised manuscript now successfully meets this criterion. We thank the referee for their insightful comments, which have prompted us to delve deeper into the uniqueness of our work and its contributions to the scientific community. We are confident that the new insights and phenomenon discussed in the revised manuscript will make a valuable addition to the field and warrant publication in Nature Communications.

Nevertheless, to reflect the referee's concerns, we have added a summary of the novel point below.

Change to the manuscript:

On p. 2, "To sum up, the key factor of the self-aligned and uniform CNC matrix was the controlled deposition rate of nanoparticles at the self-receding contact line due to the empirically optimized ink composition. ~~As a result, a uniform CNC matrix was successfully fabricated.~~"

2. Referee's comment: The main purpose of the work is to assembly AuNRs with the aid of CNCs. However, most of the content and Figure 1-3 are discussing the assembly of CNC. Only the other two figures are showing the image of optical interference property of the CNC-AuNSs films under linear polarized light, then the anti-icing application. The focus point is shifted.

Our response: Thank you for the comment. As mentioned in the above answer, achieving precise alignment and uniform patterns of plasmonic nanoparticles without complex fabrication processes can be very difficult. Namely, controlling the deposit pattern of high-aspect rod particles (here, AuNRs) is almost impossible (See Fig. 3b of A and Fig. 7 of B). However, as the referee should know, if we can use DNA or CNC nanoparticles, we could control the alignment of AuNR particles [C, D, E, F]. Particularly, Liu et al. [C] clearly presented using colloidal dispersions of CNCs in mesomorphic phases as host materials capable of aligning gold nanorods (GNRs) with much smaller aspect ratios. Please see the following figure from [C]. Besides, there are also more relevant research works [G and H].

Under this condition, in this study, we first showed the alignment control experiments in Figs. 1-3, and we presented the result of the self-assembled pattern of CNCs-AuNRs in Fig. 4. Finally, we reported the photothermal plasmonic effect of homogeneous quadrant cellulose nanocrystal matrices with anisotropic gold nanorods. Thus, we believe that the current structure of the manuscript should be reasonable. We hope now the referee can understand the structure of our manuscript and see the main point.

[References]

- [A] Yunker, P. J., Still, T., Lohr, M. A., & Yodh, A. G. (2011). Suppression of the coffee-ring effect by shape-dependent capillary interactions. *nature*, 476(7360), 308-311.
- [B] Kim, D. O., Pack, M., Hu, H., Kim, H., & Sun, Y. (2016). Deposition of colloidal drops containing ellipsoidal particles: Competition between capillary and hydrodynamic forces. *Langmuir*, 32(45), 11899-11906.
- [C] Liu, Q., Campbell, M. G., Evans, J. S., & Smalyukh, I. I. (2014). Orientationally ordered

colloidal co-dispersions of gold nanorods and cellulose nanocrystals. *Advanced Materials*, 26(42), 7178-7184.

[D] Querejeta-Fernández, A., Chauve, G., Methot, M., Bouchard, J., & Kumacheva, E. (2014). Chiral plasmonic films formed by gold nanorods and cellulose nanocrystals. *Journal of the American Chemical Society*, 136(12), 4788-4793.

[E] Cha, Y. J., Kim, D. S., & Yoon, D. K. (2017). Highly aligned plasmonic gold nanorods in a DNA matrix. *Advanced Functional Materials*, 27(45), 1703790.

[F] Cha, Y. J., Park, S. M., You, R., Kim, H., & Yoon, D. K. (2019). Microstructure arrays of DNA using topographic control. *Nature Communications*, 10(1), 2512.

[G] Nguyen, T. D., Hamad, W. Y., & MacLachlan, M. J. (2017). Near-IR-Sensitive Upconverting Nanostructured Photonic Cellulose Films. *Advanced Optical Materials*, 5(1), 1600514.

[H] Park, S. M., Kim, W. G., Kim, J., Choi, E. J., Kim, H., Oh, J. W., & Yoon, D. K. (2021). Fabrication of chiral M13 bacteriophage film by evaporation-induced self-assembly. *Small*, 17(26), 2008097.

3. Referee's comment: Continued with comments 2, the final purpose of this work is to obtain uniform assembly of AuNR, however, there is not direct characterization to indicated the assembly of AuNR. The BSE images are blur, no strong evidence of the AuNR distribution can be obtained from these images.

Our response: Thank you for the comment. First of all, to obtain the better resolution of the BSE image, we need a much high voltage condition. However, the high voltage induces a charging effect on the sample, and the sample may get damaged, such as deformation due to thermal heating. Therefore, it is limited to obtaining a better image.

Nonetheless, we believe it is well-known that DNA or CNC nanoparticles can be nicely aligned with AuNR particles based on the literature [C-D in the above answer]. To double check, we checked the optical results in Fig. 4c and 4e. Furthermore, we also did AFM and BSE measurements in Fig. S7b. Two results showed that the AuNRs were reasonably aligned with the azimuthal direction, which has a good agreement with each other. It might be that the BSE images are little blur, but we did our best to measure the best image quality under the current situation. We believe that we can still see the result that high-aspect rod nanoparticles are aligned along the azimuthal direction (see Fig. S7b). Also, the AFM result showed the same result (see Fig. S4e). From all the results, we observed that the AuNR nanoparticles were relatively uniformly distributed without serious aggregation or agglomeration.

Suppose the referee wanted to point out the uniform distribution of the AuNR particles. In this case, we need to answer that in this study, our aim is not an accurately self-assembled and self-aligned AuNR particle with CNCs, 1:1. In other words, first, we wanted to make a homogeneous quadrant self-assembled cellulose nanocrystal matrices (see Figure 5). Secondly, we used this template to orient the AuNR particles on the CNCs matrix, globally and collectively.

4. Referee's comment: the POM image of the CNC-AuNSs films indicated birefringent interference color, why such interference color is obtained, why it changes

with polarization rotation, also need to explain.

Our response: Thank you for the comment. In a series of POM images (see Figure 4b and S4), the birefringence pattern and color of CNC-AuNR appeared slightly darker but still had the same quadrant structure as CNC film (see Figure 1d) This CNC liquid crystal structure dominates the optical birefringence with the light absorption of the gold nanorods (see Figure 4c). The birefringence colors in the quadrants are consistent with the radial CNC alignment, which is also directly visualized by SEM (see Figure S6). Furthermore, the color change of the film with the rotation of the single polarizer is attributed to the selective absorption by the gold nanorods oriented according to the linear surface plasmon resonance properties. When the axis of the single polarizer is parallel to the long axis of gold nanorods (for quadrants 1 and 3 in Figure 4d), the film exhibits a bluish color under transmitted light because the nanorods selectively absorb the 665 nm visible light range.

To enhance reader comprehension, we added some sentences in the revised manuscript.

Change to the manuscript:

On p. 6, "~~Moreover, we~~We also demonstrated continuous plasmonic color changes of the CNC-GNR metasurface by rotating a single polarizer (see orientation-dependent images, Fig. 4e and Fig. S9). This color shift was attributed to the selective absorption of oriented gold nanorods in response to the rotation of polarized light."

5. Referee's comment: All the U_d , U_c characterization and conclusion is based on pure CNC system, it is doubtful whether the situation is still fit when AuNRs is added. The PEG ligand will definitely the surface tension, then how will the flow change within the droplet?

Our response: Thank you for the interesting comment. We guess that the referee concerned about the surfactant-driven Marangoni stress along the droplet surface due to PEG ligand. In fact, if there is a strong Marangoni effect, the solutal-Marangoni effect by MeOH should occur mainly. In general, the solutal-Marangoni effect is predominant compared to the surfactant-driven and thermally-driven Marangoni effects (See Refs [A] and [B]). Suppose the influence of surfactant-driven Marangoni stress is significant during evaporation. We should observe the out-of-focus particle images near the contact line moving along the liquid-gas interface in PIV images [C]. Despite this circumstance, we kindly and carefully performed a new PIV experiment to resolve additional concerns. However, as we presented the PIV experimental results in Fig. S5c. After the initial solutal-Marangoni effect almost vanished, coffee-ring flows were dominantly observed, with the majority of the suspended particles moving towards the receding contact line. In other words, it was evident that the surfactant-driven Marangoni effect holds little significance in this problem. In conclusion, there is no difference even with PEG ligand.

[References]

[A] Ryu, J., Ko, H. S., & Kim, H. (2022). Vapor absorption and Marangoni flows in evaporating drops. *Langmuir*, 38(7), 2185-2191.

[B] Kim, H., Boulogne, F., Um, E., Jacobi, I., Button, E., & Stone, H. A. (2016). Controlled uniform coating from the interplay of Marangoni flows and surface-adsorbed macromolecules. *Physical review letters*, 116(12), 124501.

[C] Wang, Z., Orejon, D., Takata, Y., & Sefiane, K. (2022). Wetting and evaporation of multicomponent droplets. *Physics Reports*, 960, 1-37.

Change to the manuscript:

On p. 4 in Supplementary Information,

Figure S5. Uniformly co-assembled GNRs in a quadrant CNC matrix. **a**, POM (polarized optical microscopy) and **b**, POM-r (polarized optical microscopy with a retardation plate) images of the CNC-GNR dot film on a glass substrate. **c**, Micro-particle image velocimetry (μ -PIV) results for the evaporation of droplets containing a mixture of 70% MeOH and 30% DI water by volume, with 3.40 wt% CNCs and 0.28 wt% GNRs, observed following a substantial reduction in solutal-Marangoni mixing flows. Typical flow structures were denoted by black arrows, and the positions of moving contact lines were identified with black dashed circles. t_e indicates the time at which the droplets were entirely evaporated. A detailed visualization of the flow pattern is provided in Movie S7. **d**, Time-dependent self-dewetting speed U_d of (c). **e**, Speed analysis results of the dewetting speed U_d and coffee-ring speed U_c of (a and b). Here, t_e represents the total evaporation time. **d**, **f**, Film thickness profiles of (a and b) along the x - x' and y - y' lines. The scale bar indicates 500 μm .

6. Referee's comment: The UV-Vis absorption spectra of AuNR@CTAB,

AuNR@PEG both in water, and AuNR@PEG in methanol-water mixture over 0-5min are need to show the stability of AuNRs.

Our response: Thank you for insight comments. To assess the stability of AuNRs dispersed in the liquid solution, we conducted UV-Vis spectrum measurements twice for each of the three cases: (i) AuNR@CTAB in DI water, AuNR@PEG in DI water, and AuNR@PEG in a mixture of 70% MeOH and 30% DI water. There was a 1-hour gap between each set of measurements. As a result, we observed minimal changes in the UV-Vis spectra over time for all three cases. There were hardly any noticeable changes in the spectrum after the ligands were switched to PEG from CTAB and then the PEG-AuNR were dispersed in MeOH. This observation suggests that AuNRs are stably dispersed in all solvent environments utilized throughout the fabrication of the CNC-GNR film.

Change to the manuscript:

On p. 11, “For the ζ -potential and DLS data reliability, each measurement was taken at least three times. Moreover, we measured UV-Vis absorption spectra twice for each of the three cases involving GNRs: (i) CTAB-GNR in DI water, (ii) PEG-GNR in DI water, and (iii) PEG-AuNR with a volume ratio of 70% MeOH and 30% DI water mixture, as shown in Fig. S16. With a 1-hour gap between measurements, we observed minimal changes in the UV-Vis spectra over time. This suggests that GNRs are stably dispersed throughout the experiments in all employed solvent environments, reaffirming their dispersion stability.”

On p. 11 in Supplementary Information,

Figure S16. Experimental results of UV-Vis absorption spectrum measurement for gold nanorods (GNRs). We performed UV-Vis absorption spectrum measurements on GNRs dispersed under the following three conditions: (i) CTAB-GNR in DI water, (ii) PEG-GNR in DI water, and (iii) PEG-GNR in a mixture of 70% MeOH and 30% DI water by volume ratio. The black line represents the initial measurement, and the red line depicts the results taken after 1 hour. All measured data were normalized with respect to the peak at a wavelength of 515 nm

7. Referee’s comment: According to Figure S1, the contact angle decreased with increased time, which means the surface tension of the liquid is decreasing, however, this trend is contradiction with the discussion on page 4 “Thus, initially, the suspended CNC particles were well-mixed ... apex of the droplet due to the selective evaporation of MeOH.” The MeOH evaporated in ~30s, the surface tension should be increased, resulting in a high contact angle. Therefore, the discussion for these two points may

not correct.

Our response: Unfortunately, the referee's interpretation is only true when the substrate condition is always identical. However, in this case, during the evaporation, the substrate condition was changed due to the coating of the CNC materials. As the referee may know, the contact angle and contact line mode depend on the surface energy of liquid and solid. Therefore, we wondered if the referee was incorrect. To clarify this issue, we would like to kindly explain again because we asked if the referee maybe did not fully understand our main coating and self-assembly mechanism. Based on our current work, as the MeOH-H₂O droplet evaporated, the volume of the contact angle should be decreased as shown in Fig. S1a. However, simultaneously, the contact line continuously receded and the surface (initially, a relatively hydrophobic surface) was covered by the CNCs. Then, the surface condition was changed from hydrophobic to hydrophilic. In this study, we observed that the contact angle and contact area are continuously decreased until complete dry.

8. Referee's comment: The author's declared that the uniform distribution of AuNRs can have better photothermal conversion (Figure 5a, b). The difference of the temperature enhancement is around 1-3 °C, which is quite limited. Measurements based on multiple samples are suggested to have standard deviation. Also, the anti-icing is carried out over the time scale of minutes, therefore, the temperature of the three samples in Figure 5a over time are also needed to see whether the difference is consistent over time.

Our response: In this study, the main point of the current manuscript is about uniformly self-assembled and self-aligned CNCs patterns using a simple drying process. We successfully aligned the AuNR particles along the CNCs' orientation using this template. We have to mention that the example of the photo-thermal heating effect could be one of the potential applications. It might be true that the temperature enhancement around 1-3 °C could be limited if the main and critical results are simply focused on the temperature increase. Furthermore, if the photothermal heating application is the major goal of the current study, we need to perform more and intense systematic experiments. In addition, if we can increase the concentration of the AuNR particles in the template, we might have more chances to enhance photothermal effects. However, this study is not the case.

To be honest, although the photothermal heating effects are not quite powerful, we need to say that the current photothermal effect is quite enough to prevent icing and frost from Figure 5e and SI Movie S11 and S12. In this study, we could not test the effect of the concentration of the AuNR particles on the photo-thermal heating effect due to the out of research scope. Here, we used a relatively low concentration of AuNR particles. We aimed to highlight the potential of CNC-AuNR films for utilization in anti/de-icing applications. We believe that if we could increase the number of AuNR particles on the CNCs matrix. Additionally, based on our review of previous works on the photothermal effect of AuNRs, a majority of the studies has primarily focused on generating the photothermal effects using high-power infrared lasers on AuNRs dispersed within a bulk liquid medium. There have been limited investigations involving the use of uniformly dried and oriented CNC-AuNR films to induce and enhance the photothermal effect. Hopefully, we will continue and further explore the better photo-thermal effects by varying the concentration of the AuNR particles with CNCs dried pattern.

For the time-dependent temperature measurement case, we believe that the SI movies 8-10 already clearly showed the temperature distribution. While the light was illuminated, the

temperature contour values were not changed. As we already mentioned in the manuscript, the temperature change was spontaneously changed by the ambient light.

Change to the manuscript: To consider the referee's comment, we added some expression how we can improve the temperature enhancement effect. We also little bit toned down about the effect of the photo-thermal heating effect in the revised manuscript.

On p. 7, "... further improved by fine-tuning the aspect ratio of anisotropic GNRs⁷³ and increasing the concentration of GNRs."

On p. 3, The multi-array CNC-GNR metasurface was experimentally demonstrated to ~~have~~ show potential for anti- and de-icing ~~performance capabilities~~. From this result, we believe that this drop-casting CNC-GNR metasurface array can be potentially used as a plasmonic photothermal film for anti-icing applications.

On p. 6, "We ~~evaluated~~ tested the plasmonic ..."

On p. 6, "... the performance of the metasurface in ~~practical~~ potential applications ..."

On p. 6, "...plasmonic multi-array pattern of the CNC-GNR film ~~holds promise as could be~~ applied for an anti-icing surface. ..."

On p. 6, "... self-assembly method, can be potentially utilized as plasmonic thermal heaters. ..."

On p. 6, "... Compared to conventional anti-icing systems, the CNC-GNR metasurface has ~~significant~~ several advantages, such as requiring no substrate..."

9. Referee's comment: More critical point is that there is no explanation why better photothermal conversion can be obtained for the AuNRs without agglomeration. The deep-insight about the mechanism is lack.

Our response: Thank you for the comment. As we showed in the submitted manuscript, if there is a strong coffee-ring pattern of AuNRs and CNCs, the nanoparticles could be localized, which induces aggregation and/or agglomeration of nanoparticles. In this case, the photo-thermal heating effect is degraded compared to the well-aligned and well-distributed case, as shown in Figure 5b. As the referee may know that as the size of spherical nanoparticles increases, their photothermal efficiency decreases due to two factors: an increase in light extinction through Mie scattering and a decrease in the surface-to-volume ratio where surface plasmon resonance can occur (J. Phys. Chem. C 2013, 117, 27073-27080). We chose rod-shaped nanoparticles, which exhibit stronger plasmon resonance than spherical particles, resulting in a higher temperature change (J. Phys. Chem. C 2020, 124, 17172-17182). Considering the possibility of rod-shaped particle agglomeration, we speculate that their aggregates would resemble spheres, reducing the unit aspect ratio, and larger units would have a smaller surface-to-volume ratio for electric plasmon resonance with light extinction through Mie scattering.

Change to the manuscript: To reflect the referee's concern, we amended our manuscript as

follows.

P. 5, "... In the absence of the CNCs, most of the GNR particles were accumulated at the contact line, i.e., coffee-ring stain, as shown in Fig. S6. Then, uniform deposition and alignment of GNR particles cannot be achieved. Figure 4a outlines the procedures in detail,"

P. 6, "... ultimately degrading their photothermal effect⁶⁹ due to two main factors, i.e., increase in light extinction through Mie scattering and a decrease in the surface-to-volume ratio where surface plasmon resonance⁷⁰. Therefore, ..."

Reviewer #2 (Remarks to the Author)

1. Referee's feedback: The authors have answered all doubts and questions satisfactory and have corrected the manuscript according the recommendations, so it can be publish in the present form.

Our response: We pretty much thank to the referee's support!

Reviewer #3 (Remarks to the Author)

The authors have performed substantial revisions, but in this reviewer's opinion, a few critical items must be determined and/or corrected prior to publication.

1. Referee's comment: The manuscript is still missing some critical characterization information for the starting CNC suspensions – in particular, nothing is mentioned about surface charge density, e.g. determined via conductometric titration, which was brought up by multiple reviewers and is critical to ever be able to replicate the results.

Our response: Thank you for providing your valuable insight. We recognize the significance of considering parameters such as surface charge when evaluating particle interactions. Following your suggestion, we employed conductometric titration to quantify the sulfate half-ester content. This process involved immersing an ion exchange resin (Amberchrom™ 50WX8, hydrogen form, 50-100 mesh, Sigma Aldrich) in a 2 wt% CNC aqueous solution (100 mg/ 5 mL) to fully protonate the charged groups on the CNC surface. To facilitate this, we placed 25 mg of resin within a batch enclosed by a dialysis membrane for 48 hours with constant stirring. Subsequently, we diluted the solution to 0.1 wt% of CNC solution and introduced NaCl to the dispersion as an electrolyte, resulting in a final concentration of 1 mM.

Given the strong acidity of the sulfate groups, direct titration of the aqueous CNC dispersion was viable using the strong base NaOH. This approach resulted in a titration curve displaying a distinct transition point. We added 100 µl of 5.5 mM NaOH at each interval and allowed the conductivity to stabilize (typically over 1 minute). Following each NaOH addition, we recorded the corrected values using a conductivity meter (DiST® 5 EC/TDS/Temperature Tester - HI98311, HANNA instruments). The graph indicates the equivalence points at ~5 ml of 5.5 mM NaOH.

Through the analysis of the outcomes, we derived a surface charge density of ~275 mmol/kg for our CNCs, aligning with the reported range for other sulfate-CNCs.

[References]

[A] Abitbol, T., Kloser, E., & Gray, D. G. (2013). Estimation of the surface sulfur content of cellulose nanocrystals prepared by sulfuric acid hydrolysis. *Cellulose*, 20, 785-794.

[B] Foster, E. J et al., (2018). Current characterization methods for cellulose nanomaterials. *Chemical Society Reviews*, 47(8), 2609-2679.

Change to the manuscript:

On p. 11, "Instead, the PEG-GNR employed a steric hindrance effect to prevent particle aggregation, ensuring the colloidal stability in the CNC-GNR ink, as depicted in Fig. 4a. This was further validated through conductometric titration analysis [HI 98311 (DiST®5), HANNA instruments, Italy]. The analysis showed that the CNC particles are well dispersed in the solvent due to electrostatic repulsion with an approximate value of 275 mmol·kg^{-191, 92}, as shown in Fig. S14."

On p. 9 in Supplementary Information,

Figure S14. Conductometric titration curves for protonated sulfate half-ester CNCs. The equivalent volume was estimated by the intersection of negative (red) and positive (blue) lines.

2. Referee's comment: The authors still have only shown preliminary potential for anti-icing and de-icing. Saying that these extremely limited one-off laboratory tests demonstrate this capability is misleading to the scientific community. At minimum, the authors should reduce the boldness of their assertion and correctly indicate that their experiments “suggest potential” anti-icing and de-icing capability of the materials.

Our response: Thank you for the referee's kind suggestion. Although our method has some advantages, for instance no substrate modification or no need for anti-icing liquid spraying, and heat wires, the demonstrated results could be limited as the referee said. Therefore, to reflect the referee's suggestion, we tried to focus on the main results and tone down the impact of the anti-icing and de-icing parts in multiple places of the revised manuscript. Please see the following modifications.

Change to the manuscript:

On p. 2, “To sum up, the key factor of the self-aligned and uniform CNC matrix was the controlled deposition rate of nanoparticles at the self-receding contact line due to the empirically optimized ink composition. As a result, a uniform CNC matrix was successfully fabricated.”

On p. 3, The multi-array CNC-GNR metasurface was experimentally demonstrated to **have** show potential for anti- and de-icing **performance capabilities**. From this result, we believe that this drop-casting CNC-GNR metasurface array can be potentially used as a plasmonic photothermal film for anti-icing applications.

On p. 6, “We **evaluated** tested the plasmonic ...”

On p. 6, “... the performance of the metasurface in **practical** potential applications ...”

On p. 6, “...plasmonic multi-array pattern of the CNC-GNR film **holds promise as** could be applied for an anti-icing surface. ...”

On p. 6, "... self-assembly method, can be potentially utilized as plasmonic thermal heaters. ..."

On p. 6, "... Compared to conventional anti-icing systems, the CNC-GNR metasurface has ~~significant~~ several advantages, such as requiring no substrate..."

3. Referee's comment: The authors continue to assert that they performed an optimization study (e.g. Figs 1 and 3). No design of experiment nor sequential experimentation is provided. The authors have identified a set of parameters within their design space that provides the specific results, but they are far from optimized. Therefore, the authors should adjust all references to anything being "optimized" but rather they have identified a set of conditions within which they observe the stated results.

Our response: Before talking about the optimization, we need to say that our group has been contributing to work a lot on the dried pattern by varying the solvent type, the concentration of the mixtures, the substrate condition, the seeded particles and additives, and so on. We are pretty confident in the novel skill to obtain a uniform pattern of the suspended particles after a complete drying. In terms of that, in this study, the optimization indicated the drop-casting drying process for the method of coffee-ring-less (uniform) and homogeneous quadrantly-aligned cellulose nanocrystal matrices. In fact, initially, we had empirically performed several trial-and-errors to achieve our results. During the trial and errors, we have learned several facts including the effect of the concentration of MeOH and CNCs. Due to the limit of the manuscript, we could not put all the cases. We tried to summarize the major parts of our manuscript. We believe that if anyone uses the same composition of materials, the current condition would be the best to achieve the coffee-ring-less (uniform) and homogeneous quadrantly-aligned cellulose nanocrystal matrices with GNRs.

Nonetheless, we did our best to minimize the referee's concerns. So, we have modified the text as below.

Change to the manuscript:

On p. 2, "To sum up, the key factor of the self-aligned and uniform CNC matrix was the controlled deposition rate of nanoparticles at the self-receding contact line due to the empirically optimized ink composition. ~~As a result, a uniform CNC matrix was successfully fabricated.~~"

On p. 6, "... To achieve better uniformity, we ~~optimized set~~ the composition ..."

Reviewer #4 (Remarks to the Author)

1. Referee's comment: The manuscript has been significantly improved and the authors have addressed many comments by the reviewers. There are however still several things that need to be covered or explained for clarity and understanding:

Our response: Thank you for the referee's countless help to improve the quality of the manuscript. We have carefully answered the questions and addressed the comments raised by the referee; see our point-by-point responses.

2. Referee's comment: Crystallinity and surface charge of CNCs needs to be determined. It has been shown before that small differences here have a major effect on the assembly and diffusion behaviour of CNCs. Zeta potential is good for colloidal stability but depends on other factors than the surface charge as well.

Our response: Thank you for providing your valuable insight. We recognize the significance of considering parameters such as surface charge when evaluating particle interactions. Following your suggestion, we employed conductometric titration to quantify the sulfate half-ester content. This process involved immersing an ion exchange resin (Amberchrom™ 50WX8, hydrogen form, 50-100 mesh, Sigma Aldrich) in a 2 wt% CNC aqueous solution (100 mg/ 5 mL) to fully protonate the charged groups on the CNC surface. To facilitate this, we placed 25 mg of resin within a batch enclosed by a dialysis membrane for 48 hours with constant stirring. Subsequently, we diluted the solution to 0.1 wt% of CNC solution and introduced NaCl to the dispersion as an electrolyte, resulting in a final concentration of 1 mM.

Given the strong acidity of the sulfate groups, direct titration of the aqueous CNC dispersion was viable using the strong base NaOH. This approach resulted in a titration curve displaying a distinct transition point. We added 100 ul of 5.5 mM NaOH at each interval and allowed the conductivity to stabilize (typically over 1 minute). Following each NaOH addition, we recorded the corrected values using a conductivity meter (DiST® 5 EC/TDS/Temperature Tester - HI98311, HANNA instruments). The graph indicates the equivalence points at ~5 ml of 5.5 mM NaOH.

Through the analysis of the outcomes, we derived a surface charge density of ~275 mmol/kg for our CNCs, aligning with the reported range for other sulfate-CNCs.

[References]

[A] Abitbol, T., Kloser, E., & Gray, D. G. (2013). Estimation of the surface sulfur content of cellulose nanocrystals prepared by sulfuric acid hydrolysis. *Cellulose*, 20, 785-794.

[B] Foster, E. J et al., (2018). Current characterization methods for cellulose nanomaterials. *Chemical Society Reviews*, 47(8), 2609-2679.

Change to the manuscript:

On p. 11, "Instead, the PEG-GNR employed a steric hindrance effect to prevent particle aggregation, ensuring the colloidal stability in the CNC-GNR ink, as depicted in Fig. 4a. This was further validated through conductometric titration analysis [HI 98311 (DiST®5), HANNA instruments, Italy]. The analysis showed that the CNC particles are well dispersed in the solvent due to electrostatic repulsion with an approximate value of 275 mmol·kg^{-191, 92}, as shown in Fig. S14."

On p. 9 in Supplementary Information,

Figure S14. Conductometric titration curves for protonated sulfate half-ester CNCs. The equivalent volume was estimated by the intersection of negative (red) and positive (blue) lines.

3. Referee's comment: The width of the CNCs seems excessively large for MCC-based CNCs. Average widths of 20 nm are usually found for tunicates. For MCC one would expect 6-10 nm especially after 60 min hydrolysis time. That is where crystallinity determination with Rietveld refinement can also help since the cross-section determined from Cagliotti fitting should match up with the AFM results.

Our response: In response to the reviewer's feedback, we endeavored to investigate the dimensions of a CNC particle by utilizing a powder X-ray diffraction pattern. However, we regret to inform you that currently, our institute lacks the necessary module to analyze the diffraction pattern using Rietveld refinement with Cagliotti fitting. Nevertheless, we were able to approximate the height profile of the CNC through AFM to characterize the cross-section of the single crystal. The height of the CNCs was determined using the SPM analysis software 'Gwyddion', developed by the Department of Nanometrology, Czech Metrology Institute.

Our analysis of a dataset comprising 279 instances in distilled water (DIW) and 210 instances in a methanol-mixed solvent (DIW-MeOH) revealed an average height distribution of 5.08 ± 1.42 nm for DIW and 5.45 ± 1.48 nm for DIW-MeOH concerning the CNCs. These findings concerning CNC width and length align with prior reports suggesting that the CNCs obtained from cotton (the source of MCC from Sigma-Aldrich) are composed of a few laterally-bound crystallites (see references A and B).

Dimension (nm)		Length (L)	Width (W)	Height (H)
Solvent	Pure DIW	106.46 ± 41.16	23.00 ± 6.67	5.08 ± 1.42
	DIW+MeOH mixture	103.63 ± 42.63	24.00 ± 9.63	5.45 ± 1.48

[References]

[A] Neto, W. P. F., Putaux, J. L., Mariano, M., Ogawa, Y., Otaguro, H., Pasquini, D., & Dufresne, A. (2016). Comprehensive morphological and structural investigation of cellulose I and II nanocrystals prepared by sulphuric acid hydrolysis. *RSC Advances*, 6(79), 76017-76027.

[B]. Parker, R. M., Guidetti, G., Williams, C. A., Zhao, T., Narkevicius, A., Vignolini, S., & Frka-Petesic, B. (2018). The self-assembly of cellulose nanocrystals: Hierarchical design of visual appearance. *Advanced Materials*, 30(19), 1704477.

Change to the manuscript:

On p. 11, “~~The averaged length and width of CNCs are 100–300 nm and 10–20 nm, respectively, so the aspect ratio is 5 to 30~~The length, width, and height of CNCs are 104.05 ± 41.90 nm, 23.50 ± 8.15 nm, and 5.27 ± 1.45 nm, respectively, so the aspect ratio is 2 to 38 (see Fig. S4a-d and Fig. S12a).”

On p. 3 in Supplementary Information,

Figure S4. Characterization of CNC particles and CNC films by Atomic Force Microscope (AFM). a, Representative AFM images of CNCs dispersed in DI water. b, Histogram of ~~width and length~~length, width, and height distribution of CNC particles in DI water. c, Representative AFM images of CNCs dispersed in a MeOH and DI water mixture. d, Histogram of ~~width and length~~length, width, and height distribution of CNC particles in a MeOH and DI water mixture. e, AFM images (top) and rotation distribution (bottom) of the CNC film at (i) ~~the outer ring~~the inner, (ii) the intermediate, and (iii) ~~the inner~~the outer ring region in the direction of 40° . The CNC film was created by evaporating a droplet containing MeOH : DI water : CNCs = 62.75 : 33.85 : 3.40 wt%. The order parameter (S) indicates the degree of alignment of the CNCs.

4. Referee's comment: DLS results are surprising (Figure S14). Because this is standard DLS, the size determined by DLS is based on the hydrodynamic radius of the particles. There is only 1 and it is based on the diffusion of the particles. This radius

should also not be the same, or similar, but rather $R_h^3 = (3LR^2)/4$ for rods. So I really can't understand where the DLS results in SI come from... It would be good to provide the autocorrelation functions with fit.

Our response: Thank you for the comment. As noted by the referee, DLS can measure the hydrodynamic diameter (R_h) of particles dispersed in a liquid medium, which represents a singular value distinct from the actual particle size measured by TEM and AFM.

However, in the case of an anisotropic particle [here, gold nanorods (GNRs)], it is common to observe two distinct maximum intensity peaks in the DLS results [A-C] due to its anisotropic geometry. In this condition, the diffusion motion of the particles could differ along their long and short axes. Thus, our DLS results for CNCs and GNRs exhibited two distinct maximum peaks. As mentioned by the referee, these two peak values were slightly larger than the sizes of GNRs measured by AFM and TEM due to the PEG ligands on the surface of the GNRs.

To reflect the referee's concerns, we further provided the z-average hydrodynamic diameter (D_z), as depicted in Fig. S15 [F]. Additionally, we added correlation function graphs for both CNCs and GNRs. These two correlation functions exhibited a smooth decrease over time without notable fluctuations. Furthermore, they displayed relatively high y-intercept values exceeding 0.6, indicating a favorable signal-to-noise. This underscores the good signal quality with minimal interference from noise sources, such as multiple light scattering.

[References]

[A] Liu, H., Pierre-Pierre, N., & Huo, Q. (2012). Dynamic light scattering for gold nanorod size characterization and study of nanorod–protein interactions. *Gold Bulletin*, 45, 187-195.

[B] Kumar, R., Binetti, L., Nguyen, T. H., Alwis, L. S., Agrawal, A., Sun, T., & Grattan, K. T. (2019). Determination of the Aspect-ratio Distribution of Gold Nanorods in a Colloidal Solution using UV-visible absorption spectroscopy. *Scientific Reports*, 9(1), 17469.

[C] Kumar, R., Binetti, L., Nguyen, T. H., Alwis, L. S., Agrawal, A., Sun, T., & Grattan, K. T. (2019). Determination of the Aspect-ratio Distribution of Gold Nanorods in a Colloidal Solution using UV-visible absorption spectroscopy. *Scientific Reports*, 9(1), 17469.

[D] Karmakar, S. A. N. A. T. (2019). Particle size distribution and zeta potential based on dynamic light scattering: Techniques to characterize stability and surface charge distribution of charged colloids. *Recent Trends Mater. Phys. Chem*, 117-159.

[F] Guichard, Y., Schmit, J., Darne, C., Gaté, L., Goutet, M., Rousset, D., Rastoix, O., Wrobel, R., Witschger, O., Martin, A., Fierro, V., & Binet, S. (2012). Cytotoxicity and genotoxicity of nanosized and microsized titanium dioxide and iron oxide particles in Syrian hamster embryo cells. *Annals of Occupational Hygiene*, 56(5), 631-644.

Change to the manuscript:

On p. 9 in Supplementary Information,

Figure S15. Size distribution of CNC and PEG-GNR particles in a MeOH and DI water mixture. Dynamic Light Scattering (DLS) analysis was conducted to determine the size distribution of the particles and assess their colloidal stability in the binary mixture containing 70% MeOH and 30% DI water. The correlation functions (on the left) and intensity-based particle size distributions (on the right) of (a) CNC and (b) PEG-GNR particles were measured. D_z represents the z-average hydrodynamic diameter. The particle concentrations were approximately 1 mg/mL.

5. Referee's comment: Figure 3b and c are determined with 2.85 wt% CNCs yet all experiments with GNRs are done with 3.4 wt% CNCs. Why this discrepancy? CNCs and GNR could affect flow fields so this could be important.

Our response: Thank you for the comment. In our optimization study aimed at achieving uniformly dried CNC films, we primarily manipulated two key experimental parameters: (i) the mixing ratio of MeOH and DI water (refer to Fig. 1b-d), and (ii) the CNC concentration (refer to Fig. 1e). Here, the parameter (i) was closely related to the evaporation speed, specifically the self-dewetting speed (U_d). In the experiments for (i), we added 2.85 wt% CNC into the mixture of MeOH and DI water. Thus, Fig. 3(b and c) were determined with 2.85 wt% CNC. Of course, we agreed with the reviewer's point that an increase in the CNC concentration could influence the internal flow fields. However, we have already shown that the time-

dependent relative speed ratio (U_d/U_c) trend was not significantly changed even when the CNC concentration increased from 2.85 wt% to 3.40 wt% (compare Fig. 3c and Fig. S5c).

6. Referee's comment: Homogeneous deposits can be obtained with 70% MeOH in water, yet figure 4a suggests mixing of CNCs and GNR is done in 30% MeOH in DI water. I assume the figure is incorrect?

Our response: Thank you for your keen eye! We appreciate your help pointing out the typo, and apologize for the confusion. We rectified the erroneous MeOH and DI water mixture ratio in Fig. 4a of the revised manuscript.

Change to the manuscript:

On p. 17,

7. Referee's comment: Figure 5b. Why is the coffee ring film for 0.56 wt% GNR missing? This should be included since it is valuable.

Our response: Thank you for the comment. As shown in Fig. 5b, we already checked how the deposition uniformity of GNRs altered the temperature distribution determined by the photothermal effect of the oriented GNRs on a CNC template, with the GNR concentration of 0.28 wt%. When GNRs are deposited in a ring-like shape, the temperature distribution becomes relatively broader and exhibit a relatively low peak value in the middle of the dried pattern. Additionally, the presence of GNR particles concentrated near the edge of the GNR film slightly increased the substrate temperature outside the pattern surface. On the contrary, a uniform GNR pattern showed a relatively narrow temperature profile with a notably higher peak. This sufficiently shows that better drying uniformity can improve the photothermal performance of the GNR films.

If, as suggested by the reviewer, an evaluation of the photothermal performance for the coffee ring pattern at 0.56 wt% GNRs is to be conducted and compared with the current results, it will be necessary to redo the experiment for all the cases for a reliable comparative analysis. This will involve reinstalling the experimental set-up, such as the light illumination system, IR camera set-up, and stages. In addition, unfortunately, for now, we cannot use IR camera, and one of our co-authors responsible for the GNR synthesis has relocated abroad for his new position. Therefore, the GNR and CNC template preparation would be difficult at the moment. Thus, the optimization processes for GNR synthesis and photothermal performance of GNR films might take longer. We partially understand that the case could be valuable, but the current comparison between the uniform and coffee-ring film with $C_{GNR} = 0.28$ wt % would be enough to see the effect of the uniformity. Therefore, we are unsure whether the new experimental result for $C_{GNR} = 0.56$ wt % is a prerequisite or not. Hopefully, we hope for the reviewer's understanding in light of our current circumstances. The reviewer's consideration would be greatly appreciated.

From our perspective, the reviewer might be concerned about the relatively small temperature difference between the coffee ring and uniform patterns. However, in this study, we used a relatively low concentration of GNR particles and low light power ($< 1\text{W}\cdot\text{cm}^2$). If the concentration of GNRs increases further, the photothermal performance might be enhanced [A], resulting in a more significant temperature difference between the coffee-ring film and the uniform film. We believe that in this scenario, despite the increase in GNR concentration, the coffee ring pattern will still have lower peaks and a broader temperature profile compared to the uniform pattern.

We would appreciate your acknowledgement of the primary objective of this study was to uniformly orient and deposit GNRs on a CNC template. In addition, we aimed to highlight the potential of CNC-GNR films for utilization in anti/de-icing applications. Please let us know if you can further clarify any specific and essential reason behind conducting a photothermal evaluation experiment on the coffee-ring film with the GNR concentration of 0.56 wt%. Then we need more time to prepare the sample and measurement setup and we will consider attempting it. Thank you!

[References]

[A] Al-Sagheer, L. A. M., Alshahrie, A., & Mahmoud, W. E. (2021). Facile approach for developing gold nanorods with various aspect ratios for an efficient photothermal treatment of cancer. *Colloids and Surfaces A: Physicochemical and Engineering Aspects*, 618, 126394.

8. Referee's comment: Figure S3 does not show the normal nematic structuring seen in dried CNC films (e.g. work by Vignolini and by McLachlan).

Our response: Thank you for the referee's comment. Upon reviewing previous works by Vignolini and McLachlan [A-C], it becomes evident that their focus lay on the twisted structure of CNCs, commonly referred to as the chiral nematic (or helical/cholesteric) structure, which is responsible for structural coloration, rather than nematic structures. The evaporation-induced self-assembly (EISA) phenomena of cellulose nanocrystals (CNCs) typically leads to two distinct self-assembly structures; helical structures and uniaxially aligned structures. The formation of these structures is contingent upon specific evaporation conditions. Specifically, helical structures were frequently observed in a relatively slow and weak flow. In contrast, the uniaxially aligned patterns were observed if the internal flows were relatively fast. Based on the literature and our preceding studies, an evaporatively-driven capillary flow (i.e., coffee-ring flow) and Marangoni flow in a droplet could be fast enough to create the uniaxially aligned patterns.

Our study focused on the uniaxially aligned structure of CNCs driven by EISA in the dynamic flow field of the evaporating millimeter-sized binary mixture droplet. During the evaporation, we observed relatively dynamic flow patterns such as solutal-Marangoni flow and coffee-ring flow during evaporation using μ -PIV. These flows contribute to the formation of the uniaxial structure, rather than helical structure. Under this circumstance, in the Introduction section, we dealt with only the uniaxial alignment structure of CNCs. To provide a clear understanding for the reviewer, we explained and compared two self-assembly mechanisms and resulting structures [(i) the helical (Bouligand) structure induced by chiral nematic phase and (ii) the uniaxially aligned structure induced by flow-induced nematic phase] of CNCs as follows. Additionally, we added some sentences for a more detailed explanation in the revised manuscript.

i) Helical (Bouligand) structure induced by chiral nematic phase

In literature studies investigating the chiral self-assembly of CNCs, they prepared bulk CNC solutions into a petri-dish with a diameter ranging from 30 mm to 100 mm [D-F]. The evaporation speed is very slow in this situation due to their large initial volume. Additionally, the Bond number is much larger than unity. Here, $Bo = \rho g R^2 / \gamma = \text{Gravitational effect} / \text{Surface tension effect} = (R/l_{\text{cap}})^2 \gg 1$, where ρ is the liquid density, g is the gravitational acceleration, R is the radius of the petri-dish (or droplet), γ is the surface tension, and l_{cap} is the capillary length [= $(\gamma/\rho g)^{1/2}$]. For the MeOH and DI water mixture with 70 : 30 vol.% ratio, l_{cap} is approximately 1.9 mm, and the Bond number is $\sim O(10^3)$. Therefore, the dominant gravitational effect results in a relatively flat liquid-gas interface, leading to even evaporation flux profiles [G]. Consequently, in such a bulk system, we would observe relatively weak Marangoni flow and coffee-ring flow, which are in stark contrast to the evaporating droplet that exhibits uneven evaporative flux profiles. Moreover, the meniscus region, is approximately on the order of $l_{\text{cap}} \sim O(1 \text{ mm})$, is very small compared to the diameter $\sim O(10 \text{ mm})$ of the petri-dish. At the same time, we can also consider the Richardson (Ri) number, which is the ratio between the buoyancy term and the flow gradient term. Namely, $Ri = gH/v^2$, H is the depth of the petri-dish $\sim O(10 \text{ mm})$ and v is the Marangoni flow speed. Here, v can be approximately scaled as $v \sim \Delta\gamma H / \mu R$ ($\because \Delta\gamma/R \sim \mu v/H$), where $\Delta\gamma/R$ is the Marangoni

stress, $\mu v/H$ is the viscous stress, $\Delta\gamma$ is the surface tension gradient, and μ is the dynamic viscosity. Eventually, Ri is much larger than unity, so the gravitational effect is still predominant. Therefore, under this circumstance, CNCs exhibited a cholesteric phase, which is a thermodynamically stable state during the deposition process.

[Detailed calculation]

$$\rho \approx 850 \text{ kg} \cdot \text{m}^{-3} \text{ for MeOH and DI water mixture (} = 70 : 30 \text{ vol.\%)}$$

$$\gamma \approx 30 \text{ mN} \cdot \text{m}^{-1} \text{ for MeOH and DI water mixture (} = 70 : 30 \text{ vol.\%)}$$

$$g \approx 9.81 \text{ m}^2/\text{s} \text{ for MeOH and DI water mixture (} = 70 : 30 \text{ vol.\%)}$$

$$\mu \approx 1 \text{ mPa} \cdot \text{s} \text{ for MeOH and DI water mixture (} = 70 : 30 \text{ vol.\%)}$$

$$R \approx 40 \text{ mm} \text{ for the petri-dish and } R \approx 1.5 \text{ mm} \text{ for the droplet}$$

$$H \approx 10 \text{ mm} \text{ for the petri-dish}$$

$$\Delta\gamma \approx 0.1 \text{ mN} \cdot \text{m}^{-1} \text{ [L]}$$

$$l_{\text{cap}} = (\gamma/\rho g)^{1/2} \approx [30 \times 10^{-3}/(850 \times 9.81)]^{1/2} \approx 0.0019 \text{ m}$$

$$Bo = (R/l_{\text{cap}})^2 \approx (40 \times 10^{-3}/0.0019) \approx 445 \gg 1 \text{ for the petri-dish (diameter: 80 mm)}$$

$$Bo = (R/l_{\text{cap}})^2 \approx \left(0.15 \times \frac{10^{-3}}{0.0019}\right) \approx 0.6 < 1 \text{ for the droplet (diameter: 3 mm)}$$

$$Ri = gH/v^2 \sim (g/H) \left(\frac{\mu R}{\Delta\gamma}\right)^2 \approx (9.81/0.01) \left(\frac{0.001 \times 0.04}{0.0001}\right)^2 \approx 157 \gg 1$$

ii) Uniaxially aligned structure induced by flow-induced nematic phase.

In contrast to the previous studies with bulk CNC solutions, for our millimeter-sized sessile droplets containing CNCs, the Bond number was less than unity, so the surface tension effect was relatively important. Initially, the droplet contact angle was smaller than 45° . As the droplet evaporated, it formed a spherical cap shape with a small radius of curvature liquid-gas interface, where the capillary effect became important. As a result, a non-uniform evaporation flux profile was generated along the droplet interface, which raises relatively strong Marangoni flows and coffee-ring flows within the evaporating droplet. In this particular situation, the cellulose nanocrystals displayed a liquid crystalline phase, but they could not retain their chiral structure [H]. Instead, they oriented themselves along the shear direction within the deposited film [I-K]. However, the global uniformity was not guaranteed.

Consequently, in this study, we excluded a discussion of the approach i) because, to achieve homogenous quadrant CNC film, we considered the droplet configuration and initially generated solutal-Marangoni flows and subsequently utilized the interplay between self-dewetting and coffee-ring flows. This approach led to the uniaxial alignment of CNCs in the dried film.

Please find the updated information in the revised manuscript.

On p. 2, “When a droplet is drop-cast onto a solid substrate and then undergoes evaporation, the liquid-gas interface of the droplet typically takes on a curved shape if the surface tension effect dominates over the gravitational effect, specifically if the Bond number is less than unity and there is a hydrophilic contact angle. ~~This droplet interface shape leads to an uneven~~

~~distribution of the evaporation flux, creating a dynamic flow system inside the evaporating droplet.~~ The low contact angle ($< 45^\circ$) and thin droplet interface shape lead to a non-uniform evaporative flux along the droplet interface, creating an evaporatively-driven capillary flow inside the evaporating droplet. In this situation, the CNCs align in a uniaxial direction rather than a helical structure¹⁸⁻²⁰. However, ~~once the droplet completely evaporates, an inhomogeneous dried morphology²¹ is inevitably generated due to evaporatively-driven capillary flows toward a contact line (so-called coffee-ring flows²²), which is the critical hurdle in coating and patterning application~~ this evaporatively-driven capillary flow (referred to as a coffee-ring flow²²) resulted in an inhomogeneous dried morphology²¹ akin to a coffee-ring stain. This issue is a critical hurdle in coating and patterning applications.”

[References]

- [A] Frka-Petesic, B., Guidetti, G., Kamita, G., & Vignolini, S. (2017). Controlling the photonic properties of cholesteric cellulose nanocrystal films with magnets. *Advanced Materials*, 29(32), 1701469.
- [B] Droguet, B. E., Liang, H. L., Frka-Petesic, B., Parker, R. M., De Volder, M. F., Baumberg, J. J., & Vignolini, S. (2022). Large-scale fabrication of structurally coloured cellulose nanocrystal films and effect pigments. *Nature materials*, 21(3), 352-358.
- [C] Parker, R. M., Guidetti, G., Williams, C. A., Zhao, T., Narkevicius, A., Vignolini, S., & Frka-Petesic, B. (2018). The self-assembly of cellulose nanocrystals: Hierarchical design of visual appearance. *Advanced Materials*, 30(19), 1704477.
- [D] Guidetti, G., Atifi, S., Vignolini, S., & Hamad, W. Y. (2016). Flexible photonic cellulose nanocrystal films. *Advanced Materials*, 28(45), 10042-10047.
- [E] Walters, C. M., Boott, C. E., Nguyen, T. D., Hamad, W. Y., & MacLachlan, M. J. (2020). Iridescent cellulose nanocrystal films modified with hydroxypropyl cellulose. *Biomacromolecules*, 21(3), 1295-1302.
- [F] Dumanli, A. G., Van Der Kooij, H. M., Kamita, G., Reisner, E., Baumberg, J. J., Steiner, U., & Vignolini, S. (2014). Digital color in cellulose nanocrystal films. *ACS Applied Materials & Interfaces*, 6(15), 12302-12306.
- [G] Zargartalebi, H., Hejazi, S. H., & Sanati-Nezhad, A. (2022). Self-assembly of highly ordered micro-and nanoparticle deposits. *Nature Communications*, 13(1), 3085.
- [H] Kádár, R., Spirk, S., & Nypelo, T. (2021). Cellulose nanocrystal liquid crystal phases: Progress and challenges in characterization using rheology coupled to optics, scattering, and spectroscopy. *ACS Nano*, 15(5), 7931-7945.
- [I] P Skogberg, A., Mäki, A. J., Mettänen, M., Lahtinen, P., & Kallio, P. (2017). Cellulose nanofiber alignment using evaporation-induced droplet-casting, and cell alignment on aligned nanocellulose surfaces. *Biomacromolecules*, 18(12), 3936-3953.
- [J] Talantikite, M., Leray, N., Durand, S., Moreau, C., & Cathala, B. (2021). Influence of arabinoxylan on the drying of cellulose nanocrystals suspension: From coffee ring to Maltese cross pattern and application to enzymatic detection. *Journal of Colloid and Interface Science*, 587, 727-735.
- [K] Pritchard, C. Q., Navarro, F., Roman, M., & Bortner, M. J. (2021). Multi-axis alignment of

Rod-like cellulose nanocrystals in drying droplets. *Journal of Colloid and Interface Science*, 603, 450-458.

[L] Pyeon, J., Song, K. M., Jung, Y. S., & Kim, H. (2022). Self-Induced Solutal Marangoni Flows Realize Coffee-Ring-Less Quantum Dot Microarrays with Extensive Geometric Tunability and Scalability. *Advanced Science*, 9(11), 2104519.

Reviewer #5 (Remarks to the Author)

1. Referee's comment: The authors have addressed most of my questions and revised the manuscript accordingly. The quality of the manuscript has been significantly improved. Before accepting, I still have some small questions for the authors.

Our response: Thank you for the referee's countless helps to improve the quality of the manuscript. We have carefully answered the questions and addressed the comments raised by the referee; see our point-by-point responses.

2. Referee's comment: Authors responded that "the aspect ratio of GNRs and the thickness of CNC matrices are two critical parameters in influencing photothermal performance. An increase in the thickness of the CNC matrix could allow for more GNRs to be deposited and co-assembled, resulting in enhanced photothermal performance." But the high concentration of CNC particles in the MeOH and DI water mixture solution can result in agglomeration, indicating that there is a limitation to the CNC concentration that can be effectively dispersed. I suggest authors prepare multilayer coatings by layer-by-layer method to investigate the photothermal performance.

Our response: As the referee pointed out, an optimized or better concentration condition of the GNRs could exist to improve the photo-thermal heating effect. However, the current work aims to show the novel method to achieve uniformly self-assembled and self-aligned CNCs patterns using a simple drying process. We successfully aligned the GNRs along the CNCs' orientation using this template. In this study, we used a relatively low concentration of GNRs and we nicely distributed the GNRs on the CNCs' template (see Figure S7).

To enhance the photothermal performance, the referee's suggestion, multilayer coatings, is very interesting but we also worried that if we have multiple layers of CNCs+GNRs, the transparency of the plasmonic metasurfaces could be an issue. Therefore, we might need more time to characterize the new system. To be honest, we will continue our research to improve the photo-thermal heating effect of the current metasurfaces. Hopefully, we will continue and further explore better photo-thermal effects using new ideas in the near future.

3. Referee's comment: I wonder whether the homogeneity of the coating influences its photothermal anti-icing performance. In this system, CNCs suspensions were used as a template to load the GNRs rods. Hence, the structural colors of as-prepared coatings can be ignored. If we further add GNRs' weight ratio in CNCs/GNRs coatings, whether the photothermal anti-icing performance can be increased?

Our response: Thank you for the comment. It seems that the referee understood quite well about our major point of the current work but simultaneously we also wondered the referee's some misunderstanding. In our work, we do not have any structural color from pure CNCs patterns and CNCs+GNRs patterns. The color images (Figs. 1 and 4) are from the polarized optical microscopy that enables to determine the orientation of the deposited particles.

For the concentration of the GNRs, as we shortly discussed in the above question, there might

have some better concentration ratio of GNRs to improve the photo-thermal heating effect. However, more important factor is to nicely distribute the GNRs on the CNCs template. Therefore, to figure out this issue, we need to investigate further this problem as a new work.

4. Referee's comment: Authors have added the optical photographs in revised Figure 1e, why does this coating lose CNCs structural color? Whether the pure CNCs coating in a mixture of MeOH and water also exhibit colorless?

Our response: In the current condition, without GNRs, the dried pattern of CNCs showed a transparent pattern. There is no structural color. We wondered that if we increase the CNC concentration much further, there will be some other color, however, which is not the current case.

5. Referee's comment: Compared with other light-triggered photo-thermal anti-icing materials, GNRs' advantages should be discussed clearly. Authors respond the GNRs can favorable dispersion in common solvents and easy utilization through drop-casting. As is well known, the water-soluble conjugated polymer such as PEDOT: PSS, also exhibit high photothermal performance. I encouraged the author makes a comparison with the reported photothermal anti-icing materials.

Our response: Thank you for informative comment. As the referee said, we recognized that PEDOT:PSS was extensively employed in studies involving the application of photothermal effects. However, in comparison to gold nanorods, PEDOT:PSS exhibited a relatively lower absorbance in the visible light range (380 - 700 nm), while high absorbance in the NIR (\approx 250 nm) and near-infrared wavelength regions (\approx 780-800 nm) [A]. It implies that the PEDOT:PSS film lacks a significant photothermal effect when exposed to natural light. Even, controlling the absorbance spectrum of the PEDOT:PSS requires a complex additional fabrication process. In contrast, gold nanorods show a relatively higher absorbance in the visible light range, and it is easy to manipulate their absorbance spectrum by adjusting the aspect ratio of the gold nanorod particles [B]. Due to this high absorbance magnitude in the visible wavelength region and ease of control, the gold nanorods can be considered optimal functional nanoparticles for anti-/de-icing.

Furthermore, previous studies on the fabrication of PEDOT and PEDOT-PSS films revealed the necessity of employing a spin-coating and heating process, rather than a straightforward evaporation process, to achieve uniform pattern [C, D]. In light of this, we believed that CNC-GNR films would be more efficient to produce through their streamlined process compared to PEDOT:PSS.

[References]

[A] Said, D. A., Ali, A. M., Khayyat, M. M., Boustimi, M., Loulou, M., & Seoudi, R. (2019). A study of the influence of plasmonic resonance of gold nanoparticle doped PEDOT: PSS on the performance of organic solar cells based on CuPc/C60. *Heliyon*, 5(11).

[B] Al-Sagheer, L. A. M., Alshahrie, A., & Mahmoud, W. E. (2021). Facile approach for developing gold nanorods with various aspect ratios for an efficient photothermal treatment of cancer. *Colloids and Surfaces A: Physicochemical and Engineering Aspects*, 618, 126394.

[C] Oh, J. H., George, G. W., Martinez, A. D., Moores, L. C., & Green, M. J. (2021). Radio frequency heating of PEDOT: PSS. *Polymer*, 230, 124077.

[D] Kim, B., Han, M., & Kim, E. (2019). Photothermally powered conductive films for absorber-free solar thermoelectric harvesting. *Journal of Materials Chemistry A*, 7(5), 2066-2074.

Change to the manuscript:

On p. 7, “Compared to conventional anti-icing systems, the CNC-GNR metasurface has significant several advantages, such as requiring no substrate modification⁴²⁻⁴⁸ or no need for anti-icing liquid spraying⁷⁴, and heat wires⁷⁵. Moreover, in comparison to anti-icing nanomaterials, such as CNTs⁷⁶ and PEDOT:PSS⁷⁷, GNRs are dispersed well in most solvents for easy utilization and have a relatively high absorbance in visible light range, especially under natural lighting conditions, resulting in better anti-icing efficiency. We believe...”

REVIEWERS' COMMENTS

Reviewer #1 (Remarks to the Author):

The authors have properly explained the doubts and revised the the manuscript. It is suggested to be published as it is.

Reviewer #3 (Remarks to the Author):

The authors have sufficiently responded to this reviewer's comments.

Reviewer #4 (Remarks to the Author):

The authors have addressed most comments sufficiently.
I would like to see repeats of the photo thermal effect though. Would be good to have standard deviation between experiments.

Reviewer #5 (Remarks to the Author):

Authors have addressed all my questions point by point, so it can be published in the present form.

Reviewer's feedback and comments to the Authors:

First of all, we are very pleased that all the referees satisfied with our revised version of the manuscript.

In addition, we thank the 4th reviewer for the comment regarding our manuscript. Please find below a list of our responses and modifications regarding the reviewer's concern.

Reviewer #4 (Remarks to the Author)

1. Referee's comment: The authors have addressed most comments sufficiently. I would like to see repeats of the photo thermal effect though. Would be good to have standard deviation between experiments.

Our response: We are happy to have that the referee likes our revision.

To reflect the referee's concern, here, we would like to provide a standard deviation for the experiments. Upon review, we found that the real-time point temperature values in the middle of the CNC-GNR film consistently exhibited a standard deviation within the range of 0.1 - 0.2°C (see Supplementary Fig. 10). This suggests that if CNC-GNR films are fabricated under the same ink composition and evaporation conditions, the temperature rise levels are expected to be measured within standard deviation range of approximately 0.1 - 0.2°C.

We compared the temperature profiles of two different coffee-ring CNC-GNR patterns obtained from previous IR experiments. Both patterns had the same CNC-GNR compositions ($C_{\text{GNR}} \approx 0.28 \text{ wt\%}$) and were exposed to identical light conditions. The temperature line profiles exhibited remarkable similarity, with a standard deviation of approximately 0.1°C, as depicted below.

In response to a reviewer's concern, we have included standard deviation information and time-dependent temperature change graphs for the three cases in Figure 5a-b.

Change to the manuscript:

On p. 6, “For the coffee-ring film case of Fig. S11a, the temperature increased only by less than 2°C with a standard deviation of 0.1°C (see the light greenish profile of Fig. 5b).”

“Therefore, the uniform single dot films with $C_{\text{GNR}} \approx 0.28$ and 0.56 wt% increased up to 3°C and 5°C with an experimental error of $\pm 0.2^\circ\text{C}$, respectively (see the greenish/dark greenish profiles of Fig. 5b and real-time temperature data of Supplementary Fig. 10).”

On p. 7 in Supplementary Information,

Supplementary Figure 10. Real-time point temperature measurement results near the center of the CNC-GNR film. The temperature data were compared depending on two parameters; (i) uniformity of the CNC-GNR films [a light greenish line vs. a greenish line] and (ii) GNR concentrations [a greenish line ($C_{\text{GNR}} \approx 0.28$ wt%) vs. a dark greenish line ($C_{\text{GNR}} \approx 0.56$ wt%)]. All data were plotted starting from the moment thermal equilibrium was almost achieved, which was approximately 20 seconds later. The standard deviations (STDV) and average temperature (T_{avg}) were calculated based on measurement data ranging from 20 - 305 seconds.